# THE FLAW OF AVERAGES: QUANTIFYING UNIFORMITY OF PERFORMANCE ON BENCHMARKS

## ABSTRACT

Benchmarks shape scientific conclusions about model capabilities and steer model development. This creates a feedback loop: stronger benchmarks drive better models, and better models demand more discriminative benchmarks. Ensuring benchmark reliability is therefore essential for trustworthy evaluation and meaningful progress. In this work, we study benchmark reliability from a *distributional* perspective and introduce benchmark HARMONY, which measures *how uniformly a model's performance is distributed across the subdomains of a benchmark*. We posit that high HARMONY is a desirable benchmark property, indicating that the aggregate metric reflects uniform competence across subdomains. Across 19 multiple-choice benchmarks and five model families, we map each benchmark onto a mean-variance plane of HARMONY computed across models, where high mean and low variance signal more reliable evaluation. Our analysis shows that less harmonious benchmarks can give misleading results, since overall accuracy may be disproportionately influenced by specific subdomains. For instance, *ARC-Easy* allocates substantially more mass to *Biological Concepts*, overshadowing other critical subdomains such as Geography, Physics, Chemistry, and Environmental Science. By recommending that HARMONY should be reported alongside accuracy, we reframe evaluation from simple performance averages to a more robust, distributionally reliable measurement of performance.

## 1 INTRODUCTION

Benchmarks lie at the crux of measuring and shaping scientific progress in language models, forming a feedback loop with model development [Reviewer EZ1V: (Hardy et al., 2024)]. Discriminative benchmarks refine learning signals and guide model design, while stronger models expose benchmark limitations and drive the creation of more rigorous evaluations. In this reciprocal process, benchmark reliability is essential to ensure that reported improvements reflect genuine capabilities rather than evaluation artifacts (Ott et al., 2022). Yet, despite its importance, benchmark auditing (Swayamdipta et al., 2020; D'Amour et al., 2020; Sainz et al., 2023) has received far less attention than algorithmic advances (Brown et al., 2020; Ouyang et al., 2022; DeepSeek-AI et al., 2025).

Motivated by this gap, recent work identifies structural issues in popular benchmarks, such as redundancy (Polo et al., 2024; Perlitz et al., 2024b) and uneven data distributions (Huang et al., 2025), that can skew results and mislead interpretations of model capability (Ruan et al., 2024; Ilić & Gignac, 2024). In response, the research community *interrogates the reliability of existing benchmarks*, in addition to proposing new ones [Reviewer EZ1V: (Reuel et al., 2024)]. Rather than treating benchmark gains as definitive, recent work urges caution about what benchmarks measure and how these measurements are obtained (Singh et al., 2025; Heineman et al., 2025). This reframes evaluation as an ongoing measurement challenge, highlighting the need for benchmarks whose properties and limitations are well understood [Reviewer EZ1V: (Gebru et al., 2021)].

In our work, we investigate benchmark reliability from a distributional perspective. Since benchmarks claim to assess competence over a stated domain, we ask whether their data evenly represents its subdomains and whether performance is uniform on these subdomains. We instantiate this idea with benchmark HARMONY, a measurement of performance uniformity among subdomains of a benchmark (§2). Figure 1 illustrates our pipeline: Given a target benchmark, we first partition the datapoints in this benchmark into semantic clusters, where each cluster represents a subdomain

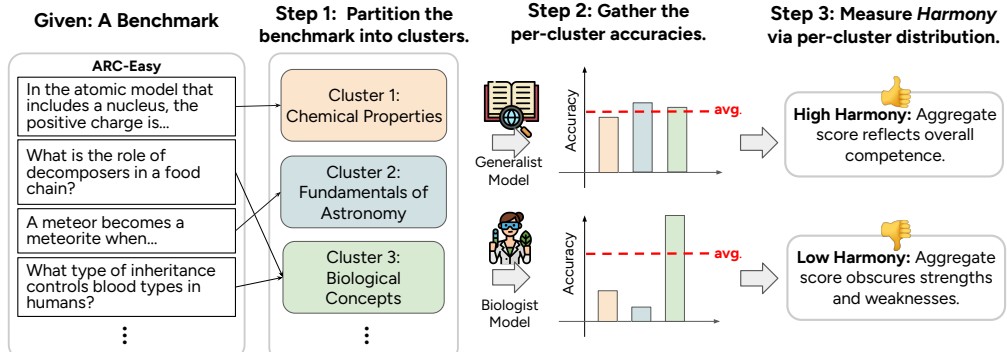

Figure 1: **Pipeline of evaluating HARMONY for a given benchmark.** Step 1: We partition the benchmark into semantic clusters (subdomains or skills). Step 2: We gather each model's performance on every cluster. Step 3: We calculate the harmony — the uniformity of the distribution of performance across subdomains. We posit that high HARMONY implies that aggregate metrics capture broad competence, whereas low HARMONY obscures strengths and weaknesses.

(Step 1); we then gather the performances per subdomain for different models (Step 2); finally, we compute HARMONY for each benchmark-model pair, where high HARMONY suggests that aggregate metrics capture broad competence, while low HARMONY obscures strengths and weaknesses of the model (Step 3).

Using HARMONY, we conduct a range of analyses on a variety of benchmarks and models to assess the reliability of benchmark evaluations (§3, §4). We posit that **high HARMONY is a desirable** benchmark property, since it implies that the benchmark reflects *overall competence* for a given model. This distinction matters because *less harmonious benchmarks can yield skewed aggregate conclusions*. For example, if a benchmark is dominated by one subdomain (e.g., Biological Concepts in ARC-Easy), aggregate accuracy can obscure weaknesses or strengths in other subdomains (e.g., Geography, Physics, Chemistry, and Environmental Science). As a result, a model that masters the dominant subdomain can appear to outperform a more generalist model despite having poorer cross-domain generalization (e.g., biologist model vs. generalist model in Figure 1). We therefore recommend reporting HARMONY alongside accuracy to move from simple averages to a distributionally reliable measurement of competence across subdomains.

In summary, our contributions are twofold:

- We introduce HARMONY, an entropy-based metric that quantifies how uniformly performance is distributed across subdomains in a benchmark. This provides a measure of how well the overall accuracy captures performance consistency across the benchmark's skill domains.
- We provide a large-scale empirical mapping of 19 MCQA benchmarks across five model families in the HARMONY mean-variance plane, revealing the spectrum of benchmark reliability.

## 2 BENCHMARK HARMONY

### 2.1 PRELIMINARIES AND NOTATION

Let $\mathcal{B} = \{(x_i, y_i)\}_{i=1}^N$ be a benchmark consisting of input-output pairs $(x, y)$. Our goal is to understand the *underlying distribution* of $\mathcal{B}$ by inducing a semantic partition $\mathcal{G} = \{A_1, \ldots, A_k\}$ of $\mathcal{B}$, where $A_i \subseteq \mathcal{B}$, $A_i \cap A_j = \emptyset$ for $i \neq j$, and $\bigcup_{i=1}^k A_i = \mathcal{B}$. The partition is guided by a similarity function $\mathcal{S} : \mathcal{X} \times \mathcal{X} \to (0, 1]$ that measures the semantic similarity between data points $x_i, x_j \sim \mathcal{B}$. Lastly, let $f$ be a model and let $\Psi(f; A_i)$ denote a measure of performance (e.g., accuracy) for $f$ computed on a subset $A_i \subseteq \mathcal{B}$.

### 2.2 HARMONY: A MEASURE OF BALANCED COVERAGE AND UNIFORM PERFORMANCE

**Intuition.** Consider a biology benchmark spanning microbiology, animal biology, and plant biology. If microbiology dominates and a model excels only there, the overall score may misleadingly suggest broad competence in biology. Conversely, if microbiology is underrepresented and the model is weak on it but strong elsewhere, the aggregate evaluation may conceal a critical weakness. Moreover, even when subdomains are equal in size, large accuracy gaps make the aggregate metric

uninformative (e.g., 90% accuracy in microbiology and 50% accuracy in plant biology averages to a number that reflects neither). A *harmonious* benchmark therefore mitigates these distortions by balancing coverage and promoting comparable performance across subdomains. [Reviewer F1zp: In contrast, low HARMONY can stem from poor benchmark design (e.g., unbalanced coverage) or genuine heterogeneity in task difficulty. In both cases, the single aggregate metric is an unreliable proxy for broad competence, and that is what Harmony is designed to diagnose. Disentangling design flaws from domain heterogeneity requires domain expertise and is beyond the scope of the metric itself.]

**Formal definition of HARMONY.** Given a partition $\mathcal{G}_f = \{A_i\}_{i=1}^k$, HARMONY measures how uniformly performance is distributed across the subsets in this partition. For each $A_i$, let $w_i = \frac{|A_i|}{|\mathcal{B}|}$ be the size weight and let $\mu = \sum_{i=1}^k w_i \Psi(f; A_i)$ be the weighted mean. We convert differences between $\mu$ and $\Psi(f; A_i)$ into smooth proximity scores via a Gaussian kernel:

$$K_i \;=\; \exp\!\Big(-\big(\tfrac{\Psi(f;A_i)-\mu}{b}\big)^2\Big),$$

where $b > 0$ is a bandwidth parameter.[1] We then form *performance masses*

$$p_i \;=\; \frac{w_i K_i}{\sum_{j=1}^k w_j K_j}, \qquad \sum_{i=1}^k p_i = 1,$$

and compute the HARMONY (normalized Shannon entropy)

$$H(\mathcal{G}_f) \;=\; -\frac{1}{\log k} \sum_{i=1}^k p_i \log\big(p_i + \varepsilon\big) \in [0, 1],$$

with a small $\varepsilon = 10^{-12}$ for numerical stability.

Subsets with accuracies far from $\mu$ receive exponentially smaller $p_i$, lowering entropy. Thus, higher HARMONY $H(\mathcal{G}_f)$ indicates performance that is evenly distributed across subsets, while lower HARMONY captures a more concentrated performance in a few subsets. Therefore, HARMONY quantifies the uniformity of performance while considering the distributional balance.

**Interpreting HARMONY.** Let $\Pi$ be a partitioning rule that maps a benchmark $\mathcal{B}$ and a model $f$ to a partition $\mathcal{G}_f(\mathcal{B}) = \Pi(\mathcal{B}; f)$ using $\mathcal{S}$. Then, define the per-model harmony of $\mathcal{B}$ as

$$H_\mathcal{B}(f) \;:=\; H\big(\mathcal{G}_f(\mathcal{B})\big) \in [0, 1].$$

Given a model set $\mathcal{F}$, we evaluate $\mathcal{B}$ by the cross-model mean and variance

$$\mu_H(\mathcal{B}) \;=\; \mathbb{E}_{f \sim \mathcal{F}}[H_\mathcal{B}(f)], \qquad \sigma_H^2(\mathcal{B}) \;=\; \mathrm{Var}_{f \sim \mathcal{F}}(H_\mathcal{B}(f)). \tag{1}$$

Higher $\mu_H(\mathcal{B})$ indicates that, on average across models, performance is more uniformly distributed across the subsets of $\mathcal{B}$, while lower $\sigma_H^2(\mathcal{B})$ indicates that this property is stable across models. Rather than dichotomizing benchmarks as *good* or *bad*, we adopt a comparative view, where $\mathcal{B}_1$ is preferred to $\mathcal{B}_2$ if it attains a higher expectation and a lower variance.

**Implications.** We approach benchmarks as diagnostic tools rather than scoreboards. [Reviewer EZ1V: Throughout this paper, we use the term *benchmark reliability* to refer to the extent to which a benchmark's aggregate score is a stable and representative summary of a model's performance across its subdomains, rather than being dominated by a small number of subsets or being highly sensitive to the particular model under evaluation.] A benchmark with *high mean* HARMONY and *low cross-model variance* indicates that aggregate metrics consistently capture broad competence rather than artifacts of data composition. In contrast, either *low mean* or *large variance* signals fragility, since the conclusions about model evaluation may depend excessively on a few subdomains and be less reliable. Notably, models with similar aggregate accuracy can differ in HARMONY, implying different breadth of competence. In practice, benchmarks with favorable mean-variance HARMONY profiles enable more trustworthy evaluation, fairer comparisons, and clearer measure of progress [Reviewer EZ1V: and, in our terminology, are therefore more reliable.]

---

[1]We set $b$ by a robust scale of $\{\Psi(f; A_i)\}$. Let $\tilde{a} = \mathrm{median}_i \, \Psi(f; A_i)$ and $\mathrm{MAD} = \mathrm{median}_i |\Psi(f; A_i) - \tilde{a}|$, then $b = \max\{0.02, \; 1.4826 \cdot \mathrm{MAD}\}$.

## 2.3 PARTITION INDUCTION

To compute HARMONY, we require a semantic partition of the benchmark. To this end, we introduce a novel similarity metric named **predictive similarity**, a model-aware similarity between data points based on the divergence of their probability distributions, and induce $\mathcal{G}_f$ via spectral clustering on the resulting affinity matrix $\mathcal{S}$.

**Predictive Similarity.** We define predictive similarity as a model-aware similarity measure that quantifies how similarly a model $f$ distributes probability over the output space for two data points. For $x_i, x_j \sim \mathcal{B}$, let $\bar{p}_f(x)$ denote the model's length-normalized probability distribution over tokens. Then, predictive similarity is computed as

$$S(x_i, x_j) = \exp\left(-\frac{\tau}{2}\Big[D_{\mathrm{KL}}\big(\bar{p}_f(x_i)\,\|\,\bar{p}_f(x_j)\big) + D_{\mathrm{KL}}\big(\bar{p}_f(x_j)\,\|\,\bar{p}_f(x_i)\big)\Big]\right), \qquad (2)$$

where $D_{\mathrm{KL}}(\cdot\,\|\,\cdot)$ denotes the Kullback–Leibler divergence, $\tau$ is a scaling factor chosen as the reciprocal of the median symmetric divergence, and the averaged predictive distribution is given by $\bar{p}_f(x_i) = \frac{1}{T}\sum_{t=1}^{T} p_f(x_i, y_i^{<t})$, with $y_i^{<t}$ denoting the ground-truth prefix up to token $t-1$.[2]

Intuitively, predictive similarity $S$ is large when the model treats $x_i$ and $x_j$ as interchangeable from a predictive standpoint and small when the model sharply distinguishes them. We defer in-depth discussion on different aspects of predictive similarity to Appendix B.

**Clustering.** Given the predictive similarity matrix $S \in (0, 1]^{N \times N}$, we induce the partition of a benchmark via spectral clustering (Ng et al., 2001). We treat $S$ as a precomputed affinity, form the symmetric normalized Laplacian $L = D^{-1/2}(D - S)D^{-1/2}$ with $D = \mathrm{diag}(S\mathbf{1})$, compute the $k$ eigenvectors of $L$ associated with its smallest eigenvalues, and apply $k$-means in this spectral embedding to obtain a partition $\mathcal{G} = \{A_1, \ldots, A_k\}$. To determine the optimal number of subsets, we sweep $2 \le k \le 20$ and select the value maximizing the silhouette score $s(k) \in [-1, 1]$ as an intrinsic compactness diagnostic (Rousseeuw, 1987). [Reviewer EZ1V: Since $S$ defines a weighted similarity graph over benchmark items, spectral clustering is a natural choice. It operates directly on affinity graphs rather than on raw representations, allowing to recover non-spherical and imbalanced clusters that $k$-means on raw representations may fail to separate.]

## 2.4 EMPIRICAL VALIDATION OF PARTITION INDUCTION

We need a controlled benchmark with a known partition to evaluate our method's ability to induce well-defined semantic partitions. We therefore introduce RedundantQA, a synthetic, four-domain[3] MCQA benchmark where each item pairs a reference question with two *true-similar* paraphrases (same underlying knowledge) and two *false-similar* distractors (high lexical overlap, different answers). This structure cleanly separates semantic from lexical similarity and allows us to control underlying data distribution. See Appendix A for construction and

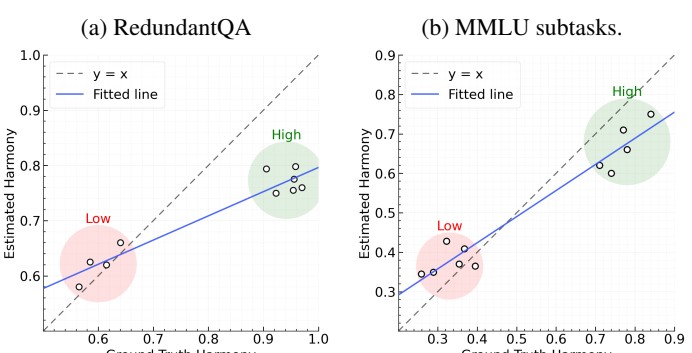

Figure 2: **Validation of our approach on (a) RedundantQA and (b) MMLU high school subtasks.** Estimated HARMONY strongly correlates with the ground truth and clearly separates low from high HARMONY variants. Each dot represents one variant averaged across five random seeds. [Reviewer 85MX: Full results are provided in Fig. 7 and 8.]

---

[2]For $t > 1$, we condition on the ground-truth answer tokens rather than on the model's own autoregressive predictions, ensuring that accumulated model errors do not affect the similarity measure.

[3]Biology, History, Economics, Popular Culture.

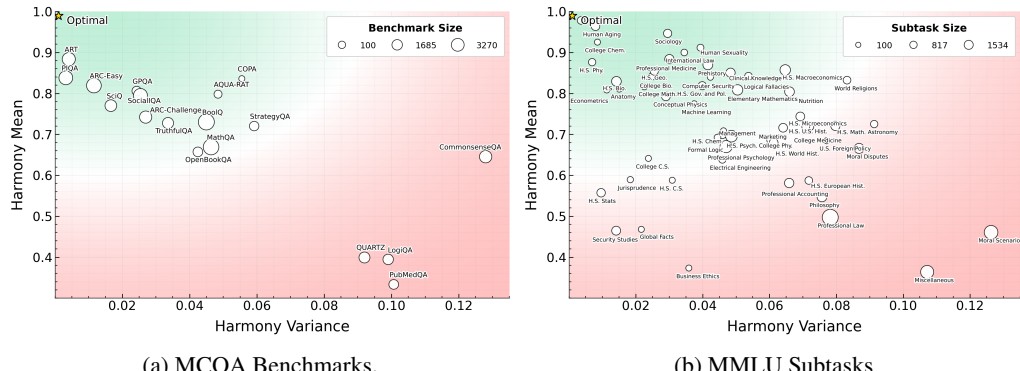

(a) MCQA Benchmarks.

(b) MMLU Subtasks

Figure 3: **Mean-variance plane for HARMONY across (a) MCQA Benchmarks and (b) MMLU subtasks.** Each point represents a benchmark or subtask plotted by the HARMONY mean ($\mu_H(\mathcal{B})$) and variance ($\sigma_H^2(\mathcal{B})$) over 36 models. Upper-left (high mean, low variance) indicates higher *benchmark reliability*; rightward (higher variance) and downward (lower mean) shifts signal diminished reliability. The star at top-left represents an optimal benchmark. Harmony mean/variance are defined in Eq. 1. [Reviewer 5RAY: In (a), Harmony mean and variance show a strong negative Pearson correlation ($r = -0.785$ with $p = 0.0001$), while in (b) the correlation is more moderate ($r = -0.404$ with $p = 0.0025$).]

validation details of Redundan-
tQA, along with representative examples.

We empirically validate our partitioning approach on controlled variants of RedundantQA and a compilation of MMLU high school subtasks.[4] In each variant, we designate a domain as dominant and assign it a proportion $r \in \{0.3, 0.4, 0.5, 0.6, 0.7\}$ of the benchmark, with the remaining domains sharing $1 - r$ equally. This yields a spectrum of distributional imbalance with known ground-truth HARMONY. We repeat every (dominant domain, ratio) variant with five random seeds.

As shown in Fig. 2, HARMONY estimated by our method exhibits a strong positive correlation with the ground-truth HARMONY. This alignment demonstrates that our measure reliably distinguishes between high HARMONY and low HARMONY regimes across different degrees of imbalance. Importantly, the trend persists across different (dominant domain, ratio) variants, indicating that the signal is robust to variations in benchmark construction.

We further validate predictive similarity along three axes and defer all details and results to Appendix B: (i) discrimination of semantic vs. lexical similarity (App. B.2), (ii) recovery of ground-truth domains on RedundantQA and MMLU (App. B.3), and (iii) fidelity of HARMONY estimates under controlled distributional shifts (App. B.4).

## 3    MAIN ANALYSES: HOW HARMONIOUS ARE THE BENCHMARKS?

We now examine how harmonious widely used MCQA benchmarks are. [Reviewer F1zp: Accordingly, we compute HARMONY for each model and benchmark pair (as detailed in §2) and then aggregate it across models to position each benchmark in the mean-variance plane given by $(\mu_H(\mathcal{B}), \sigma_H^2(\mathcal{B}))$ in Eq. 1.] This section first details the experimental setup (§3.1), then maps each benchmark to this plane and provides an interpretation of this mapping (§3.2).

---

[4][Reviewer 85MX: We use four MMLU high school subtasks (Biology, Geography, European History, Computer Science) to span a wide range of dominant ratios while keeping the number of synthetic variants (and thus the figures) interpretable. Adding more subdomains would substantially increase the number of variants and clutter the visualization, without qualitatively changing our conclusions.]

### 3.1 EXPERIMENTAL SETUP

We conduct evaluations using a modified version of `lm-evaluation-harness`[5], covering a wide range of model sizes across five prominent model families: `Llama 3` (Grattafiori & et al, 2024), `Qwen3` (Yang et al., 2025), `Gemma 3` (Team et al., 2025), `Phi-3` (Abdin et al., 2024), and `OLMo 2` (OLMo et al., 2025) (see App. D for full model list). Our setup spans 19 MCQA benchmarks that assess a broad range of model capabilities:

- **Reasoning**: ARC-Challenge (Clark et al., 2018), ARC-Easy (Clark et al., 2018), ART (Bhagavatula et al., 2020), BoolQ (Clark et al., 2019), CommonsenseQA (Talmor et al., 2018), COPA (Roemmele et al., 2011), LogiQA (Liu et al., 2020), PIQA (Bisk et al., 2020), QUARTZ (Tafjord et al., "2019"), SocialIQA (Sap et al., 2019a), StrategyQA (Geva et al., 2021).
- **Mathematical Problem-Solving**: AQUA-RAT (Ling et al., 2017), MathQA (Amini et al., 2019).
- **World Knowledge**: GPQA (Rein et al., 2023), MMLU (Hendrycks et al., 2020), OpenBookQA (Mihaylov et al., 2018a), PubMedQA (Jin et al., 2019), SciQ (Johannes Welbl, 2017).
- **Truthfulness**: TruthfulQA (Lin et al., 2021).

For evaluation, we use average token-level log-likelihood scoring over answer options as implemented in the harness, selecting the option with the highest average log-probability. We follow each benchmark's evaluation protocol as implemented in the harness, using zero-shot evaluation by default, and report accuracy. We focus on MCQA benchmarks because (i) the discrete label space $\mathcal{Y}$ yields unambiguous ground truth and exact accuracy,[6] and (ii) evaluation is automatic and does not rely on a judge, avoiding grading variance that is common in free-form scoring [Reviewer EZ1V: (Chiang et al., 2024)]. However, we note that our methodology extends to free-response benchmarks given an appropriate evaluation metric.

### 3.2 MAPPING BENCHMARK HARMONY

[Reviewer F1zp: We analyze the widely used benchmarks from §3.1 with HARMONY by separately inducing partitions for each model in our suite, computing a Harmony score for each model, and then placing each benchmark in a two-dimensional plane whose axes are the mean and variance of these per-model Harmony scores.]

Fig. 3a and 3b respectively position benchmarks and MMLU subtasks in the cross-model mean-variance plane of HARMONY. The vertical axis is $\mu_H(\mathcal{B})$ (average uniformity of performance across subdomains) and the horizontal axis is $\sigma_H^2(\mathcal{B})$ (stability of that uniformity across models). Moving *upward* increases average distributional uniformity, while moving *leftward* increases cross-model stability. Consequently, the *upper-left* region (high mean, low variance) identifies benchmarks whose aggregate scores consistently reflect broad competence. In contrast, model performances on benchmarks with *low mean* are distributionally skewed on average. If accompanied by *low variance*, this skew is consistent across models (i.e., *consistently fragile*), whereas if accompanied by *high variance*, reliability becomes model-dependent. Thus, *upward* and *leftward* trajectories indicate more reliable evaluation, whereas *downward* and *rightward* shifts suggest more concentrated model performance on a few subdomains and conclusions that vary substantially across models.

## 4 CONTROLLED ANALYSES OF CONFOUNDING FACTORS

In this section, we (i) show how less harmonious benchmarks can distort model evaluations and (ii) examine whether low HARMONY benchmarks warrant extra caution for larger models or those trained with more tokens.

### 4.1 HOW DOES MODEL PERFORMANCE CHANGE WITH INCREASED HARMONY?

We analyze the extent to which less harmonious benchmarks can distort model evaluations via unrepresentative aggregate metrics. To this end, we prune benchmarks using predictive similarity to

---

[5]https://github.com/EleutherAI/lm-evaluation-harness

[6]In free-response settings, multiple plausible intermediate steps can lead to the same answer, complicating ground truth.

eliminate overly similar items. The pruning ratio is set to be inversely proportional to benchmark HARMONY, such that high HARMONY benchmarks receive minimal pruning while low HARMONY benchmarks are pruned more aggressively.[7] By mitigating the skewness, this procedure reveals models' uniform performance on benchmarks.

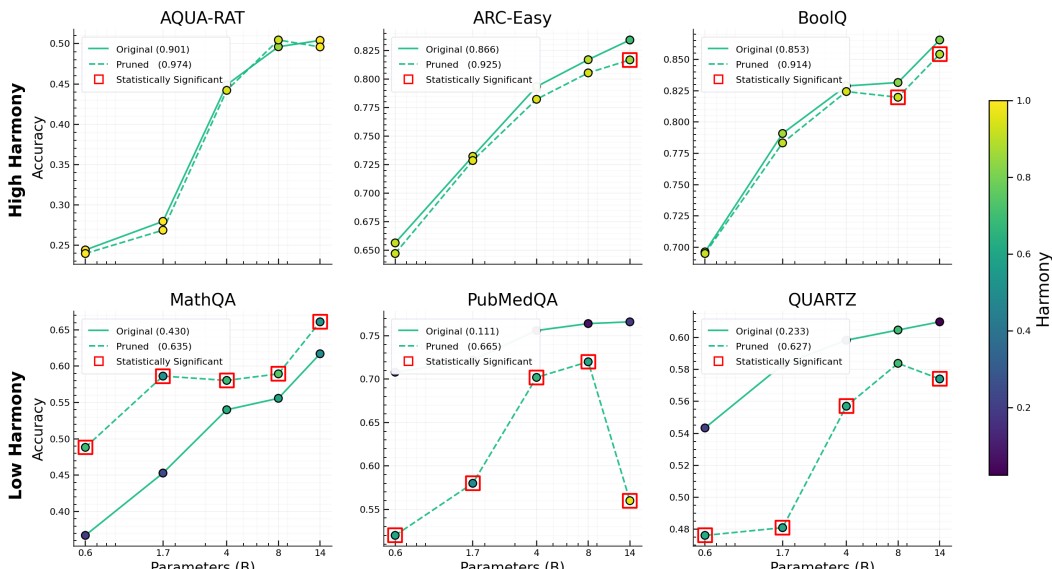

Figure 4: **Balancing benchmarks via pruning.** We remove overly similar items with a pruning rate inversely proportional to HARMONY. *Top row* shows more harmonious benchmarks, where accuracy remains stable as HARMONY increases. *Bottom row* shows less harmonious benchmarks, where HARMONY rises and accuracy shifts significantly. Model-averaged HARMONY values for the original and pruned benchmarks are reported in parentheses in the legends.

As shown in Figure 4, model accuracies on high HARMONY benchmarks remain stable under pruning, with differences that are not statistically significant despite increased HARMONY. In contrast, low HARMONY benchmarks are fragile, where pruning notably improves HARMONY and aligns with statistically significant accuracy changes. Details of our significance tests appear in Appendix G. As HARMONY increases, per-subdomain accuracies tighten around the benchmark mean, making the aggregate a more faithful representation of the underlying accuracy distribution. Overall, low HARMONY benchmarks can be misleading as they skew aggregate scores, whereas high HARMONY benchmarks provide more reliable and representative evaluations.

While Figure 4 illustrates our findings for the Qwen3 family, we provide the comprehensive results for the full experimental setup in Appendix H.

### 4.2 HOW DOES HARMONY CHANGE ACROSS MODEL SIZES AND TOKEN BUDGETS?

Given that low HARMONY signals fragility, we now ask whether this risk depends on model scale or training budget. We therefore seek to characterize how HARMONY scales with model parameters and pre-training budget. Rather than focusing on raw accuracy, our goal is to understand whether larger models or longer pre-training runs yield more uniform performance across subdomains. Concretely, we pose two questions. *Model size:* As parameter count increases within a family, does HARMONY steadily rise, indicating broader competence across subsets? *Token budget:* Along a fixed architecture, does increasing pre-training token budget improve HARMONY, suggesting a more even reallocation of accuracy on the benchmark?

**Model parameters.** We observe that **the relationship between model parameter count and HARMONY is *family-specific* rather than universal**. As shown in Fig. 5, within-family comparisons reveal a negative correlation for Qwen and Llama families, indicating that larger models

---

[7]Specifically, we use the formula $p = \text{clip}_{[0.05, 0.5]}\big(0.05 + (0.5 - 0.05)\big(\frac{1 - \text{clip}(H; 0.1, 1)}{1 - 0.1}\big)^{1.5}\big)$.

in these families concentrate performance more on a few subdomains. In contrast, Gemma and OLMo families exhibit a positive correlation between model size and HARMONY, with larger models distributing accuracy more evenly across the subdomains in the benchmark. This suggests that parameter count alone is not a sufficient indicator of uniformity of the performance.

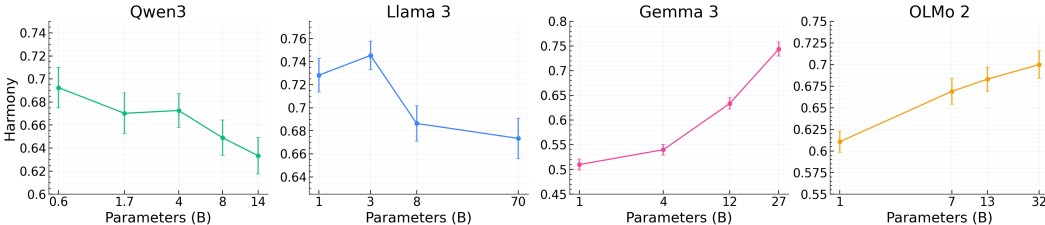

Figure 5: **Model size vs. HARMONY.** Scaling trends are *family-specific*: Qwen and Llama show negative correlations, while Gemma and OLMo show positive correlations (larger models perform more uniformly). Thus, parameter count alone is not predictive of performance uniformity. Y-axis shows each model's average HARMONY over all benchmarks (§3.1).

**Pre-training tokens.** Under a fixed architecture, **increasing the pre-training token budget yields a rise in HARMONY**, after an initial dip. We examine how HARMONY evolves under a fixed architecture by tracking OLMo2 1B and OLMo2 7B across increased token budgets. As shown in Fig. 6, HARMONY dips early and then rises steadily, while aggregate accuracy increases minimally across checkpoints. Thus, we find that the distribution of performance improves (i.e., increased HARMONY) even as aggregate accuracy remains nearly unchanged. In other words, additional pre-training reallocates competence from a few dominant subdomains toward a more uniform spread, yielding a strictly more favorable accuracy profile without changing the aggregate score drastically.

In addition to HARMONY, we also formulate the uniformity of improvements that come with scaling and share our findings in Appendix F. We emphasize that these findings are empirical rather than causal. We leave modeling the mechanisms underlying these trends as valuable future work.

## 5 RELATED WORK

**Assessing Benchmark Reliability.** Beyond proposing new tasks, a growing body of work interrogates the *reliability* of benchmarks themselves. A line of work targets the robustness of the test sets, focusing on building dynamic benchmarks to replace static benchmarks (Kiela et al., 2021; Chiang et al., 2024) and building adversarial perturbations to eliminate spurious cues present in static benchmarks (Nie et al., 2020; Croce et al., 2021). Closely related are concerns about overfitting to public test sets and contamination from pre-training corpora, which can inflate reported gains (Deng et al., 2024; Golchin & Surdeanu, 2024; Roberts et al., 2023; Dong et al., 2024).

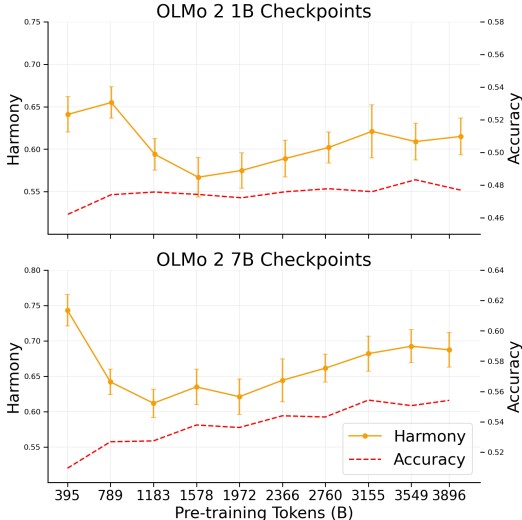

Figure 6: **Pre-training tokens vs. HARMONY.** For OLMo 2 1B/7B, HARMONY dips then steadily rises with more pre-training tokens while aggregate accuracy improves slightly, indicating competence shifts from dominant subsets toward greater uniformity. Y-axis shows each model's average HARMONY over all benchmarks (§3.1).

Barton (2025) analyzes a collection of benchmarks, showing that some benchmarks (e.g. Hellaswag (Zellers et al., 2019)) scale smoothly with increased scale and compute, while others (e.g. CommonsenseQA (Talmor et al., 2018)) do not. Another branch of literature audits data reliability and distributional coverage by introducing shifted test sets to probe generalization (Recht et al., 2019; Taori et al., 2020; Teney et al., 2020) and correcting pervasive label errors in widely used benchmarks

(Northcutt et al., 2021; Gema et al., 2025). Beyond individual datasets, meta-evaluation work proposes frameworks and documentation practices to systematically assess benchmark design, provenance, and intended use (Reuel et al., 2024; Mazumder et al., 2023; Gebru et al., 2021). [Reviewer EZ1V: A complementary line of work focuses on the *validity* of benchmarks. Yin et al. (2019) show that people's trust in a predictive model depends jointly on its reported accuracy on held-out data and its experienced accuracy in use, underscoring that *how* performance is summarized and communicated shapes the inferences stakeholders draw from evaluation metrics. More broadly, Chouldechova et al. (2024) and Salaudeen et al. (2025) import psychometric notions of valid measurement into AI evaluation, articulating how benchmarks should be designed so that scores support specific inferential claims about capabilities, risks, and impacts. In these frameworks, *validity* concerns whether a benchmark score can be interpreted as evidence for a given construct and use case, whereas *reliability* is a narrower requirement that the measurement procedure yields stable and consistent scores. Related work also studies the external validity of benchmarks, such as how well leaderboard gains translate to real-world performance (Ott et al., 2022) and what reported scores actually measure (Dehghani et al., 2021; Singh et al., 2025).] Finally, another line of work separates signal from noise in benchmark results by quantifying variance and prescribing protocols that stabilize rankings in order to make comparative conclusions more reliable (Madaan et al., 2024; Wang et al., 2024a; Heineman et al., 2025). Advancing this field of work, we contribute a distributional perspective on benchmark reliability. Rather than treating a benchmark evaluation as a single score, we model the benchmark as a mixture over the subdomains of the stated benchmark domain. We then measure how *performance mass* is distributed across these subdomains. This perspective diagnoses whether aggregate metrics reflect a broad competence over the benchmark or are dominated by certain subdomains. This perspective diagnoses whether aggregate metrics reflect a broad competence over the benchmark or are dominated by certain subdomains.

**Distributional Frameworks for Efficient Evaluation.** Scaling laws of neural language models suggest that performance improves with model size (Kaplan et al., 2020), encouraging the development of increasingly larger and costlier models. Consequently, there has been growing interest in developing efficient evaluation methods that reduce computational and financial costs without compromising reliability. Perlitz et al. (2024a) introduce a reliability metric that dynamically adjusts compute by performance tier while preserving rank fidelity. Rodriguez et al. (2021) propose Item Response Theory (Tatsuoka et al., 1971) based leaderboards that jointly model difficulty and discrimination to identify examples that best differentiate model performance. Similarly, Polo et al. (2024) propose tinyBenchmarks, an efficient evaluation method that uses IRT to model the discriminative power of benchmark examples, allowing the selection of a small yet representative subset of items that can accurately estimate performance. Vivek et al. (2024) propose anchor point selection to identify small, representative subsets by leveraging cross-model correlations in instance-level predictions. Ethayarajh et al. (2022) identify informative data points via *usable information* (how much input a model family can exploit) extending Shannon information to account for model constraints. Notably, these works introduce distinct metrics such as IRT item parameters, cross-model instance correlations, and information-theoretic usable information to characterize the benchmark distribution and guide principled compression of benchmarks. Ultimately, these metrics enable targeted downsampling (e.g., selecting maximally discriminative or most informative items) that preserves rankings and reduces evaluation cost while maintaining coverage. In contrast, we do not seek cheaper evaluations. We instead assess whether a benchmark reliably measures its stated domain and, where it does not, we question the original evaluation rather than preserve it.

Prior work mainly (i) proposes new or dynamic tests, (ii) stabilizes leaderboards through variance control and guidelines, and (iii) compresses evaluation via discriminative selection. We instead audit *existing* benchmarks through a distributional lens, modeling a benchmark as a mixture over subdomains and measuring whether models spread accuracy uniformly. Unlike efficiency work that preserves overall scores while reducing cost, HARMONY reveals where aggregate metrics fail to provide a representative understanding of model competency. Our method is post hoc and lightweight, complements robustness and contamination audits, and yields practical guidance: report HARMONY with accuracy and rebalance low HARMONY benchmarks.

We further discuss additional related work on language model evaluation in Appendix K.

## 6 CONCLUSION

We introduce HARMONY, an entropy-based measure of how uniformly performance is distributed across a benchmark's subdomains. Mapping 19 MCQA benchmarks across five model families on the HARMONY mean-variance plane reveals a spectrum of reliability. High mean and low variance indicate that aggregate metrics consistently reflect broad competence across models. In contrast, low mean signals that performance concentrates on a few subdomains and high variance indicates model-dependent reliability. Therefore, benchmarks with high mean and low variance of HARMONY enable more reliable evaluation.

Controlled pruning shows that increasing HARMONY stabilizes aggregate accuracy by reducing overrepresented subdomains. Moreover, we find that scaling trends of performance uniformity are family specific, rendering the number of parameters as an unreliable indicator for the uniformity of model performance. Nevertheless, models perform more uniformly on average as the pre-training budget increases. HARMONY complements aggregate accuracy by exposing when performance gains reflect uniform competence versus concentrated strengths and supports multi-dimensional evaluation that makes subdomain trade-offs explicit.

## ETHICS STATEMENT

This work evaluates publicly available MCQA benchmarks and introduces *RedundantQA*, a synthetic dataset generated from author-written seeds and LLM generations with no human subjects, personal data, or sensitive attributes collected. All examples were screened to avoid offensive content and verbatim copyrighted material. We respect the licenses of all benchmarks and models used. We will release our code and RedundantQA with documentation of construction, intended use, and limitations.

HARMONY is intended to complement (and *not replace*) standard accuracy and robustness analyses. Potential risks include misinterpretation of HARMONY or benchmark rebalancing to mask undesired failure modes. We therefore report both harmony and accuracy and encourage transparent, multi-metric evaluation. Our experiments rely on inference with open-sourced models. The authors declare no conflicts of interest or external sponsorship that could bias this work. We declare use of Large Language Models in this work in Appendix Q.

## REPRODUCIBILITY STATEMENT

We describe all experiments, datasets, models, and evaluation protocols openly and in detail in the main paper and appendix, including the construction of RedundantQA, the predictive similarity computation, partition induction, HARMONY definition and computation, pruning procedures, and statistical significance tests. We report model and benchmark versions, inference settings, and random seeds, and we specify hyperparameters and implementation choices (e.g., bandwidth selection, similarity scaling, and clustering criteria) to enable reproducibility. We also provide clear references to where each component is defined (Sections §2–§4 and Appendices A–L). We will open-source our codebase and findings, as well as RedundantQA, to facilitate exact replication.

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

## A  REDUNDANTQA

To rigorously evaluate the discriminative power of similarity metrics, we construct RedundantQA, a controlled benchmark designed to disentangle genuine semantic similarity from superficial lexical overlap. Each set in RedundantQA consists of a reference question accompanied by two *true-similar* and two *false-similar* questions. The true-similar questions are paraphrases that evaluate the same underlying knowledge as the reference, while differing in surface form.[8] In contrast, the false-similar questions exhibit high lexical similarity to the reference but target distinct conceptual content. This design ensures that strong similarity metrics must go beyond surface-level cues, rewarding semantic alignment while ignoring spurious correlations.

In this section, we detail the construction (A.1) and validation (A.2) of RedundantQA, as well as showcasing examples (A.3).

### A.1  CONSTRUCTION

We construct RedundantQA through a two-phase pipeline followed by strict validation: **(i) Seed Set Selection.** We begin by manually authoring three high-quality reference questions across four domains (Biology, Economics, Popular Culture, History). For each reference question, we also craft two paraphrases that target the same underlying knowledge (*true-similar*) and two distractors that share surface tokens but probe different concepts (*false-similar*). **(ii) Generative Expansion.** Using the seed sets as in-context learning examples, we prompt `Gemini-2.0-flash` (DeepMind, 2023) to generate 100 sets that consist of one reference question, two true-similar questions, and two false-similar questions for each domain. For different domains, we use a fixed template (Listing A.1) with domain-specific examples. This pipeline yields a large, automatically generated candidate pool.

### A.2  VALIDATION

We validate each set generated by `Gemini-2.0-flash` through a two-stage pipeline: (a) an automated and simple consistency check using `Gemini-2.0-flash` to confirm that true-similar paraphrases produce identical answers while false-similar distractors yield divergent ones (using Listing A.2); and (b) a manual review by expert annotators to correct any misclassifications, formatting issues, or errors introduced during automated filtering. After the validation step, we obtain 71, 39, 72, and 73 sets from Biology, Economics, Culture, and History domains respectively, with each set consisting of one reference question, two true-similar questions, and two false-similar questions.

This procedure yields a benchmark in which effective similarity metrics must discriminate semantic equivalence from mere lexical coincidence.

### A.3  EXAMPLES

We provide examples from RedundantQA across all four domains in Table 1.

---

[8]E.g., variations in vocabulary, syntax, or phrasing.

**Prompt for Generating RedundantQA (Biology)**

Come up with question sets. Each set must contain:

- A reference question,
- Two same-meaning questions: These should require the same factual answer and test the same biological concept as the reference question, but they should use different wording, phrasing styles, and sentence structures.
- Two distractor questions: These should look superficially very similar to the reference question but evaluate a different knowledge or skill with different answers than the reference question.

**Notes:**

- Focus on the domain of biological knowledge.
- Same-meaning questions should preserve deep semantic equivalence but vary stylistically. These must have the same answer.
- Distractor questions should maximize shallow textual similarity (e.g., shared nouns, verbs, syntactic patterns) while changing the underlying meaning. So, distractor questions should trick an incapable similarity measure into thinking they are similar.

**Examples:**
**Set 1**

- *Reference Question:* What organ pumps blood throughout the human body?
- *Same-meaning Question 1:* Which organ circulates blood to deliver oxygen and nutrients?
- *Same-meaning Question 2:* What body system structure maintains blood flow across the body?
- *Distracting Question 1:* What organ removes carbon dioxide from the blood?
- *Distracting Question 2:* What organ transports nutrients through the blood?

**Set 2**

- *Reference Question:* What process converts glucose into energy in cells?
- *Same-meaning Question 1:* Which process produces ATP from sugar molecules?
- *Same-meaning Question 2:* What pathway transforms glucose into usable cellular energy?
- *Distracting Question 1:* What process stores glucose in cells?
- *Distracting Question 2:* What process breaks down proteins for energy?

**Set 3**

- *Reference Question:* What type of blood vessel carries blood away from the heart?
- *Same-meaning Question 1:* Which vessels transport oxygenated blood from the heart?
- *Same-meaning Question 2:* What structures move blood outward from the heart?
- *Distracting Question 1:* What type of blood vessel brings blood to the heart?
- *Distracting Question 2:* What blood vessel type filters blood in the kidneys?

---

**Prompt for Validating RedundantQA**

Do the following questions have the same answer? Output only yes or no.
Question 1: REFERENCE_QUESTION
Question 2: TRUE_SIM_1

## B PREDICTIVE SIMILARITY

### B.1 ALTERNATIVE BASELINES

[Reviewer F1zp: To better situate our predictive similarity metric relative to existing work, we first position it against a diverse set of alternative similarity measures from the literature. These baselines

| Biology | Economics | History | Popular Culture |
|---|---|---|---|
| **Reference** | **Reference** | **Reference** | **Reference** |
| What process converts glucose into energy in cells? A: Cellular respiration B: Photosynthesis C: Osmosis D: Transcription | How does increased government spending affect aggregate demand? A: Increases it. B: Decreases it. C: Has no effect. D: Only affects aggregate supply. | Who was the first president of the United States? A: George Washington B: Abraham Lincoln C: Thomas Jefferson D: John Adams | Who played Iron Man in the Marvel Cinematic Universe? A: Robert Downey Jr. B: Chris Evans C: Hugh Jackman D: Tobey Maguire |
| **True Similar** | **True Similar** | **True Similar** | **True Similar** |
| Which process produces ATP from sugar molecules? A: Cellular respiration B: Photosynthesis C: Osmosis D: Transcription | What happens to total demand in economy when the government increase its spending? A: Increases it. B: Decreases it. C: Has no effect. D: Only affects aggregate supply. | Who assumed leadership as America's first head of state? A: George Washington B: Abraham Lincoln C: Thomas Jefferson D: John Adams | Which actor portrayed Tony Stark in the MCU? A: Robert Downey Jr. B: Chris Evans C: Hugh Jackman D: Tobey Maguire |
| **False Similar** | **False Similar** | **False Similar** | **False Similar** |
| What process stores glucose in cells? A: Glycogenolysis B: Gluconeogenesis C: Glycogenesis D: Glycolysis | How does increased government spending affect government debt? A: Increases it. B: Decreases it. C: Has no effect. D: Only affects short-term debt. | Who was the first vice president of the United States? A: John Adams B: Thomas Jefferson C: Alexander Hamilton D: James Madison | Who played Captain America in the Marvel Cinematic Universe? A: Chris Evans B: Chris Pratt C: John Krasinski D: Matt Damon |

Table 1: Example sets across all domains in RedundantQA.

span surface-form (n-gram), embedding-based, gradient-based, model-output-based similarities that are commonly used in prior work. In contrast, to the best of our knowledge, predictive similarity is the only metric that explicitly leverages the full probability distributions produced by the models under evaluation.]

In this section, we describe the alternative baselines compared against predictive similarity across a range of controlled settings.

**Bigram.** We compute an $n$-gram-overlap Jaccard similarity matrix. For each text $x_i$, we lowercase and split on whitespace, then form the set $G_i$ of contiguous bigrams. The pairwise similarity is $S_{ij} = \dfrac{|G_i \cap G_j|}{|G_i \cup G_j|}$ with $S_{ii} = 1$ for all $i$.[9] The resulting $S \in [0,1]^{N \times N}$ is symmetric and measures surface-form overlap.

**BERTScore F1.** We use BERTScore (Zhang et al., 2020) to measure semantic similarity between pairs of texts by comparing their contextualized token embeddings. Tokens are greedily matched via cosine similarity to compute precision and recall, and the final sentence-sentence score is the F1 aggregate, where $F_1 = \frac{2PR}{P+R}$. We treat this F1 value as the pairwise similarity, which yields a symmetric matrix $S \in [-1,1]^{N \times N}$. We employ $\texttt{Roberta}_{\texttt{Large}}$ (Liu et al., 2019) for obtaining the contextualized token embeddings.

**Input Embeddings Cosine Similarity.** We map each input to a single vector and measure pairwise similarity via cosine in embedding space. We use two variants: (i) for each example, we take the last-token hidden state from the model under evaluation, $\ell_2$-normalize it, and set $S_{ij} = \hat{h}_i^\top \hat{h}_j$, (ii) we encode each input with a frozen sentence-embedding model, normalize the embeddings, and compute the same cosine-based matrix. In both cases, we obtain a symmetric matrix $S \in [-1,1]^{N \times N}$. For the sentence-embedding variant, we use $\texttt{MiniLM}$[10] (Wang et al., 2020) and $\texttt{gte-Qwen2-7B-instruct}$[11] (Li et al., 2023); for the model-under-evaluation variant, we use $\texttt{microsoft/Phi-3-mini-4k-instruct}$[12] (Abdin et al., 2024), which yields the best performance in Appendix B.2.

---

[9]Note that if a text has fewer than $n$ tokens, its $n$-gram set is empty. In such a case, the pairwise similarity is set to be 1. However, this is not observed in practice.

[10]https://huggingface.co/sentence-transformers/all-MiniLM-L6-v2

[11]https://huggingface.co/Alibaba-NLP/gte-Qwen2-7B-instruct

[12]https://huggingface.co/microsoft/Phi-3-mini-4k-instruct

**Input and Output Embeddings Cosine Similarity.** We represent each input-output pair (e.g., question+answer) as a single vector by taking the last-token hidden state of the concatenated sequence from the model under evaluation. We $\ell_2$-normalize these vectors and define pairwise similarity via cosine with $S_{ij} = \hat{h}_i^\top \hat{h}_j$. The resulting $S \in [-1, 1]^{N \times N}$ is symmetric and reflects similarity over both the question and its associated answer. We use `microsoft/Phi-3-mini-4k-instruct` (Abdin et al., 2024) for obtaining the hidden states, as it yields the best performance in Appendix B.2.

**G-Vendi.** Following Jung et al. (2025), we quantify the diversity of per–example gradients via a sketch-based spectral entropy. For each example, we form a compact count-sketch of the gradient of the (negative) log-probability of the correct answer under a proxy LM, yielding a matrix $G \in \mathbb{R}^{N \times d}$. We compute $C = \frac{1}{N} G^\top G$ and its eigenvalues $\{\lambda_i\}$. Let $p_i = \lambda_i / \sum_j \lambda_j$; the *G-Vendi* score is the exponential Shannon entropy of this spectrum:

$$\text{G-Vendi} = \exp\Big( -\sum_i p_i \log p_i \Big),$$

which acts as an effective rank where higher values indicate gradients spread across more orthogonal directions and lower values indicate concentration in a low dimensional subspace. For pairwise similarity, we $\ell_1$-normalize the sketch rows of $G$ and take their dot products to obtain a symmetric similarity matrix with unit diagonal. Following the original implementation, we employ `Qwen2.5-0.5B-Instruct` (Qwen et al., 2025) as the proxy model.

**CORRS.** Following Vivek et al. (2024), given a bank of source models, we map each input $i$ to a vector $v_i \in \mathbb{R}^M$ whose $m$-th entry is the logit of the probability that model $m$ assigns to the correct choice. Then, the similarity between two examples is defined as the Pearson correlation of these vectors with $S_{ij} = \text{corr}(v_i, v_j)$. The resulting $S \in [-1, 1]^{N \times N}$ is symmetric and represents the cross-model agreement in correct class confidence across inputs. We instantiate the source bank using the Llama model family (Grattafiori & et al, 2024) and the full set of models used in our experiments.

| Method | Duplicate Catch Ratio ($\uparrow$) |
|---|---|
| *N-gram & Token* | |
| Bigram | 96.3 |
| BERTScore F1 | 100.0 |
| *Embedding-Based* | |
| Input Embeddings$_{\text{MiniLM}}$ | 100.0 |
| Input Embeddings$_{\text{gte-Qwen2}}$ | 100.0 |
| Input Embeddings$_{\text{phi3}}$ | 100.0 |
| Input+Output Embeddings$_{\text{phi3}}$ | 100.0 |
| *Literature* | |
| G-Vendi | 100.0 |
| CORRS$_{\text{Llama}}$ | 100.0 |
| CORRS$_{\text{all}}$ | 100.0 |
| IRT Representation | 100.0 |
| Predictive Similarity | 100.0 |

Table 2: **Validation of our metric implementations.** All metrics other than bigram similarity perfectly catch exact duplicate question, satisfying the minimum requirement to ensure implementation accuracy.

**IRT Representation.** Following Polo et al. (2024), from a bank of source models, we form a binary response matrix $Y \in \{0, 1\}^{L \times N}$ whose $(\ell, i)$ entry indicates whether model $\ell$ answered example $i$ correctly. We then fit a $d$-dimensional IRT model with per-example parameters $(\alpha_i \in \mathbb{R}^d, \beta_i \in \mathbb{R})$ and per-model ability vectors $\theta_\ell \in \mathbb{R}^d$, using

$$\Pr(Y_{\ell i} = 1) = \sigma\big( -\theta_\ell^\top \alpha_i + \beta_i \big).$$

Here, optimization alternates between gradient updates for $\theta$ (with $\ell_2$ regularization and recentring) and logistic regressions to update $(\alpha_i, \beta_i)$. Finally, we obtain the embedding $E_i = [\alpha_i; \beta_i] \in \mathbb{R}^{d+1}$ and define pairwise similarity by cosine similarity as $S_{ij} = \frac{E_i^\top E_j}{\|E_i\| \|E_j\|}$, which yields a symmetric matrix $S \in [-1, 1]^{N \times N}$. We use $d = 200$ and instantiate the source bank using the full set of models used in our experiments.

As open-source implementations of efficient evaluation metrics are unavailable, we re-implement them following the specifications in prior work. We then validate our implementations via a sanity check in which each metric was tasked with detecting verbatim duplicate questions. As shown in Table 2, all metrics (with the exception of bigram similarity) achieve perfect performance, satisfying the minimum requirement to ensure implementation accuracy.

## B.2 Measuring Semantic Similarity

An effective similarity measure for uncovering underlying data distributions must exhibit strong discriminative power, reliably identifying semantically similar data points while rejecting distractors. We empirically validate that predictive similarity meets this criterion, as it consistently distinguishes true semantic matches from misleading surface-level overlaps in RedundantQA.

| Method | True Similar ($\uparrow$) | | | | | False Similar ($\downarrow$) | | | | |
|---|---|---|---|---|---|---|---|---|---|---|
| | Biology | Economics | Culture | History | All | Biology | Economics | Culture | History | All |
| *N-gram & Token* | | | | | | | | | | |
| Bigram | 1.4 | 0.0 | 6.9 | 7.0 | 4.5 | 70.0 | 67.6 | 30.6 | 47.9 | 53.7 |
| BERTScore F1 | 8.6 | 0.0 | 12.5 | 21.1 | 12.4 | 74.3 | 83.8 | 33.3 | 54.9 | 60.3 |
| *Embedding-Based* | | | | | | | | | | |
| Input Embeddings$_{MiniLM}$ | 42.9 | 24.3 | 62.5 | 66.2 | 54.1 | 27.1 | 51.4 | 5.6 | 12.7 | 21.1 |
| Input Embeddings$_{gte-Qwen2}$ | 18.6 | 18.9 | 23.6 | 39.4 | 26.9 | 38.6 | 51.4 | 26.4 | 22.5 | 33.5 |
| Input Embeddings$_{phi3}$ | 21.4 | 5.4 | 16.7 | 36.6 | 22.7 | 47.1 | 75.7 | 37.5 | 31.0 | 45.5 |
| Input+Output Embeddings$_{phi3}$ | 51.4 | 62.2 | 40.3 | 56.3 | 52.9 | 22.9 | 27.0 | 16.7 | 15.5 | 20.2 |
| *Literature* | | | | | | | | | | |
| G-Vendi | 62.9 | 37.8 | 36.1 | 46.5 | 47.9 | 5.7 | 2.7 | 6.9 | 5.6 | 5.8 |
| CORRS$_{Llama}$ | 2.9 | 8.1 | 9.7 | 14.1 | 9.1 | 1.4 | 0.0 | 1.4 | 1.4 | 1.2 |
| CORRS$_{all}$ | 35.7 | 13.5 | 59.7 | 59.2 | 47.5 | 1.4 | 5.4 | 0.0 | 1.4 | 1.7 |
| IRT Representation | 1.4 | 0.0 | 1.4 | 0.0 | 0.8 | 1.4 | 0.0 | 0.0 | 2.8 | 1.2 |
| Predictive Similarity | **80.0** | **86.5** | **66.6** | **73.2** | **77.7** | 1.4 | 2.7 | 0.0 | 4.2 | 2.1 |

Table 3: Proportion of identified true-similar ($\uparrow$) and false-similar ($\downarrow$) pairs by method and domain.

As shown in Table 3, predictive similarity achieves the highest retrieval of true semantic matches across all domains, while maintaining one of the lowest rate of false matches. This indicates that it captures semantic equivalence without being misled by superficial lexical similarity. In contrast, embedding-based and bigram baselines suffer from high false positives, conflating surface-level resemblance with meaning. Metrics from efficient evaluation literature show stronger performance but still fall short of predictive similarity. Overall, these results highlight the unique discriminative advantage of predictive similarity in measuring semantic similarity.

## B.3 Inducing the Semantic Partition

A core requirement for our work is that the similarity function should induce a semantic partition of the data. We evaluate this property by inducing cluster assignments from each metric and measuring agreement with the ground-truth domain labels in RedundantQA and its reference-only subset (RedundantQA-Ref) using Adjusted Rand Index (ARI) and Normalized Mutual Information (NMI). In addition, we apply the same protocol to MMLU high school subtasks, testing whether clusters recover canonical subject domains (e.g., computer science, biology, physics). [Reviewer 5RAY: Finally, we validate the top-performing similarity baselines from MMLU and RedundantQA on GPQA and M3Exam (Zhang et al., 2023).]

| Method | RedundantQA | | RedundantQA-Ref | | MMLU-HS | |
|---|---|---|---|---|---|---|
| | ARI | NMI | ARI | NMI | ARI | NMI |
| *N-gram & Token* | | | | | | |
| Bigram | -0.3 | 2.5 | -0.4 | 6.7 | 0.1 | 6.2 |
| BERTScore F1 | -0.2 | 1.6 | 2.2 | 8.6 | 1.4 | 7.9 |
| *Embedding-Based* | | | | | | |
| Input Embeddings$_{MiniLM}$ | 55.8 | 64.4 | 59.4 | 68.4 | 47.4 | 56.8 |
| Input Embeddings$_{gte-qwen2-7b-instruct}$ | 28.6 | 34.9 | 36.3 | 45.9 | 27.1 | 34.4 |
| Input Embeddings$_{phi3}$ | 27.8 | 37.1 | 33.0 | 45.6 | 25.1 | 31.6 |
| Input+Output Embeddings$_{phi3}$ | 57.1 | 67.0 | 58.9 | 70.3 | 51.3 | 59.6 |
| *Literature* | | | | | | |
| G-Vendi | 6.5 | 11.3 | 4.7 | 10.9 | 3.2 | 9.2 |
| CORRS$_{all}$ | 6.1 | 8.2 | 4.8 | 8.2 | 2.8 | 8.4 |
| IRT Representation | 2.1 | 2.7 | 0.4 | 1.9 | 0.4 | 1.5 |
| Predictive Similarity$_{Qwen3\ 1.7B}$ | 60.4 | 70.4 | 62.5 | 76.5 | 55.4 | 62.1 |

Table 4: Validation of our partition induction method on RedundantQA and MMLU. Adjusted Rand Index (ARI) and Normalized Mutual Information (NMI) (both are higher is better) are shown for different methods on **RedundantQA**, **RedundantQA-Ref**, and **MMLU-HS**. Predictive similarity consistently achieves the best domain recovery.

As shown in Table 4 and 5, predictive similarity achieves the highest agreement on all three sets, while strong embedding-based baselines are competitive yet consistently behind. By contrast, token/ngram measures and metrics from the efficient evaluation literature fail to recover domain structure, indicating that they are unreliable for semantic grouping. Taken together with the pairwise retrieval evidence, these results show that predictive similarity not only discriminates true semantic matches from distractors, but also organizes instances into compact and consistent clusters. This be-

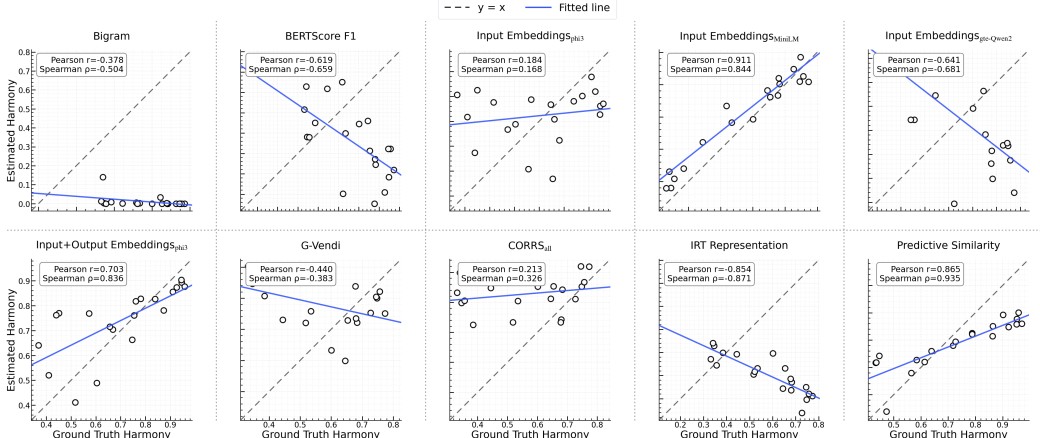

Figure 7: **Validation of our partition induction method on RedundantQA**. Predictive similarity achieves the strongest correlation between the ground truth HARMONY and estimated HARMONY, while input embeddings derived from `MiniLM` is a close runner-up.

havior is precisely what enables a cluster-centric analysis of benchmarks, yielding low within-cluster variance and high between-cluster separation.

### B.4 CAPTURING BENCHMARK HARMONY

We evaluate all alternative similarity baselines from Appendix B.1 in the identical setting of §2.4. For each baseline, we induce clusters from its similarity matrix, compute HARMONY $H(\mathcal{G})$, and report correlations between the ground-truth HARMONY and its counterpart computed from the partition induced by each baseline. [Reviewer 5RAY: We additionally validate the top-performing similarity baselines from MMLU and RedundantQA on GPQA and M3Exam.]

| Method | M3Exam | | GPQA | |
|---|---|---|---|---|
| | ARI | NMI | ARI | NMI |
| Input Embeddings$_{\text{MiniLM}}$ | 60.8 | 65.5 | 58.4 | 64.3 |
| Input+Output Embeddings$_{\text{Phi3}}$ | 61.7 | 68.1 | 59.3 | 66.1 |
| Predictive Similarity$_{\text{Qwen3 1.7B}}$ | **64.7** | **71.1** | **63.1** | **69.9** |

Table 5: Further validation of our partition induction method on **GPQA** and **M3Exam**. Predictive similarity achieves the best domain recovery on both benchmarks.

As shown in Fig. 7, 8 and 9, predictive similarity achieves the strongest correlation with ground-truth entropy. Similar to prior validation experiments (App. B.2, B.3), embedding-based baselines are the next-best performers but consistently lag behind, whereas token-and-$n$-gram overlap measures perform substantially worse. These results establish predictive similarity as the most reliable similarity metric choice for capturing the benchmark dynamics.

[Reviewer 85MX: Lastly, Figure 10 pools all validation points from RedundantQA, MMLU, GPQA, and M3Exam, complementing the per-benchmark results in Figure 7, 8, and 9, where predictive similarity consistently achieves the strongest alignment with ground-truth Harmony across imbalance variants. When we combine these experiments into a single pool, the relationship remains strongly positive (Pearson $r = 0.869$ and Spearman $\rho = 0.879$), showing that our estimator preserves Harmony ordering not only within each benchmark family but also across benchmarks. This supports our claim that estimated Harmony provides a robust, benchmark-agnostic reliability signal.]

### B.5 CONSISTENCY OF PREDICTIVE SIMILARITY ACROSS MODELS

As defined in §2.3, predictive similarity induces, for each benchmark $\mathcal{B}$ and model $f$, a symmetric similarity matrix $S^{(f,\mathcal{B})} \in (0,1]^{N_\mathcal{B} \times N_\mathcal{B}}$ over the $N_\mathcal{B}$ items of $\mathcal{B}$. In this section, we ask whether these model-specific neighborhoods are idiosyncratic. To quantify cross-model consistency, we correlate the upper-triangular entries of the corresponding similarity matrices using both rank-based

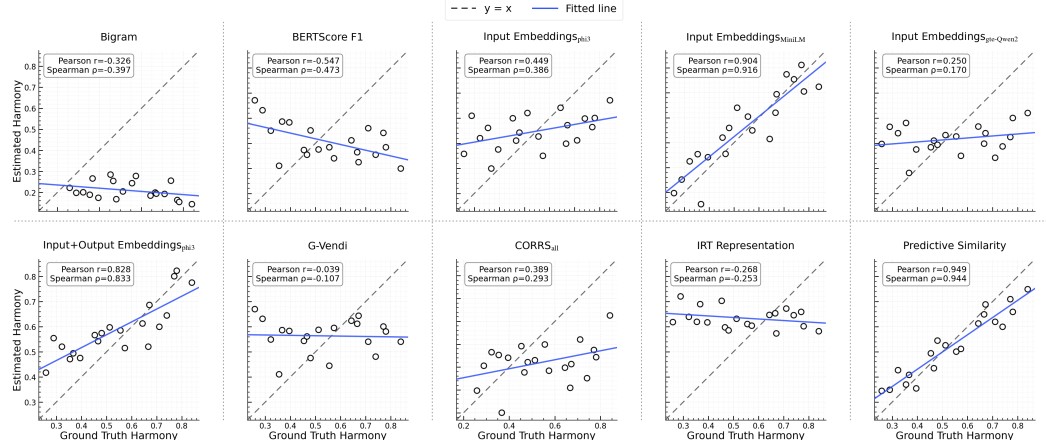

Figure 8: **Validation of our partition induction method on MMLU**. Similar to the results on RedundantQA, predictive similarity achieves the strongest correlation between the ground truth HARMONY and estimated HARMONY, while input embeddings derived from MiniLM is a close runner-up.

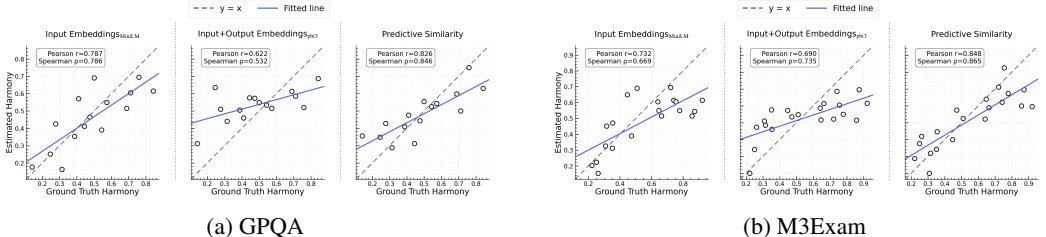

(a) GPQA          (b) M3Exam

Figure 9: **Validation of our partition induction method on GPQA and M3Exam**. Similar to the results on RedundantQA and MMLU, predictive similarity achieves the strongest correlation between the ground truth HARMONY and estimated HARMONY.

(Spearman) and linear (Pearson) correlations:

$$r_S^{\mathcal{B}}(f_1, f_2) = \rho_S\big(\text{vec}(S^{(f_1,\mathcal{B})}),\ \text{vec}(S^{(f_2,\mathcal{B})})\big), \qquad r_P^{\mathcal{B}}(f_1, f_2) = \rho_P\big(\text{vec}(S^{(f_1,\mathcal{B})}),\ \text{vec}(S^{(f_2,\mathcal{B})})\big),$$

where $\text{vec}(\cdot)$ stacks the upper triangle of a matrix into a vector, $\rho_S$ is Spearman's rank correlation, and $\rho_P$ is Pearson's correlation. We report both, as Spearman is invariant to monotone re-scalings and thus robust to calibration differences across models, while Pearson captures linear alignment in similarity magnitudes.

As shown in Table 6, predictive similarity neighborhoods exhibit substantial within-family consistency across many benchmarks. Averaging across families yields high per-benchmark means clustered in the mid-60s, with notable peaks for *LogiQA* (70.3 Spearman / 71.3 Pearson), *TruthfulQA* (72.5 / 74.5), *PIQA* (67.2 / 68.8), and *ART* (68.1 / 69.1). Family-wise averages further show broad stability for Llama (66.7 / 67.7), Qwen (62.9 / 63.7), and Phi (61.2 / 62.1), with OLMo close behind (60.3 / 61.3) and Gemma lower (37.9 / 38.7). In particular, *ART*, *COPA*, *LogiQA*, *PIQA*, and *TruthfulQA* attain high agreement for Llama, Phi, and Qwen (typically 70–85), indicating that the induced item-item structure is largely task-driven rather than model idiosyncratic. Knowledge-centric *GPQA* is also strong for Phi and Qwen (82–86). By contrast, *StrategyQA*, *BoolQ*, and, to a lesser extent, *SocialQA* show weaker agreement particularly for Gemma and Phi, suggesting greater family-specific effects on these benchmarks.

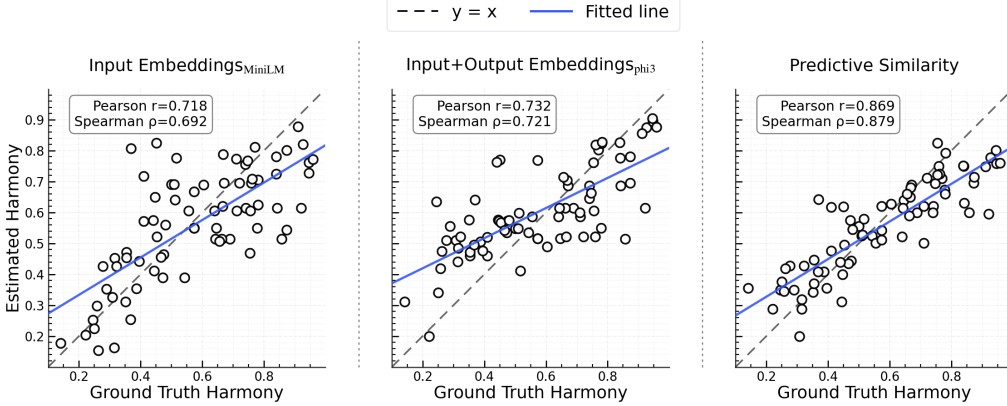

Figure 10: **Validation of our partition induction method on combination of RedundantQA, MMLU, GPQA, and M3Exam**. Similar to the individual results, predictive similarity achieves the strongest correlation between the ground truth HARMONY and estimated HARMONY.

| | Gemma | | Llama | | OLMo | | Phi | | Qwen | | Average | |
|---|---|---|---|---|---|---|---|---|---|---|---|---|
| **Benchmark** | **Spearman** | **Pearson** | **Spearman** | **Pearson** | **Spearman** | **Pearson** | **Spearman** | **Pearson** | **Spearman** | **Pearson** | **Spearman** | **Pearson** |
| AQUA-RAT | 61.8 | 62.7 | 63.3 | 65.9 | 47.0 | 49.3 | 49.8 | 50.9 | 76.1 | 76.5 | 59.6 | 61.1 |
| ARC-Challenge | 41.9 | 42.7 | 76.3 | 76.9 | 72.9 | 73.9 | 71.9 | 73.9 | 74.5 | 76.1 | 67.5 | 68.7 |
| ARC-Easy | 41.6 | 42.0 | 66.6 | 67.1 | 67.9 | 68.7 | 75.3 | 76.3 | 62.7 | 64.0 | 62.8 | 63.6 |
| ART | 25.5 | 26.4 | 84.1 | 84.6 | 76.6 | 77.9 | 78.6 | 79.8 | 75.7 | 76.8 | 68.1 | 69.1 |
| BoolQ | 25.5 | 25.9 | 58.3 | 59.8 | 35.3 | 36.3 | 18.2 | 18.9 | 27.5 | 28.9 | 33.0 | 34.0 |
| CommonsenseQA | 33.8 | 35.6 | 33.6 | 34.4 | 62.8 | 64.5 | 42.4 | 43.8 | 39.1 | 39.8 | 42.3 | 43.6 |
| COPA | 27.9 | 29.5 | 76.7 | 78.4 | 68.7 | 70.2 | 64.8 | 67.2 | 51.4 | 52.8 | 57.9 | 59.6 |
| GPQA | 77.8 | 78.0 | 53.4 | 51.9 | 64.1 | 65.2 | 82.1 | 83.0 | 85.5 | 86.1 | 72.6 | 72.8 |
| LogiQA | 55.4 | 57.3 | 77.7 | 79.8 | 67.8 | 67.2 | 70.3 | 72.1 | 80.1 | 80.3 | 70.3 | 71.3 |
| MathQA | 33.5 | 34.2 | 61.0 | 62.0 | 55.2 | 56.0 | 54.9 | 54.0 | 57.1 | 56.4 | 52.3 | 52.5 |
| OpenBookQA | 33.2 | 34.0 | 73.9 | 75.0 | 70.1 | 72.3 | 74.0 | 75.5 | 71.0 | 72.5 | 64.4 | 65.9 |
| PIQA | 38.6 | 39.6 | 78.6 | 80.2 | 77.3 | 79.4 | 67.0 | 68.2 | 74.6 | 76.4 | 67.2 | 68.8 |
| PubMedQA | 50.5 | 50.0 | 60.8 | 60.8 | 37.8 | 36.8 | 50.6 | 50.6 | 63.0 | 61.1 | 52.5 | 51.9 |
| QUARTZ | 37.9 | 39.0 | 81.4 | 80.3 | 65.9 | 63.9 | 76.0 | 74.9 | 70.0 | 68.1 | 66.2 | 65.2 |
| SciQ | 22.0 | 22.5 | 55.7 | 58.3 | 60.3 | 61.4 | 56.0 | 56.0 | 58.0 | 60.3 | 50.4 | 51.7 |
| SocialIQA | 20.7 | 21.3 | 72.9 | 73.5 | 64.6 | 65.6 | 63.5 | 64.9 | 61.2 | 62.0 | 56.6 | 57.5 |
| StrategyQA | 3.6 | 3.7 | 48.2 | 49.6 | 20.5 | 20.9 | 21.6 | 21.4 | 27.9 | 28.9 | 24.4 | 24.9 |
| TruthfulQA | 50.4 | 52.7 | 78.4 | 80.7 | 71.4 | 73.6 | 85.3 | 86.4 | 77.1 | 79.3 | 72.5 | 74.5 |
| **Average** | 37.9 | 38.7 | 66.7 | 67.7 | 60.3 | 61.3 | 61.2 | 62.1 | 62.9 | 63.7 | 57.8 | 58.7 |

Table 6: **Cross-model consistency of predictive similarity**: Spearman and Pearson correlations (values $\times 100$) between the upper-triangle entries of affinity matrices, by benchmark (rows) and model family (columns). The rightmost block reports per-benchmark averages across families; the bottom row reports per-family averages across benchmarks; the bottom-right cell shows overall means.

### B.6  PROBING THE DETERMINANTS OF PREDICTIVE SIMILARITY

**Dependence on the Tail.** We test sensitivity to the probability tail by constructing a truncated variant that re-normalizes mass over the union of the `top-50` tokens, yielding $S_{\text{KL-top-50}}^{(f,\mathcal{B})}$, and contrasting it with the full $S_{\text{KL}}^{(f,\mathcal{B})}$.

**JS vs. KL-based Similarity.** Equation 2 defines $S$ as an RBF of the Jeffreys divergence, producing *sharp, tunable* neighborhoods that strongly penalize coverage errors (as near-zeros drive $J$ higher and $S$ lower). Jensen-Shannon (JS) instead compares to the mixture $M = \frac{1}{2}(\bar{p}_f(x_i) + \bar{p}_f(x_j))$, yielding a bounded, tail-robust divergence that is easier to compare across benchmarks. To place JS on the same similarity scale, we apply the same RBF transform from Equation 2 entrywise, obtaining $S_{\text{JS}}^{(f,\mathcal{B})}$.

Following Appendix B.5, we compute Spearman and Pearson correlations between the upper-triangular entries of $S_{\text{KL}}^{(f,\mathcal{B})}$ and, respectively, $S_{\text{KL-top-50}}^{(f,\mathcal{B})}$ and $S_{\text{JS}}^{(f,\mathcal{B})}$. We report per-

| | $S_{\text{JS}}$ | | $S_{\text{KL-top-50}}$ | |
|---|---|---|---|---|
| **Benchmark** | Pearson | Spearman | Pearson | Spearman |
| AQUA-RAT | 96.1 | 95.5 | 97.4 | 97.4 |
| ARC-Challenge | 95.7 | 95.2 | 98.5 | 98.5 |
| ARC-Easy | 95.4 | 95.0 | 98.7 | 98.7 |
| ART | 95.5 | 95.4 | 97.1 | 96.9 |
| BoolQ | 93.5 | 92.5 | 98.4 | 98.4 |
| CommonsenseQA | 97.0 | 96.7 | 99.5 | 99.5 |
| COPA | 94.7 | 95.2 | 95.1 | 94.9 |
| GPQA | 97.1 | 97.1 | 99.1 | 99.0 |
| LogiQA | 98.7 | 98.5 | 98.5 | 98.5 |

benchmark scores where $f$ is OLMo 2 7B. *High agreement* indicates that neighborhoods are driven by high probability mass and remain stable under truncation or mixture smoothing, while *low agreement* indicates sensitivity to tail mismatches or calibration asymmetries that Jeffreys magnifies but JS attenuates.

Across benchmarks, correlations between $S_{\text{KL}}^{(f,\mathcal{B})}$ and (i) $S_{\text{KL-top-50}}^{(f,\mathcal{B})}$ and (ii) $S_{\text{JS}}^{(f,\mathcal{B})}$ variants are uniformly high (typically 95-99%) (Table 7). This indicates that the divergence of probability distributions generated by OLMo 2 7B are governed by head probability mass rather than the tail. Truncation preserves structure nearly perfectly as $S_{\text{KL-top-50}}$ matches or exceeds $S_{\text{JS}}$ on most tasks, while $S_{\text{JS}}$ remains strongly aligned, reflecting robustness to calibration and coverage noise. Modest dips (e.g., COPA, PIQA) suggest settings where tail mismatches or asymmetries matter more, but overall the stability under truncation and mixture smoothing supports that $S$ captures meaningful, head-driven divergence.

### B.7 THEORETICAL AND COMPUTATIONAL DISCUSSIONS

**A Theoretical Perspective on Predictive Similarity.** We measure similarity via the (symmetrized) $D_{\text{KL}}$ between model predictive distributions because it aligns with how models differ operationally and geometrically. First, $D_{\text{KL}}$ has a clear testing meaning, as it governs optimal error exponents in distinguishing two distributions. Hence, larger $D_{\text{KL}}$ divergence implies that the model would more reliably tell the two inputs apart (by Stein/Chernoff asymptotics) (Cover & Thomas, 2006). Second, small $D_{\text{KL}}$ guarantees closeness in total variation by Pinsker's inequality, implying high indistinguishability and hence high similarity for our purposes (Cover & Thomas, 2006). Third, $D_{\text{KL}}$ is information monotone under coarse-graining, making the measure stable to relabeling or merging answer tokens/options that preserve semantics (Csiszár & Shields, 2004). Finally, locally $D_{\text{KL}}$ induces the Fisher-Rao geometry on the probability simplex, so $\exp(-\tau\,D_{\text{KL}})$ behaves like a Gaussian kernel in the natural metric of the model's predictive space, yielding compact clusters of similar predictive behavior (Amari, 2016). We use the Jeffreys (symmetrized) form to remove directionality while retaining these properties.

**Computational Overhead of Predictive Similarity.** Predictive similarity is a *post hoc* computation, since we operate on the logits already cached from the benchmark evaluation. Hence, no additional model forward passes are required. Given these logits, we convert them to predictive distributions and evaluate the pairwise KL terms that define the similarity in Eq. 2.

The principal cost arises from forming pairwise interactions across $N$ items, which is quadratic in $N$ and linear in the label-space size $D$ (i.e., $O(N^2 \cdot D)$ time). Memory is dominated by storing the evaluation logits ($O(N \cdot D)$) and the similarity matrix ($O(N^2)$). In practice, $D$ corresponds to the size of the vocabulary of a given model and can be large. We therefore view the cost as $O(N^2 D)$ and the memory requirement as $O(N \cdot D)$. When $N$ is large, standard remedies (e.g. blockwise evaluation) reduce peak memory without changing the definition of the metric. Overall, computing predictive simlarity adds negligible *inference* overhead and modest *analysis* overhead relative to running the benchmarks themselves.

Compared to alternative baselines discussed in Appendix B.1, predictive similarity is computationally frugal: it reuses cached logits and requires neither additional inference nor any backward passes. By contrast, embedding baselines, as well as BERTScore, invoke separate encoders (extra forward passes), G-Vendi relies on gradients (backward passes), and CORRS/IRT aggregate signals from a bank of models (multiple evaluations per item). While string-based methods such as Bigram are

lightweight, they do not leverage model behavior. Thus, predictive similarity offers a favorable trade-off between compute and quality when benchmarks are already being run.

### B.8 Discussion on Model-Specific Similarity

Our goal is to *evaluate benchmarks*, not to define a single, task-agnostic neighborhood data points. A benchmark can be *reliable for one model but unreliable for another*. Accordingly, the similarity function used to induce the partition $\mathcal{G}_f$ should be *conditional on the model* $f$. We provide our rationale below.

**Evaluation target is** $H_{\mathcal{B}}(f)$**.** Section 2 defines harmony *per model*, $H_{\mathcal{B}}(f)$, and then aggregates across $f$ via $(\mu_H, \sigma_H^2)$ in Eq. 1. Using a *global, model-agnostic* similarity collapses distinct predictive neighborhoods into a single partition, implicitly assuming that $\mathcal{G}_f$ is invariant across $f$. This undermines the very statistic we report: two models with the same accuracy profile but different predictive structure could receive the same $H$ under a fixed partition, obscuring model specialities.

**Benchmarks are instruments relative to a model.** A benchmark is a diagnostic instrument for a *given* model: priors, tokenization, calibration, and pre-training exposure all change which items are *similar* from the model's perspective. Hence, a model can perform uniformly on a benchmark while another one overfits to certain subdomains. Model specific similarity preserves this relativity, letting reliability vary meaningfully across families.

[Reviewer F1zp: [Reviewer 85MX:

## C Discussion on Design Choices for Harmony

We briefly motivate our definition of Harmony and our choice of normalized Shannon entropy. The primary reason is pragmatic: Shannon entropy is the canonical measure of uncertainty (and thus uniformity) in information theory, so its behavior and scale are well understood and easy to interpret.

Recall that we first compute non-negative weights $\pi_i = w_i K_i$ over subdomains $k = 1, \ldots, K$ that increase with (i) the mass of the dataset assigned to subdomain $i$ and (ii) how close the model's performance on $i$ is to the benchmark mean. Let

$$p_i \;=\; \frac{\pi_i}{\sum_{j=1}^K \pi_j}, \qquad \sum_{i=1}^K p_i = 1,$$

denote the normalized weights as in Section 2.2, so that $p = (p_1, \ldots, p_K)$ yields a probability vector that summarizes *where* in the benchmark the model is performing well.

Our design is guided by three criteria:

(i) **Joint coverage and uniformity:** Harmony should increase when more dataset mass lies on subdomains whose performance is close to the benchmark mean, and decrease when performance is concentrated on a few subdomains.

(ii) **Comparability across benchmarks:** Harmony should lie in $[0, 1]$ so benchmarks with different numbers of subdomains are directly comparable.

(iii) **Interpretability as reliability:** High Harmony should indicate that the benchmark behaves *as if* many subdomains are near the mean, while low Harmony signals effective concentration on a smaller subset.

We therefore define Harmony as the normalized Shannon entropy:

$$H(\mathcal{G}_f) \;=\; -\frac{1}{\log k} \sum_{i=1}^k p_i \log(p_i + \varepsilon) \in [0, 1],$$

This choice yields:

- **Boundedness:** $0 \leq H(\mathcal{G}_f) \leq 1$, with 1 attained only when $p_i = 1/k$ for all $i$, and values approaching 0 as mass concentrates on a single subdomain.

- **Effective number of well-performing subdomains:** Let $\tilde{H}(p) = -\sum_{i=1}^{k} p_i \log p_i$ denote the (unnormalized) Shannon entropy. Its exponential $\exp(\tilde{H}(p))$ is often interpreted as the *effective number* of equally weighted categories. Since $H(\mathcal{G}_f) = \tilde{H}(p)/\log k$, this can be read as a normalized effective number, i.e., the effective fraction of subdomains whose performance is close to the benchmark mean.

- **Sensitivity to non-uniformity:** Shannon entropy is strictly concave and Schur-concave, which makes any move from a more uniform to a more concentrated distribution $p = (p_1, \ldots, p_k)$ strictly decrease $H(\mathcal{G}_f)$.

## C.1 ALTERNATIVE UNIFORMITY MEASURES

**Variance and coefficient of variation.** A natural idea is to measure dispersion of subdomain performances $\{\Psi(f; A_i)\}_{i=1}^{k}$ via a (weighted) variance

$$\mathrm{Var}\big(\Psi(f; \cdot)\big) \;=\; \sum_{i=1}^{k} w_i \Big(\Psi(f; A_i) - \bar{\Psi}\Big)^2, \qquad \bar{\Psi} \;=\; \sum_{i=1}^{k} w_i \Psi(f; A_i).$$

However, variance is scale-dependent and not naturally normalized to $[0, 1]$, and it does not admit an *effective number of subdomains* interpretation. The coefficient of variation partly addresses scale but becomes unstable when the mean is close to 0 or 1, which is common for strong models. In contrast, Harmony depends only on the probability distribution $p = (p_1, \ldots, p_k)$ over subdomains and is automatically normalized.

**Gini coefficient and related indices.** Inequality indices such as the Gini coefficient measure concentration but are typically defined directly on the performance values $\{\Psi(f; A_i)\}_{i=1}^{k}$, not on a distribution that already combines dataset mass and closeness-to-mean. This makes it harder to interpret them as describing how *dataset mass* is distributed over near-mean vs. outlier subdomains, and they lack the effective-number interpretation that entropy has.

**Rényi entropies.** Rényi entropies form a one-parameter family

$$H_\alpha(p) \;=\; \frac{1}{1-\alpha} \log\Big(\sum_{i=1}^{k} p_i^\alpha\Big), \quad \alpha > 0, \alpha \neq 1,$$

with Shannon entropy recovered as the limit $\alpha \to 1$. Different orders $\alpha$ emphasize different parts of the distribution ($\alpha > 1$ focuses on high-probability subdomains and $\alpha < 1$ emphasizes low-probability ones), effectively introducing a hyperparameter that controls how aggressively Harmony responds to tails and modes. In this sense, choosing a particular Rényi order $\alpha$ is analogous to choosing a norm order (e.g., $\ell_\alpha$) when measuring distances. Therefore, Rényi entropy yields a whole family of alternatives, but also an extra design choice about which order to prefer. We deliberately avoid this additional degree of freedom. Using the Shannon case $\alpha = 1$ gives a single, parameter-free and widely interpretable notion of entropy that treats subdomains according to their weights $p_i$, without further reweighting.

## C.2 ROBUSTNESS OF HARMONY

Harmony is designed to penalize benchmarks that leave small, poorly performing regions unaddressed. Ignoring the smoothing term $\varepsilon$ for clarity, the underlying (unnormalized) Shannon entropy

$$\tilde{H}(p) = -\sum_{i=1}^{k} p_i \log p_i$$

can be viewed as the average information content under $p$. Each subdomain contributes $p_i \log(1/p_i)$, and $\log(1/p_i)$ grows as $p_i$ becomes small. As a result, shifting even modest probability mass away

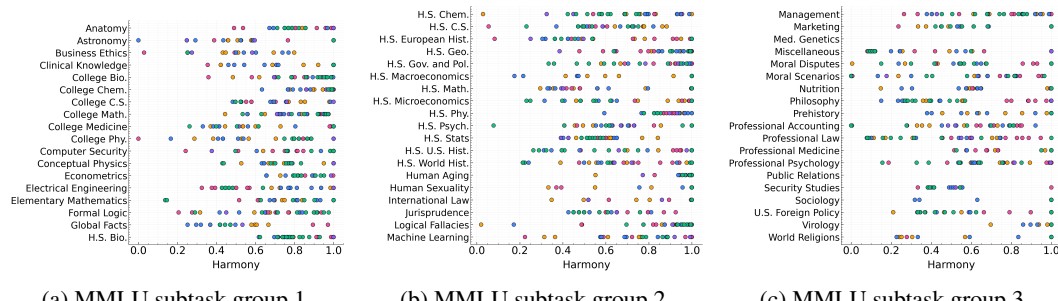

(a) MMLU subtask group 1.     (b) MMLU subtask group 2.     (c) MMLU subtask group 3.

Figure 12: Model-wise decomposition of HARMONY for MMLU subtasks.

from under-represented, well-performing subdomains (or toward poorly performing ones) leads to a noticeable drop in $H(\mathcal{G}_f)$. In this sense, Harmony is particularly sensitive to *gaps* in coverage, whereas variance-based dispersion measures tend to be dominated by large deviations in high-mass components and can largely ignore small but problematic regions.

Our formulation is, like any scalar index, not completely immune to adversarial benchmark design (e.g., redefining the subdomain partition). However, for a fixed partition and mean performance, increasing $H(\mathcal{G}_f)$ requires genuinely redistributing dataset mass so that a larger share lies in subdomains whose performance is close to the mean, while simple rescalings of scores or adding redundant, near-duplicate subdomains do not help. This makes Harmony behave as an interpretable reliability index with higher values corresponding to benchmarks where performance is spread broadly across the subdomains, rather than confined to certain subdomains. ]]

# D    MODEL LIST

We list all evaluated models and provide links to their open-source weights.

- Qwen3: Qwen3-0.6B-Base, Qwen3-1.7B-Base, Qwen3-4B-Base, Qwen3-8B-Base, Qwen3-14B-Base, Qwen3-0.6B, Qwen3-1.7B, Qwen3-4B, Qwen3-8B, Qwen3-14B.

- Llama 3: Llama-3.2-1B, Llama-3.2-3B, Llama-3.1-8B, Llama-3.1-70B, Llama-3.2-1B-Instruct, Llama-3.2-3B-Instruct, Llama-3.1-8B-Instruct, Llama-3.1-70B-Instruct.

- Olmo 2: OLMo-2-0425-1B, OLMo-2-1124-7B, OLMo-2-1124-13B, OLMo-2-0325-32B, OLMo-2-0425-1B-Instruct, OLMo-2-1124-7B-Instruct, OLMo-2-1124-13B-Instruct, OLMo-2-0325-32B-Instruct.

- Gemma 3: gemma-3-1b-pt, gemma-3-4b-pt, gemma-3-12b-pt, gemma-3-27b-pt, gemma-3-1b-it, gemma-3-4b-it, gemma-3-12b-it, gemma-3-27b-it.

- Phi-3: Phi-3-mini-4k-instruct, Phi-3-medium-4k-instruct.

# E    MODEL-WISE DECOMPOSITION OF BENCHMARK HARMONY

In §3.2, we position each benchmark $\mathcal{B}$ using the cross-model mean $\mu_H(\mathcal{B})$ and variance $\sigma_H^2(\mathcal{B})$. We now resolve this view at the model level. For each benchmark, Fig. 11 plots the per-model vector $\{H_{\mathcal{B}}(f)\}_{f \in \mathcal{F}}$, revealing structure that is obscured by aggregation. Similarly, Fig. 12 provides the analogous decomposition for MMLU subtasks, treating each subtask as a benchmark on its own.

Tight *horizontal* groupings (small spread across models) indicate *model-invariant* distributional balance, where different families assign similar HARMONY to the same benchmark, suggesting that ag-

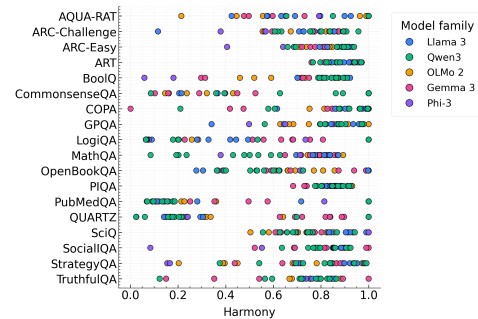

Figure 11: Model-wise decomposition of HARMONY for MCQA benchmarks.

gregate accuracy reflects uniform competence irre-
spective of architectural or training choices. Con-
versely, wide horizontal scatter exposes *model-dependent reliability*, as some families concentrate
performance on a few subsets (low HARMONY), while others distribute performance more evenly
(high HARMONY).

We note that benchmarks with tight clusters are favorable for cross-family comparison, as accuracy
rankings are less likely to be artifacts of benchmark composition. In contrast, wide scatter warns
that leaderboard deltas may be driven by subsets that particular families exploit. In such cases,
we suggest reporting accuracy alongside the HARMONY profiles of the models under evaluation,
$\{H(\mathcal{G}_f)\}_f$.

## F    IMPROVEMENT HARMONY

In §4.2, we show that scaling behavior varies across model families as parameter count increases:
some families (e.g., Qwen3) exhibit increasing HARMONY, while others (e.g., Gemma 3) show the
opposite. We now ask whether *performance improvements* from scaling are distributed evenly across
subsets. For two adjacent model sizes within a family, let the per-subset change be

$$d_i \;=\; \Psi(f_{\text{large}}; A_i) \;-\; \Psi(f_{\text{small}}; A_i),$$

with subset weights $w_i$ and partition $\mathcal{G} = \{A_i\}_{i=1}^k$ defined as in §2.1. Let $\bar{d} = \sum_i w_i d_i$ be
the weighted mean and reuse the HARMONY computation by replacing accuracies $\Psi(f; A_i)$ with
changes $d_i$:

$$K_i \;=\; \exp\!\Big(-\big(\tfrac{d_i-\bar{d}}{b}\big)^2\Big), \qquad p_i \;=\; \frac{w_i K_i}{\sum_j w_j K_j}, \qquad H_\Delta(\mathcal{G}) \;=\; -\frac{1}{\log k}\sum_{i=1}^k p_i \log\big(p_i + \varepsilon\big).$$

High $H_\Delta$ indicates that scaling yields *uniform* changes across subsets, while low $H_\Delta$ indicates *spiky*
changes concentrated in a few clusters. Similar to § 2, we adopt a comparative perspective, asking
which models improve more uniformly and which benchmarks most facilitate uniform gains. To
ensure within-family comparability, we fix the partition to that induced by the smallest model in
each family and evaluate all larger models on these partitions.

**Improvement HARMONY of Benchmarks (Fig. 13).**    Due to the lack of a principled baseline for
$H_\Delta$, we interpret results comparatively rather than absolutely. Benchmarks vary in improvement
HARMONY, and those with higher performance HARMONY tend to exhibit higher $H_\Delta$. Using all
benchmarks in our setup, the fitted-line correlation is $r = 0.226$, which increases to $r = 0.387$
after excluding the three lowest-HARMONY benchmarks. Thus, higher HARMONY benchmarks are
associated with more uniform improvements in a comparative sense, though the effect size is modest
and sensitive to less harmonious outliers.

**Improvement HARMONY of Models (Fig. 14).**    We measure improvement HARMONY $H_\Delta$ for
adjacent sizes within each family. For Qwen and Llama, despite a *decline* in performance HAR-
MONY with scale (§4.2), $H_\Delta$ *increases* with the model scale, as larger models distribute their gains
more evenly across subsets, whereas smaller variants exhibit spikier changes. Gemma shows the
complementary pattern, where its larger models, which had higher performance HARMONY, dis-
play *lower* $H_\Delta$, indicating that improvements concentrate on fewer subsets as scale grows. By
contrast, in OLMo model family, both performance HARMONY and improvement HARMONY *rise*
with model size. Taken together, these results underscore that aggregate HARMONY and improve-
ment HARMONY can decouple, since models may become less harmonious overall yet still scale
their improvements uniformly, or vice versa.

## G    DETAILS OF STATISTICAL SIGNIFICANCE TEST

We assess whether the subset we keep after pruning has a higher mean than the full set using a
nonparametric, *coupled* bootstrap sign test. Let $a_1, \ldots, a_N$ be per-example accuracy and $k_i \in \{0, 1\}$
indicate membership in the keep subset $K = \{i : k_i = 1\}$. For $b = 1, \ldots, B$, we draw a bootstrap

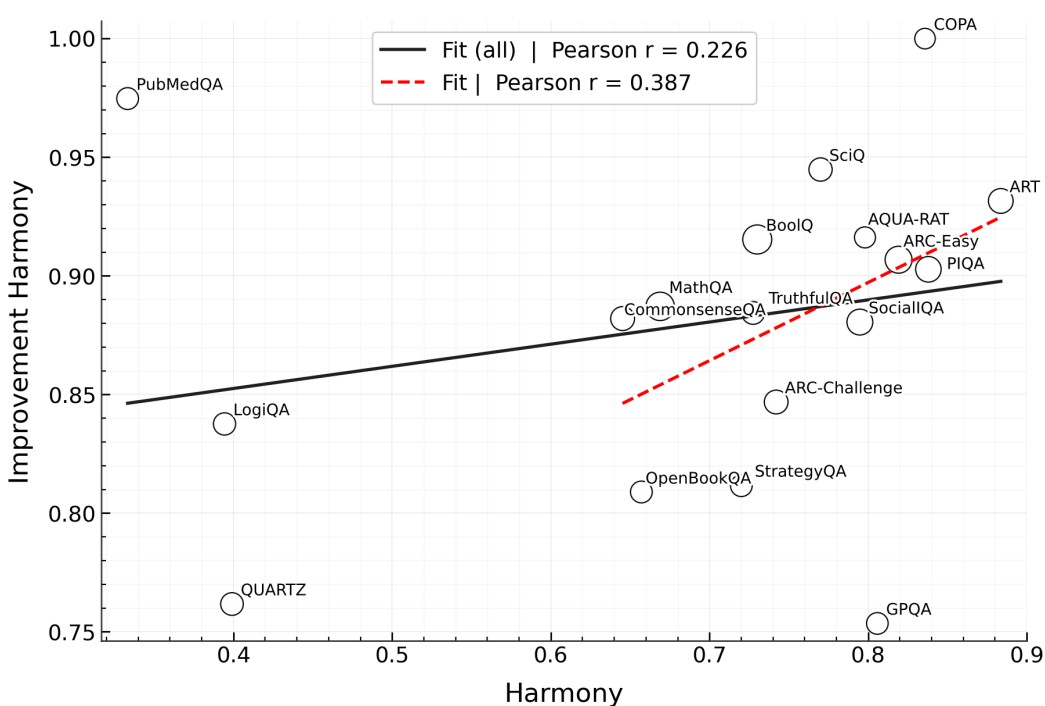

Figure 13: $H_\Delta$ **across benchmarks.** Higher performance HARMONY modestly correlates with improvement HARMONY ($r = 0.226$; $r = 0.387$ excluding the three lowest HARMONY) benchmarks, indicating an outlier-sensitive correlation.

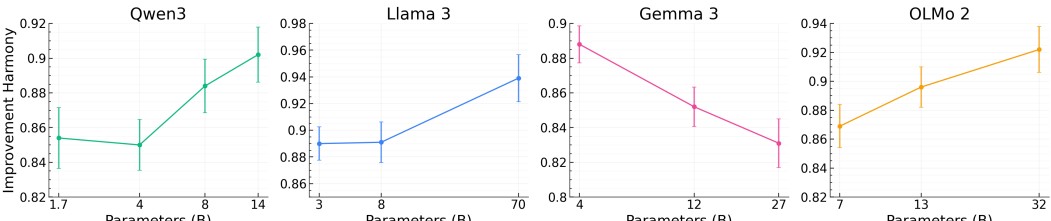

Figure 14: **Model size vs. $H_\Delta$.** Improvement harmony scales differently by family: it increases with size for `Qwen`, `Llama`, and `OLMo`, while decreasing for `Gemma`.

sample of indices $S^{(b)}$ of size $N$ (with replacement), compute both means on the *same* resample, and then take their difference:

$$\bar{a}_{\text{all}}^{(b)} = \frac{1}{N} \sum_{i \in S^{(b)}} a_i, \tag{3}$$

$$n_{\text{keep}}^{(b)} = \sum_{i \in S^{(b)}} k_i, \tag{4}$$

$$\bar{a}_{\text{keep}}^{(b)} = \frac{1}{n_{\text{keep}}^{(b)}} \sum_{i \in S^{(b)}} k_i \, a_i, \tag{5}$$

$$\Delta^{(b)} = \bar{a}_{\text{keep}}^{(b)} - \bar{a}_{\text{all}}^{(b)}. \tag{6}$$

Resamples with $n_{\text{keep}}^{(b)} = 0$ are discarded (to avoid degenerate runs we cap total draws at $3B$); let $m \le B$ be the number of valid differences retained. We then form a two-sided $p$-value from the sign statistic with a plus-one small-sample correction:

$$r = \sum_{b=1}^{m} \mathbf{1}\left\{ \Delta^{(b)} \ge 0 \right\}, \tag{7}$$

$$p = \min\left\{ 1, \, 2\min\left( \frac{r+1}{m+1}, \, \frac{m-r+1}{m+1} \right) \right\}. \tag{8}$$

We fix the random seed for reproducibility and declare significance at level $\alpha$ when $p < \alpha$ (testing $H_0 : \mathbb{E}[\Delta] = 0$ vs. $H_1 : \mathbb{E}[\Delta] \ne 0$). We use $B = 10000$ and set $\alpha(N)$ as follows:

$$\alpha(N) = \begin{cases} 0.1, & N < 500, \\ 0.05, & 500 \le N < 1500, \\ 0.01, & 1500 \le N < 3000 \end{cases}$$

## H    EXTENDED RESULTS: HOW DOES MODEL PERFORMANCE CHANGE WITH INCREASED HARMONY?

In this section, we generalize the pruning experiments from §4.1 beyond the illustrative cases to *all* model families and benchmarks in our setup. Our aim is methodological: we examine how aggregate accuracy and per-subset dispersion evolve as we progressively rebalance a benchmark. Concretely, for each (model, benchmark) pair we sweep a pruning budget (scheduled inversely to baseline HARMONY), recompute HARMONY and accuracy at each budget, and compare the pruned-set accuracy to the full-set accuracy using the coupled bootstrap significance test detailed in App. G. Family-wise plots in this section visualize these trajectories, allowing us to observe whether increased HARMONY coincides with stable (or shifting) aggregate scores and tighter per-subset distribution of performance.

Across all model families (Fig. 15, 16, 17, 18), two patterns consistently hold. **(i) Accuracy shifts with increased harmony.** As pruning raises HARMONY, aggregate accuracy frequently changes in a statistically significant manner (App. G), indicating that low HARMONY composition can result in a misleading aggregate score. **(ii) Low HARMONY benchmarks are fragile.** Benchmarks starting with lower HARMONY exhibit more instances of significant accuracy change under the pruning procedure than high HARMONY benchmarks, underscoring their susceptibility to presenting misleading aggregate scores.

## I    MULTI-DIMENSIONAL EVALUATION

Motivated by the skewed aggregate scores in low HARMONY benchmarks, we conduct model evaluation at finer granularity. Following recent work on fine-grained evaluation (Zeng et al., 2025), we recursively induce partitions as described in §2.3. This procedure yields a *labeled tree*, where the root is the full benchmark; each internal node is a subset from the partitioning of its parent node; and

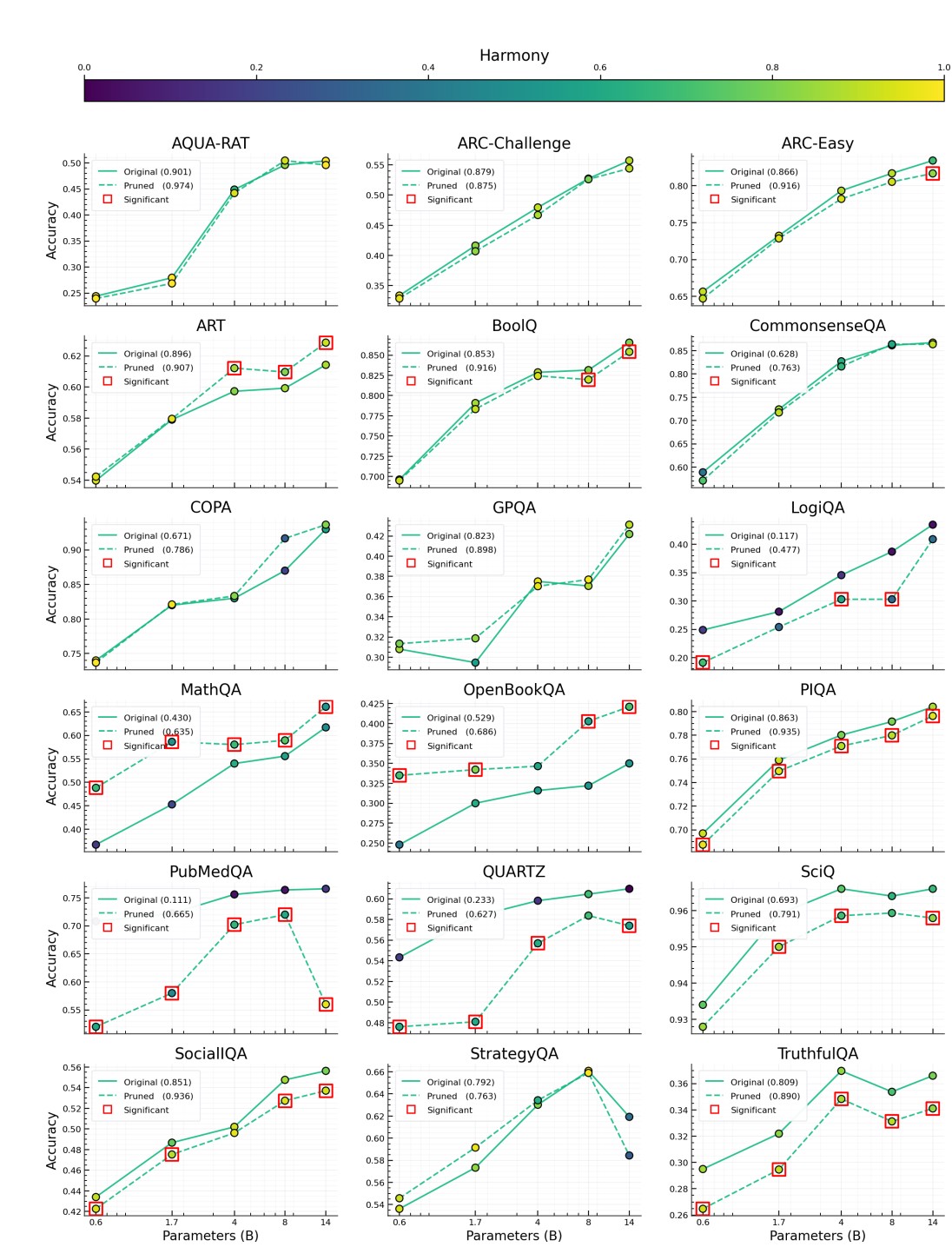

Figure 15: **Full results of balancing benchmarks via pruning in Qwen3 model family.** We remove overly similar items with a pruning rate inversely proportional to HARMONY, which consistently improves HARMONY. We find that aggregate scores often change statistically significantly on less harmonious benchmarks, whereas they remain more stable on more harmonious benchmarks.

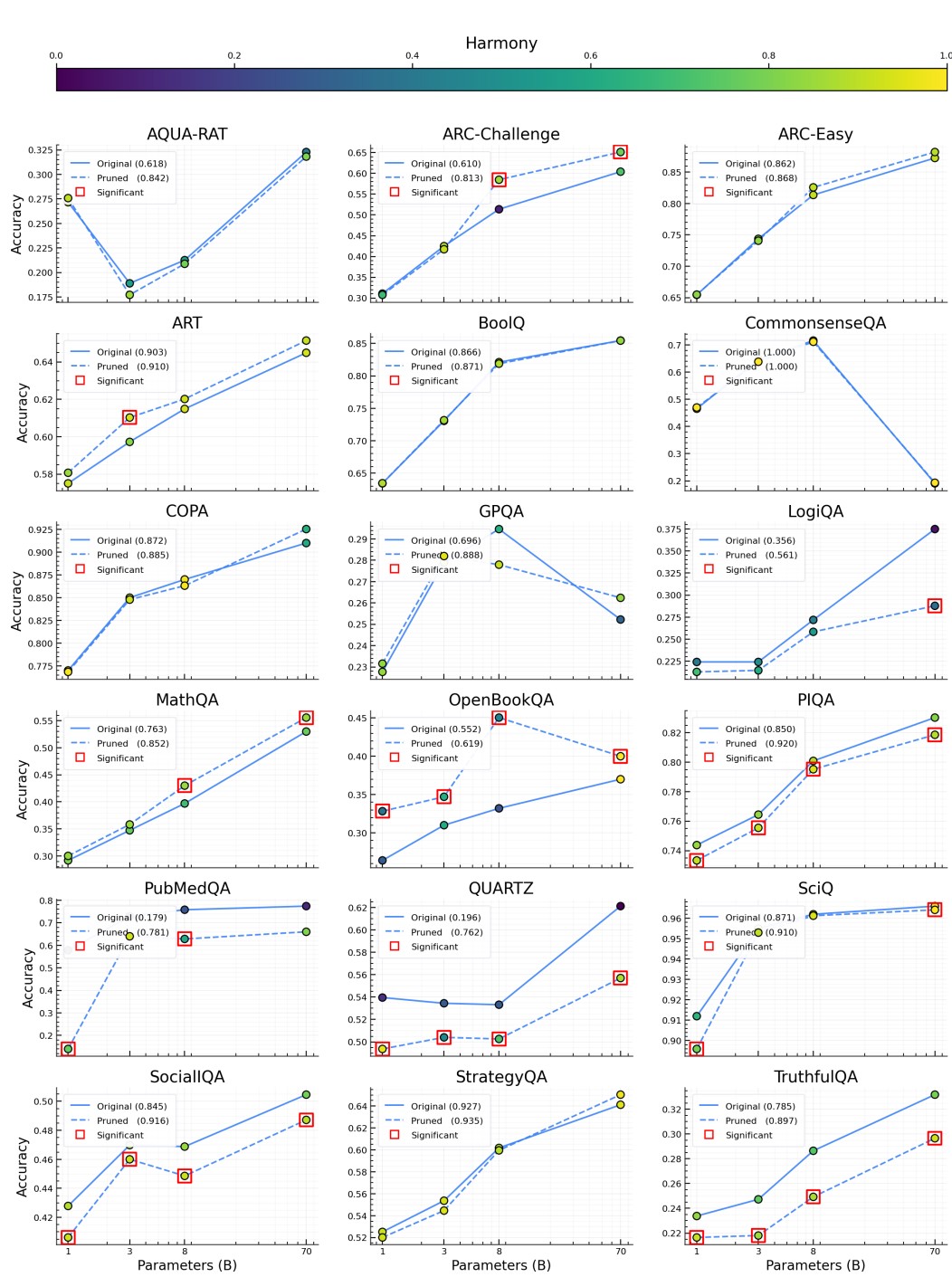

Figure 16: **Full results of balancing benchmarks via pruning in Llama 3 model family.** We remove overly similar items with a pruning rate inversely proportional to HARMONY, which consistently improves HARMONY. We find that aggregate scores often change statistically significantly on less harmonious benchmarks, whereas they remain more stable on more harmonious benchmarks.

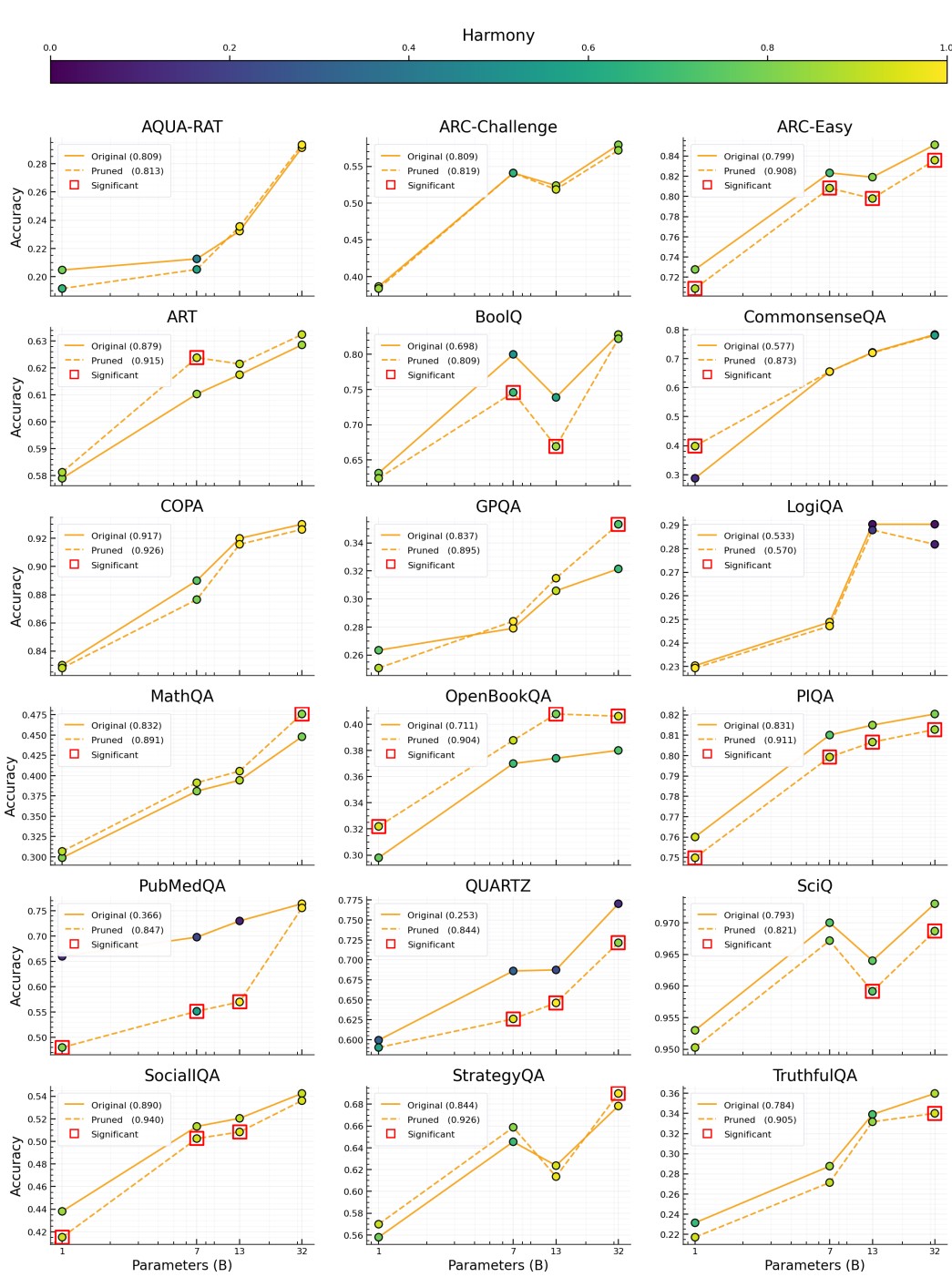

Figure 17: **Full results of balancing benchmarks via pruning in Olmo 2 model family.** We remove overly similar items with a pruning rate inversely proportional to HARMONY, which consistently improves HARMONY. We find that aggregate scores often change statistically significantly on less harmonious benchmarks, whereas they remain more stable on more harmonious benchmarks.

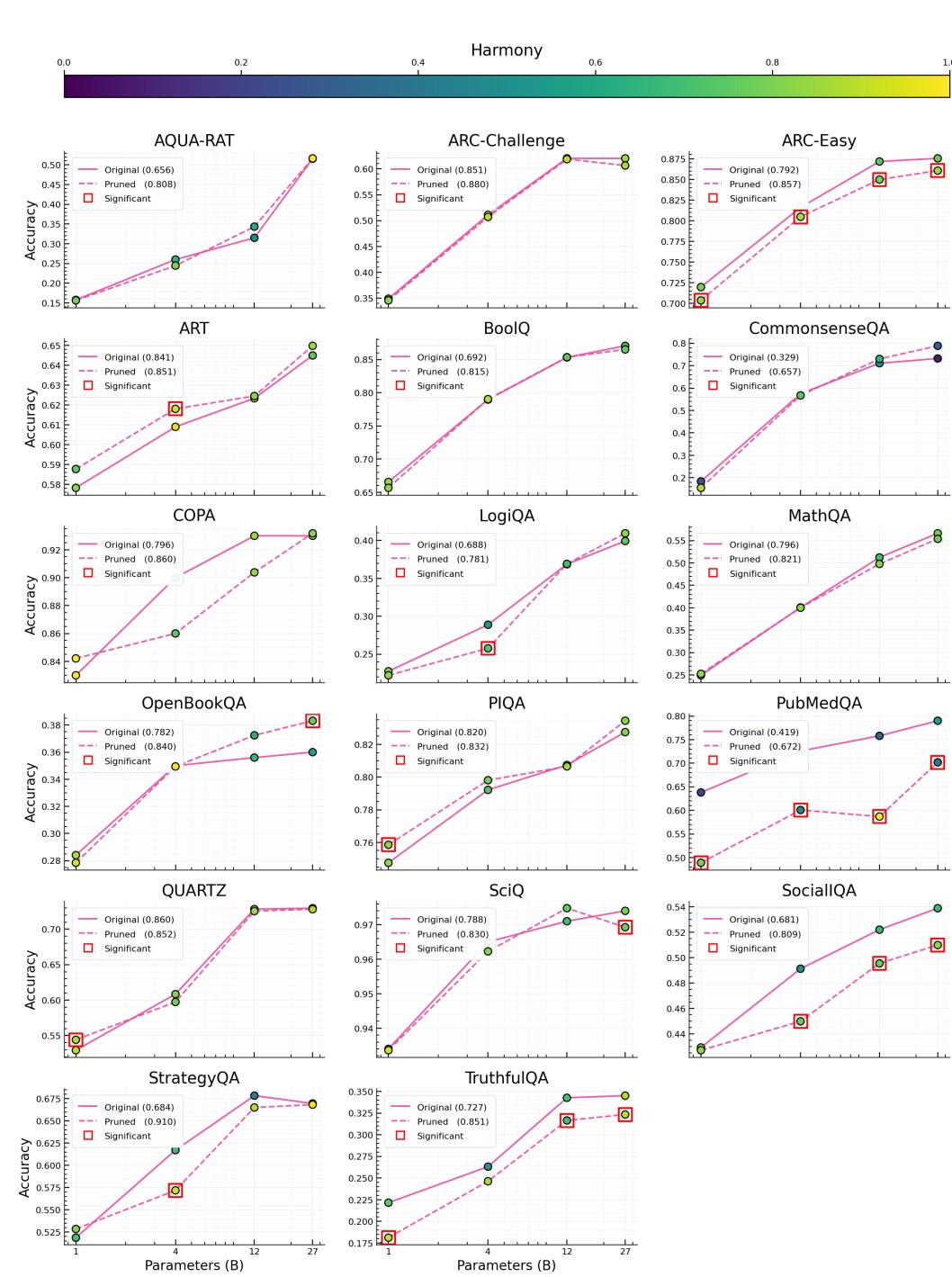

Figure 18: **Full results of balancing benchmarks via pruning in Gemma 3 model family.** We remove overly similar items with a pruning rate inversely proportional to HARMONY, which consistently improves HARMONY. We find that aggregate scores often change statistically significantly on less harmonious benchmarks, whereas they remain more stable on more harmonious benchmarks.

| | Qwen3-0.6B | Qwen3-1.7B | Qwen3-4B | Qwen3-8B | Qwen3-14B |
|---|---|---|---|---|---|
| Overall | 47.2 | 68.1 | 82.6 | 86.8 | **91.0** |
| Multicellular Biology (42.4%) | 45.9 | 68.9 | 88.5 | 88.5 | **93.4** |
| Evolutionary & Ecological Processes (25.7%) | 51.4 | 59.5 | 86.5 | **91.9** | **91.9** |
| Molecular & Cellular Biology (31.9%) | 45.7 | 73.9 | 71.7 | 80.4 | **87.0** |
| HARMONY | 0.951 | 0.918 | 0.757 | 0.784 | 0.859 |

Table 8: Multi-dimensional evaluation results of Qwen3 model family in MMLU College Biology. **Bold** implies the best performance.

| | Qwen3-0.6B | Qwen3-1.7B | Qwen3-4B | Qwen3-8B | Qwen3-14B |
|---|---|---|---|---|---|
| Overall | 24.5 | 34.3 | 58.8 | 57.8 | **69.6** |
| Quantum Mechanics Principles and Applications (22.5%) | 56.5 | 34.8 | 65.2 | 69.6 | **73.9** |
| Thermodynamics (8.8%) | 0.0 | 55.6 | 66.7 | 55.6 | **77.8** |
| Special Relativity Concepts (13.7%) | 21.4 | 14.3 | **71.4** | 50.0 | 57.1 |
| Classical Physics Principles and Relationships (20.6%) | 9.5 | 38.1 | 47.6 | 52.4 | **61.9** |
| Physics Phenomena and Application (16.7%) | 0.0 | 11.8 | 11.8 | 17.6 | **58.8** |
| Electromagnetism (5.9%) | 16.7 | 16.7 | **83.3** | **83.3** | 66.7 |
| Solid State Physics (11.8%) | 50.0 | 75.0 | **100.0** | **100.0** | **100.0** |
| HARMONY | 0.686 | 0.712 | 0.765 | 0.815 | 0.789 |

Table 9: Multi-dimensional evaluation results of Qwen3 model family in MMLU College Physics. **Bold** implies the best performance.

leaves are atomic subdomains that admit no further valid split (see App. L for details). The resulting hierarchy enables interpretable, multi-dimensional evaluation, where each dimension corresponds to a subdomain of the benchmark.

We illustrate this approach on two examples: *MMLU College Biology* and *MMLU College Physics*. The average HARMONY across Qwen3 models is markedly higher for biology (0.8538) than for physics (0.7534). This suggests that the aggregate accuracy for biology is a more representative reflection of the performance across subdomains. Indeed, Table 8 shows that rankings in biology subdomains mirror the overall ordering: models that achieve higher overall accuracy also achieve higher accuracy in every subdomain. In other words, no model with superior overall performance is ever surpassed by a model with lower overall performance in any biology subdomain.[13] This alignment underscores that the aggregate score is a consistent and reliable summary of subdomain performance in biology.

In contrast, physics exhibits lower HARMONY and more notable divergences (Table 9). For example, Qwen3-4B lags behind Qwen3-14B in overall accuracy (58.8% vs. 69.6%), yet it surpasses it in *Special Relativity* (71.4% vs. 57.1%) and *Electromagnetism* (83.3% vs. 66.7%). Similarly, Qwen3-0.6B, despite its weak overall score (24.5%), achieves competitive performance in *Quantum Mechanics* (56.5%), outperforming Qwen3-1.7B (34.8%). These cases highlight how aggregate scores can obscure areas of relative strength, and how fine-grained, multi-dimensional evaluation reveals nuanced interpretation of model competence across subdomains.

We provide extended results for Qwen3 and Llama 3 model families across 2 MCQA benchmarks and 6 MMLU subtasks in Appendix J.

## J EXTENDED RESULTS: MULTI-DIMENSIONAL EVALUATION

We provide the extended results of multi-dimensional evaluation conducted as described in Appendix I. Our setup consists of Qwen3 and Gemma 3 model families and ARC-Easy, BoolQ, MMLU Anatomy, MMLU College Biology, MMLU College Computer Science, MMLU College Mathematics, MMLU College Physics, MMLU High School US History.

---

[13]Comparison of 1.7B and 4B in *Molecular & Cellular Biology* is the only exception.

| | Qwen3-0.6B | Qwen3-1.7B | Qwen3-4B | Qwen3-8B | Qwen3-14B |
|---|---|---|---|---|---|
| Overall | 60.7 | 72.2 | 80.5 | 83.5 | 84.2 |
| Geology and Earth Sciences (15.0%) | 64.4 | 73.4 | 83.8 | 83.8 | 83.8 |
| Scientific Principles and Processes (32.6%) | 56.3 | 70.1 | 79.9 | 83.5 | 84.3 |
| Biological Processes and Concepts (24.1%) | 65.4 | 74.0 | 78.8 | 82.9 | 83.6 |
| Physics Principles in Engineering and Science (9.4%) | 64.3 | 75.9 | 82.1 | 86.6 | 86.6 |
| Environmental and Energy Assessment (6.5%) | 60.4 | 63.6 | 78.6 | 81.2 | 79.9 |
| Fundamental Concepts in Astronomy (7.6%) | 56.1 | 76.7 | 80.6 | 83.3 | 87.2 |
| Fundamentals of Chemical and Material Properties (4.8%) | 56.1 | 71.9 | 81.6 | 84.2 | 84.2 |

Table 10: Multi-dimensional evaluation results for the Qwen3 model family on ARC-Easy.

| | Qwen3-0.6B | Qwen3-1.7B | Qwen3-4B | Qwen3-8B | Qwen3-14B |
|---|---|---|---|---|---|
| Overall | 64.1 | 77.5 | 85.0 | 86.6 | 89.3 |
| Product Composition, Properties, and Standards (8.8%) | 66.8 | 80.6 | 83.7 | 86.9 | 88.9 |
| Geographic, Operational, and Temporal Analysis (11.0%) | 69.3 | 77.3 | 85.3 | 88.4 | 91.7 |
| Media Standards and Analysis (24.8%) | 61.3 | 77.7 | 86.8 | 88.7 | 91.0 |
| Scientific and Analytical Principles (9.0%) | 67.9 | 81.2 | 83.3 | 84.6 | 89.1 |
| Sports History and Regulations (10.1%) | 63.5 | 75.1 | 82.1 | 84.5 | 87.2 |
| Governmental Laws and Regulations (10.6%) | 64.1 | 72.8 | 80.3 | 84.6 | 86.1 |
| Human Biology and Medical Science (6.5%) | 60.6 | 81.2 | 87.8 | 85.4 | 87.8 |
| Economic Systems (8.3%) | 61.6 | 75.3 | 85.2 | 85.6 | 87.5 |
| Sociocultural, Geopolitical, and Linguistic Analysis (7.1%) | 63.9 | 76.8 | 88.0 | 86.7 | 89.7 |
| Fictional Narrative Analysis and Elements (3.8%) | 64.8 | 80.8 | 89.6 | 88.0 | 93.6 |

Table 11: Evaluation results for the Qwen3 model family on BoolQ.

## J.1 QWEN3 FAMILY

See Tables 10, 11, 12, 13, 14, 15, 16, and 17 for the extended results of Qwen3 model family.

## J.2 GEMMA 3 FAMILY

See Tables 18, 19, 20, 21, 22, 23, 24, and 25 for the extended results of Gemma 3 model family.

## K ADDITIONAL RELATED WORK

**Language Model Evaluation.** Reliable evaluation is essential for accurately assessing model capabilities and enabling fair comparisons, which in turn informs future developments. Hence, there has been a surge in the development of benchmarks designed to test various model capabilities, such as reasoning (Bisk et al., 2019; Sap et al., 2019b; Zellers et al., 2019; Liu et al., 2020), world knowledge (Mihaylov et al., 2018b; Hendrycks et al., 2020; et al., 2023), and truthfulness (Lin et al., 2022; Khatun & Brown, 2024). Beyond individual benchmarks, holistic frameworks have emerged to offer a more comprehensive assessment of model performance (Liang et al., 2023; Chiang et al., 2024; Gao et al., 2024; Fourrier et al., 2023; et al., 2023). Reciprocally, understanding and improving current benchmarks have been equally important. MMLU Pro (Wang et al., 2024b) and Big-Bench-Hard (Suzgun et al., 2022) address benchmark saturation by constructing more challenging variants of MMLU (Hendrycks et al., 2020) and Big-Bench (bench authors, 2023) respectively. As top models approach ceiling effects on narrow probes, evaluation has shifted toward complex end-to-end tasks and composite suites. HLE and ARC-AGI assess multi-step reasoning, tool use, and robustness across domains (Phan et al., 2025; Chollet et al., 2025). Execution-grounded tasks such as SWE-bench measure real-world software problems and end-to-end correctness (Jimenez et al., 2024). Competitive exams like AIME and IMO, and professional exams such as the bar, push systems toward expert-level competence. Another recent practice is evaluation with online leaderboards, which use hidden test sets, fixed prompts, and compute disclosures in order to support fair comparison and consistent progress tracking (Chiang et al., 2024). Yet, these advances rest on a common premise that benchmarks reliably evaluate models on their stated domains. We audit this premise by testing whether benchmarks provide balanced coverage and promote comparable performance across subdomains.

|  | Qwen3-0.6B | Qwen3-1.7B | Qwen3-4B | Qwen3-8B | Qwen3-14B |
|---|---|---|---|---|---|
| Overall | 37.8 | 56.3 | 62.2 | 71.1 | 80.7 |
| General Human Anatomy, Physiology, and Terminology (31.9%) | 51.2 | 67.4 | 79.1 | 76.7 | 86.0 |
| Head and Neck Anatomy (21.5%) | 20.7 | 34.5 | 34.5 | 62.1 | 82.8 |
| Skeletal Development and Anatomy (14.1%) | 26.3 | 42.1 | 47.4 | 52.6 | 57.9 |
| Neurological Disorders (10.4%) | 21.4 | 50.0 | 50.0 | 71.4 | 78.6 |
| Bone Anatomy and Terminology (3.0%) | 0.0 | 50.0 | 50.0 | 50.0 | 50.0 |
| Anatomy of Circulatory System (2.2%) | 66.7 | 100.0 | 100.0 | 100.0 | 100.0 |
| Developmental Structures (6.7%) | 22.2 | 44.4 | 55.6 | 77.8 | 88.9 |
| Nephrology (10.4%) | 78.6 | 92.9 | 100.0 | 92.9 | 92.9 |

Table 12: Evaluation results for the Qwen3 model family on MMLU Anatomy.

|  | Qwen3-0.6B | Qwen3-1.7B | Qwen3-4B | Qwen3-8B | Qwen3-14B |
|---|---|---|---|---|---|
| Overall | 47.2 | 68.1 | 82.6 | 86.8 | 91.0 |
| Multicellular Biology (42.4%) | 45.9 | 68.9 | 88.5 | 88.5 | 93.4 |
| Evolutionary and Ecological Processes (25.7%) | 51.4 | 59.5 | 86.5 | 91.9 | 91.9 |
| Molecular and Cellular Biology (31.9%) | 45.7 | 73.9 | 71.7 | 80.4 | 87.0 |

Table 13: Evaluation results for the Qwen3 model family on MMLU College Biology.

## L  HIERARCHICAL LABELING FOR MULTI-DIMENSIONAL EVALUATION

We build a tree benchmark over questions, then assign concise, human-readable labels to every node. Leaves summarize the shared evaluation focus of their questions, while internal nodes summarize their children.

To build the tree, we recursively induce partitions as discussed in §2.3, starting from the root (i.e., the entire benchmark) and ending at leaves (i.e., the clusters that do not admit a valid partition). For labeling leaves, we gather brief question annotations within a leaf and ask a model for one specific noun-phrase label. For labeling internal nodes of the tree, we pass the child labels to the model and ask for a slightly more abstract label that still captures the shared theme. Therefore, this procedure yields a bottom-up label propagation from leaves to internal nodes then to the root. We use `Gemini-2.0-flash` to annotate individual questions, assign each leaf a label from its question annotations, and propagate labels upward by aggregating child labels.

**Prompts.** We share the prompts we use for annotating the questions (Prompt L), labeling the leaves (Prompt L), and labeling the internal nodes (Prompt L).

---

**Prompt for annotating questions.**

You are given a question from the BENCHMARK benchmark.
Given this question, generate a single, concise sentence that clearly describes the **specific evaluation focus** of the question.
Question: QUESTION
Requirements:
- Do not have a prefix, simply provide a brief phrase or a gerund.
- Do not add commentary.

---

[Reviewer F1zp: [Reviewer 5RAY:

## M  EXAMPLE CLUSTERS FROM MULTI-DIMENSIONAL EVALUATION

We present qualitative examples of the clusters produced by our hierarchical spectral clustering method, which groups questions using predictive similarity. Tables 26 and 27 illustrate representative clusters for ARC-Easy and BoolQ, respectively. Notably, successful hierarchical clustering requires coherence at each depth level, and these examples show that our method produces consistently coherent clusters even at lower depths, which is a particularly stringent and informative criterion.

| | Qwen3-0.6B | Qwen3-1.7B | Qwen3-4B | Qwen3-8B | Qwen3-14B |
|---|---|---|---|---|---|
| Overall | 28.0 | 41.0 | 66.0 | 72.0 | 68.0 |
| Theoretical Foundations of Computation (52.0%) | 26.9 | 40.4 | 59.6 | 69.2 | 65.4 |
| Computer Architecture and Optimization (7.0%) | 28.6 | 42.9 | 71.4 | 57.1 | 57.1 |
| Operating Systems (10.0%) | 50.0 | 80.0 | 90.0 | 90.0 | 80.0 |
| Network Layer Protocols and Technologies (5.0%) | 60.0 | 60.0 | 80.0 | 100.0 | 100.0 |
| Data Processing (12.0%) | 8.3 | 8.3 | 66.7 | 50.0 | 66.7 |
| Sorting Algorithms (4.0%) | 25.0 | 25.0 | 75.0 | 100.0 | 100.0 |
| Graph Algorithms and Data Structures (10.0%) | 20.0 | 40.0 | 60.0 | 80.0 | 50.0 |

Table 14: Evaluation results for the Qwen3 model family on MMLU College Computer Science.

| | Qwen3-0.6B | Qwen3-1.7B | Qwen3-4B | Qwen3-8B | Qwen3-14B |
|---|---|---|---|---|---|
| Overall | 31.0 | 39.0 | 55.0 | 59.0 | 68.0 |
| Advanced Real Analysis (19.0%) | 26.3 | 26.3 | 42.1 | 52.6 | 52.6 |
| Abstract Algebra (11.0%) | 27.3 | 45.5 | 90.9 | 63.6 | 81.8 |
| Probability (7.0%) | 28.6 | 71.4 | 28.6 | 57.1 | 57.1 |
| Properties of Mathematical Operations and Functions (5.0%) | 20.0 | 20.0 | 40.0 | 80.0 | 80.0 |
| Advanced Mathematical Concepts and Applications (16.0%) | 31.3 | 37.5 | 75.0 | 56.3 | 75.0 |
| Mathematical Modeling and Algorithms (10.0%) | 30.0 | 50.0 | 60.0 | 60.0 | 70.0 |
| Multivariable Calculus (27.0%) | 25.9 | 37.0 | 40.7 | 59.3 | 63.0 |
| Mathematical Optimization Methods (5.0%) | 100.0 | 40.0 | 80.0 | 60.0 | 100.0 |

Table 15: Evaluation results for the Qwen3 model family on MMLU College Mathematics.

]]

[Reviewer 5RAY:

# N   ASSESSING BENCHMARK SIZE AS A CONFOUND

A natural concern is that benchmark size might confound our mean-variance characterization of Harmony. In this section, we show analytically that Harmony is invariant to uniform rescaling of benchmark size and empirically that benchmark size does not explain the characteristics observed in the mean-variance plane.

Recall that, given a partition $G_f = \{A_i\}_{i=1}^k$ of benchmark $\mathcal{B}$, Harmony is computed using the relative subset weights $w_i = |A_i|/|\mathcal{B}|$ and a normalized Shannon entropy over proximity-based scores, scaled by $\log k$. Because only the normalized weights $\{w_i\}$ and the relative distribution of performance across subdomains are present in the definition, operations that uniformly scale the benchmark size (e.g., duplicating all items or uniformly sub-sampling) should leave Harmony unchanged in principle. Harmony is therefore designed to reflect how coverage and performance are distributed across semantic subdomains in a manner that is not impacted by the number of examples given that subdomain proportions remain the same.

To assess benchmark size as a potential confound in our setting, we first compute the Pearson correlation between benchmark size and (i) the mean of Harmony across models and (ii) the variance of

---

**Prompt for labeling leaves.**

You are a taxonomy assistant. Your task is to read short annotations that describe what each question evaluates and produce one concise but descriptive label that summarizes the shared knowledge or concept.
Guidelines:
- The label must be highly specific, directly capturing the core idea, while still generalizable across closely related items.
- Prioritize specificity: avoid vague or overly broad terms.
- Use a clear noun phrase.
- Return only the label text.

| | Qwen3-0.6B | Qwen3-1.7B | Qwen3-4B | Qwen3-8B | Qwen3-14B |
|---|---|---|---|---|---|
| Overall | 24.5 | 34.3 | 58.8 | 57.8 | 69.6 |
| Quantum Mechanics Principles and Applications (22.5%) | 56.5 | 34.8 | 65.2 | 69.6 | 73.9 |
| Thermodynamics (8.8%) | 0.0 | 55.6 | 66.7 | 55.6 | 77.8 |
| Special Relativity Concepts (13.7%) | 21.4 | 14.3 | 71.4 | 50.0 | 57.1 |
| Classical Physics Principles and Relationships (20.6%) | 9.5 | 38.1 | 47.6 | 52.4 | 61.9 |
| Physics Phenomena and Applications (16.7%) | 0.0 | 11.8 | 11.8 | 17.6 | 58.8 |
| Electromagnetism (5.9%) | 16.7 | 16.7 | 83.3 | 83.3 | 66.7 |
| Solid State Physics Concepts (11.8%) | 50.0 | 75.0 | 100.0 | 100.0 | 100.0 |

Table 16: Evaluation results for the Qwen3 model family on MMLU College Physics.

| | Qwen3-0.6B | Qwen3-1.7B | Qwen3-4B | Qwen3-8B | Qwen3-14B |
|---|---|---|---|---|---|
| Overall | 52.5 | 66.2 | 83.3 | 88.7 | 91.7 |
| US Sociopolitical Ideologies, Movements, and Issues (46.1%) | 52.1 | 64.9 | 83.0 | 90.4 | 95.7 |
| Progressive Era Economic and Social Initiatives (3.9%) | 75.0 | 87.5 | 100.0 | 100.0 | 100.0 |
| United States Governance and Politics (37.3%) | 46.1 | 61.8 | 81.6 | 85.5 | 88.2 |
| Ideological and Territorial Expansion in the Americas (9.8%) | 70.0 | 80.0 | 90.0 | 95.0 | 85.0 |
| American History Eras (2.9%) | 50.0 | 66.7 | 66.7 | 66.7 | 83.3 |

Table 17: Evaluation results for the Qwen3 model family on MMLU High School U.S. History.

Harmony across models for the set of 18 MCQA benchmarks in our experimental setup. We obtain:

$$\mathbb{E}_{f \sim \mathcal{F}}[H_{\mathcal{B}}(f)] \text{ vs. size: } r = -0.310, \quad p = 0.2105$$
$$\text{Var}_{f \sim \mathcal{F}}(H_{\mathcal{B}}(f)) \text{ vs. size: } r = 0.227, \quad p = 0.3653$$

The associated $p$-values show no statistically significant evidence of linear dependence between benchmark size and either axis of the mean-variance plane. In particular, larger benchmarks do not exhibit systematic shifts in mean Harmony or in cross-model variance, indicating that benchmark size does not explain the structure observed in Figure 3.

Second, we perform a controlled sub-sampling experiment on three benchmarks selected from different regions of the mean-variance plane. For each benchmark, we apply several down-sampling ratios uniformly, hence preserving the relative frequencies of semantic subdomains. We then re-compute Harmony across models. The resulting trajectories in the mean-variance plane are shown in Figure 19. Across all three benchmarks, we observe only minor fluctuations in both the mean and variance of Harmony, and the points remain within the same qualitative regions of the plane.

Taken together, the size-invariance of Harmony under uniform rescaling, the lack of statistically significant correlations between benchmark size and Harmony statistics, and the stability observed in the sub-sampling experiment support our view that the mean-variance plane captures differences in *distributional reliability*, rather than merely reflecting benchmark size.]

[Reviewer 5RAY:

## O  ASSESSING $k$ AS A CONFOUND

Our construction of Harmony relies on a partition of each benchmark into $k$ semantic subdomains. A natural concern is that our choice of $k$ might introduce a confound, if $k$ systematically tracked Har-

---

**Prompt for labeling internal nodes.**

You are a taxonomy assistant. Your task is to read the labels of child clusters and generate one concise but descriptive parent label that captures their common theme at a higher level of abstraction.
Guidelines:
- The label must be specific and clearly meaningful, while still broad enough to encompass all children.
- Prioritize specificity: avoid vague or generic terms that do not capture the essence of the group.
- Use a clear noun phrase.
- Return only the label text.

| | Gemma 3 1B | Gemma 3 4B | Gemma 3 12B | Gemma 3 27B |
|---|---|---|---|---|
| Overall | 72.0 | 81.6 | 87.2 | 87.5 |
| Geology and Earth Sciences (15.0%) | 74.5 | 84.6 | 87.7 | 89.4 |
| Scientific Principles and Processes (32.6%) | 71.2 | 79.7 | 86.5 | 87.0 |
| Biological Processes and Concepts (24.1%) | 74.3 | 83.0 | 88.5 | 87.8 |
| Physics Principles in Engineering and Science (9.4%) | 70.5 | 83.0 | 88.8 | 88.8 |
| Environmental and Energy Assessment (6.5%) | 68.2 | 80.5 | 83.1 | 84.4 |
| Fundamental Concepts in Astronomy (7.6%) | 72.8 | 79.4 | 86.7 | 87.2 |
| Fundamentals of Chemical and Material Properties (4.8%) | 64.0 | 79.8 | 86.8 | 86.8 |

Table 18: Multi-dimensional evaluation results for the Gemma 3 model family on ARC-Easy.

| | Gemma 3 1B | Gemma 3 4B | Gemma 3 12B | Gemma 3 27B |
|---|---|---|---|---|
| Overall | 66.5 | 79.0 | 85.3 | 87.1 |
| Product Composition, Properties, and Standards (8.8%) | 67.5 | 76.1 | 82.4 | 88.6 |
| Geographic, Operational, and Temporal Analysis (11.0%) | 67.6 | 83.7 | 84.8 | 87.3 |
| Media Standards and Analysis (24.8%) | 66.7 | 80.6 | 88.5 | 89.6 |
| Scientific and Analytical Principles (9.0%) | 65.2 | 75.1 | 82.9 | 86.0 |
| Sports History and Regulations (10.1%) | 63.2 | 78.7 | 83.0 | 86.0 |
| Governmental Laws and Regulations (10.6%) | 64.1 | 75.7 | 80.3 | 82.0 |
| Human Biology and Medical Science (6.5%) | 77.0 | 82.6 | 86.9 | 89.2 |
| Economic Systems (8.3%) | 66.1 | 77.1 | 85.6 | 85.6 |
| Sociocultural, Geopolitical, and Linguistic Analysis (7.1%) | 61.8 | 78.1 | 87.6 | 84.1 |
| Fictional Narrative Analysis and Elements (3.8%) | 71.2 | 79.2 | 91.2 | 90.4 |

Table 19: Multi-dimensional evaluation results for the Gemma 3 model family on BoolQ.

mony in a way that would inflate or suppress our reliability measure. In this section, we empirically study $k$'s relationship to benchmark size and Harmony.

As described in Sec. 2.3, for each benchmark we sweep $k \in \{2, \ldots, 20\}$ and select the value that maximizes the silhouette score, a well established metric of clustering quality (Rousseeuw, 1987). This procedure determines $k$ in a fully data-driven manner based on the geometry of the representation space, and $k$ is never hand-tuned to optimize Harmony.

To quantify how $k$ relates to benchmark size and HARMONY, we compute Pearson correlations between (i) benchmark size and the selected $k$, and (ii) $k$ and the HARMONY for each benchmark-model pair. We obtain:

$$\text{HARMONY vs. } k: \quad r = 0.274, \quad p = 0.2818,$$
$$\text{Benchmark size vs. } k: \quad r = 0.914, \quad p = 0.0000.$$

The strong correlation between benchmark size and $k$ confirms the intuitive behavior that larger benchmarks admit more clusters. Crucially, however, there is no statistically significant linear relationship between $k$ and Harmony: the correlation coefficient is weak and the associated $p$-value indicates no evidence against the null hypothesis of zero correlation at standard significance levels.

This distinction matters because our conclusions rely on Harmony rather than $k$ or benchmark size itself. Harmony already accounts for varying cluster sizes through the weights $w_i = |A_i|/|B|$, so differences in the number of clusters primarily reflect how finely the benchmark can be partitioned, not a direct change in the reliability measure. The absence of a statistically significant correlation between $k$ and Harmony indicates that our reliability measure is not spuriously driven by the specific value of $k$, even though $k$ increases with benchmark size as expected.]

[Reviewer 85MX:

## P    ASSESSING PRUNING RATIO AS A CONFOUND

To explicitly rule out pruning ratio as a potential confound in the experiments of Sec. 4.1, we repeat the pruning procedure while holding the pruning budget fixed across all benchmarks. Concretely, we evaluate five ratios, $p \in \{0.1, 0.2, 0.3, 0.4, 0.5\}$ and report the corresponding results in Fig. 20-24.

Across all pruning ratios, we observe the same pattern: (i) pruning reliably increases Harmony, (ii) benchmarks with low Harmony exhibit statistically significant accuracy shifts, and (iii) benchmarks

| | Gemma 3 1B | Gemma 3 4B | Gemma 3 12B | Gemma 3 27B |
|---|---|---|---|---|
| Overall | 25.9 | 61.5 | 70.4 | 70.4 |
| General Human Anatomy, Physiology, and Terminology (31.9%) | 27.9 | 69.8 | 83.7 | 79.1 |
| Head and Neck Anatomy (21.5%) | 27.6 | 41.4 | 55.2 | 48.3 |
| Skeletal Development and Anatomy (14.1%) | 21.1 | 52.6 | 52.6 | 57.9 |
| Neurological Disorders (10.4%) | 42.9 | 64.3 | 57.1 | 71.4 |
| Bone Anatomy and Terminology (3.0%) | 50.0 | 50.0 | 50.0 | 50.0 |
| Anatomy of Circulatory System (2.2%) | 0.0 | 100.0 | 100.0 | 100.0 |
| Developmental Structures (6.7%) | 11.1 | 44.4 | 77.8 | 88.9 |
| Nephrology (10.4%) | 14.3 | 92.9 | 92.9 | 92.9 |

Table 20: Multi-dimensional evaluation results for the Gemma 3 model family on MMLU Anatomy.

| | Gemma 3 1B | Gemma 3 4B | Gemma 3 12B | Gemma 3 27B |
|---|---|---|---|---|
| Multicellular Biology (42.4%) | 27.9 | 68.9 | 98.4 | 93.4 |
| Evolutionary and Ecological Processes (25.7%) | 21.6 | 67.6 | 86.5 | 86.5 |
| Molecular and Cellular Biology (31.9%) | 21.7 | 65.2 | 87.0 | 91.3 |

Table 21: Multi-dimensional evaluation results for the Gemma 3 model family on MMLU College Biology.

with high Harmony rarely show such shifts, even under more aggressive pruning. Although post pruning accuracies change with the amount of data removed, these effects are stable across pruning budgets. Moreover, the magnitude of the accuracy change tends to saturate, as beyond moderate pruning (around $p = 0.3$), further increases in the pruning ratio produce only marginal additional shifts. Taken together, these results indicate that the results in Figure 4 are not explained by pruning ratio alone and are robust to the choice of pruning budget.

]

# Q  THE USE OF LARGE LANGUAGE MODELS (LLMS)

In this work, we used large language models (LLMs) only for light polishing (grammar, wording, and clarity) after the technical content was written. LLMs were not used for research ideation, experimental design or execution, analysis, figure or table generation, or drafting technical sections. All substantive content, results, and conclusions are authored by the listed authors, who take full responsibility for the paper's contents, including any text edited with LLM assistance. LLMs are not eligible for authorship, and no LLM is listed as an author.

|  | Gemma 3 1B | Gemma 3 4B | Gemma 3 12B | Gemma 3 27B |
|---|---|---|---|---|
| Overall | 30.0 | 48.0 | 57.0 | 63.0 |
| Theoretical Foundations of Computation (52.0%) | 36.5 | 38.5 | 51.9 | 61.5 |
| Computer Architecture and Optimization (7.0%) | 28.6 | 57.1 | 71.4 | 57.1 |
| Operating Systems (10.0%) | 20.0 | 60.0 | 80.0 | 80.0 |
| Network Layer Protocols and Technologies (5.0%) | 20.0 | 60.0 | 100.0 | 100.0 |
| Data Processing (12.0%) | 25.0 | 33.3 | 41.7 | 41.7 |
| Sorting Algorithms (4.0%) | 0.0 | 100.0 | 100.0 | 75.0 |
| Graph Algorithms and Data Structures (10.0%) | 30.0 | 70.0 | 30.0 | 60.0 |

Table 22: Multi-dimensional evaluation results for the Gemma 3 model family on MMLU College Computer Science.

|  | Gemma 3 1B | Gemma 3 4B | Gemma 3 12B | Gemma 3 27B |
|---|---|---|---|---|
| Overall | 33.0 | 41.0 | 50.0 | 58.0 |
| Advanced Real Analysis (19.0%) | 57.9 | 52.6 | 47.4 | 52.6 |
| Abstract Algebra (11.0%) | 27.3 | 72.7 | 63.6 | 54.5 |
| Probability (7.0%) | 14.3 | 57.1 | 42.9 | 57.1 |
| Properties of Mathematical Operations and Functions (5.0%) | 60.0 | 40.0 | 40.0 | 80.0 |
| Advanced Mathematical Concepts and Applications (16.0%) | 25.0 | 31.3 | 56.3 | 68.8 |
| Mathematical Modeling and Algorithms (10.0%) | 20.0 | 30.0 | 50.0 | 60.0 |
| Multivariable Calculus (27.0%) | 29.6 | 25.9 | 40.7 | 55.6 |
| Mathematical Optimization Methods (5.0%) | 20.0 | 40.0 | 80.0 | 40.0 |

Table 23: Multi-dimensional evaluation results for the Gemma 3 model family on MMLU College Mathematics.

|  | Gemma 3 1B | Gemma 3 4B | Gemma 3 12B | Gemma 3 27B |
|---|---|---|---|---|
| Overall | 20.6 | 41.2 | 52.9 | 63.7 |
| Quantum Mechanics Principles and Applications (22.5%) | 8.7 | 39.1 | 43.5 | 73.9 |
| Thermodynamics (8.8%) | 11.1 | 22.2 | 55.6 | 55.6 |
| Special Relativity Concepts (13.7%) | 28.6 | 35.7 | 42.9 | 50.0 |
| Classical Physics Principles and Relationships (20.6%) | 42.9 | 28.6 | 57.1 | 66.7 |
| Physics Phenomena and Applications (16.7%) | 5.9 | 41.2 | 29.4 | 41.2 |
| Electromagnetism (5.9%) | 0.0 | 66.7 | 83.3 | 50.0 |
| Solid State Physics Concepts (11.8%) | 33.3 | 75.0 | 91.7 | 100.0 |

Table 24: Multi-dimensional evaluation results for the Gemma 3 model family on MMLU College Physics.

|  | Gemma 3 1B | Gemma 3 4B | Gemma 3 12B | Gemma 3 27B |
|---|---|---|---|---|
| Overall | 26.0 | 75.5 | 88.2 | 91.2 |
| US Sociopolitical Ideologies, Movements, and Issues (46.1%) | 26.6 | 80.9 | 87.2 | 92.6 |
| Progressive Era Economic and Social Initiatives (3.9%) | 25.0 | 100.0 | 100.0 | 87.5 |
| United States Governance and Politics (37.3%) | 26.3 | 67.1 | 89.5 | 90.8 |
| Ideological and Territorial Expansion in the Americas (9.8%) | 20.0 | 75.0 | 85.0 | 90.0 |
| American History Eras (2.9%) | 33.3 | 66.7 | 83.3 | 83.3 |

Table 25: Multi-dimensional evaluation results for the Gemma 3 model family on MMLU High School U.S. History.

| ID | Depth | Topic | Questions |
|---|---|---|---|
| 2139 | 5 | Reproductive Systems | 1) Which is the greatest benefit of sexual reproduction?
2) Which of these is the best example of sexual reproduction?
3) Which is an example of asexual reproduction?
4) Which statement describes a characteristic of both sexual and asexual reproduction? |
| 1244 | 5 | Photosynthesis | 1) In what form do plants store the energy produced from sunlight?
2) Which of the following best describes how plants use the energy they receive from sunlight?
3) Which of the following processes makes it possible for plants to use energy from sunlight to produce their own food?
4) Which form of energy do plants need to capture in order to perform photosynthesis?
5) What is the green pigment that allows plants to change the Sun's energy into chemical energy?
6) Which of the following do plants need to make their own food? |
| 163 | 3 | Waves | 1) What causes sound?
2) A substance that carries sound waves is called
3) An echo is a sound wave that has been
4) Sound travels as a |
| 131 | 2 | Landform Processes | 1) A beach is formed when sediment is deposited along a shoreline. What would most likely happen if rivers that empty into the ocean were dammed?
2) As a river enters a larger body of water, sediments are deposited over a wide area. Which of these landforms is likely to be formed at the site of deposition?
3) Deposition of sediment will most likely form a
4) Which statement describes the formation of a delta?
5) A delta at the mouth of a river is the direct result of
6) Rivers leave behind small pieces of rocks after they flood. These small pieces of rock form a floodplain. Which words best describe a floodplain?
7) Rivers and streams can carry sediments long distances before they are deposited. What is formed when sediments are deposited at the mouth of a river?
8) Sediment that is deposited on a beach may come from a local source or be transported by which action?
9) The shape of a riverbed changes over time as a result of which gradual process? |
| 2206 | 3 | Laboratory Equipment | 1) Which tool is best used to observe a soil sample?
2) Which tool would be most useful for observing the details of an insect's wings?
3) Which tools are used to determine the boiling point of water?
4) Which device is used to determine the volume of a liquid?
5) Which tool should be used to measure the stem length of a plant?
6) Which tool is used to observe the cell wall of a leaf?
7) Which tool is best to use when comparing an animal cell to a plant cell?
8) Which object is best seen with a microscope? |

Table 26: Example ARC-Easy clusters with their associated ID, depth, and topic.

| ID | Depth | Topic | Questions |
|----|-------|-------|-----------|
| 2818 | 2 | Locations | 1) is st augustine the oldest city in florida
2) is san juan puerto rico in the caribbean
3) is st augustine florida the oldest city in america
4) is there an amtrak station in pensacola florida
5) is key west part of the united states
6) is riviera maya on the gulf of mexico
7) is panama city beach on the gulf of mexico |
| 3026 | 2 | Species Hybridization | 1) is artificial selection and selective breeding the same thing
2) can a grizzly bear mate with a polar bear
3) can a horse and a donkey have a baby
4) can a polar bear and a grizzly mate
5) is it possible for a zebra and a horse to interbreed |
| 1751 | 3 | Sports | 1) did kaka play in the 2002 world cup
2) is the uefa champions league final one game
3) did alphonse areola play in world cup 2018
4) has a keeper ever won the ballon d'or
5) has christiano ronaldo ever won the world cup |
| 1149 | 3 | Breaking Bad | 1) will there be a season six of breaking bad
2) is better call saul set after breaking bad
3) was better call saul filmed before breaking bad
4) is there a spin off from breaking bad |
| 497 | 2 | Constitutional Rights | 1) is the first amendment in the bill of rights
2) was the right to bear arms in the original constitution
3) does the us constitution protect the right to privacy
4) do corporations have the same free speech rights as persons
5) are fighting words protected under freedom of speech
6) does the first amendment separate church and state
7) does the right to privacy exist in the constitution
8) is there a limit on freedom of speech
9) is the second amendment part of the constitution |

Table 27: Example BoolQ clusters with their associated ID, depth, and topic.

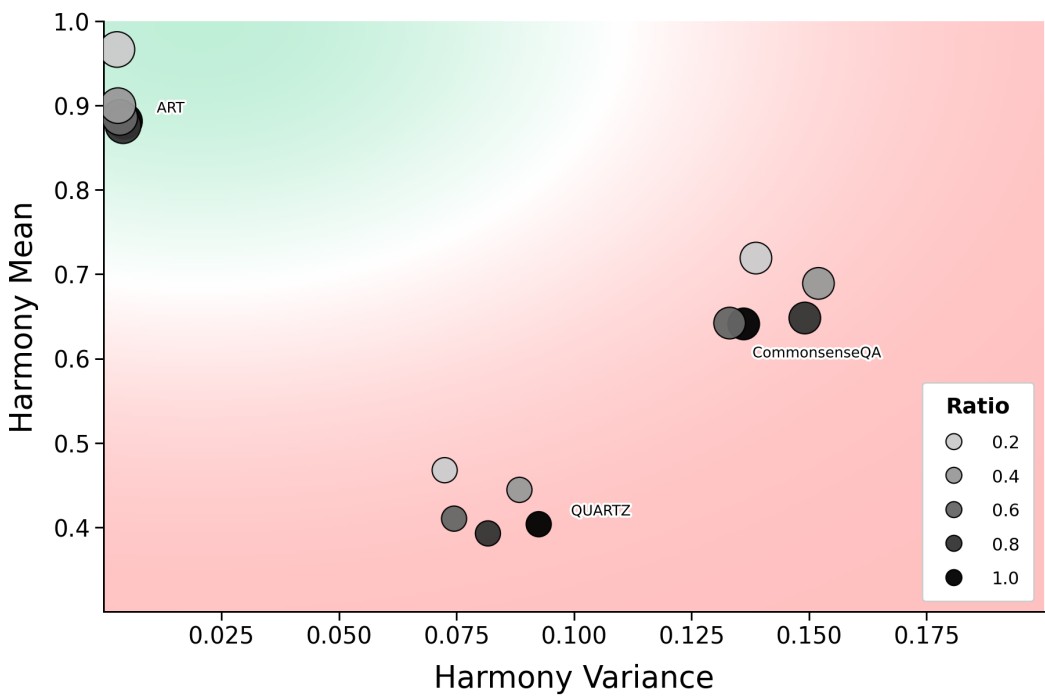

Figure 19: Effect of benchmark size on Harmony. For ART, QUARTZ, and CommonsenseQA, we down-sample at several ratios while preserving subdomain proportions and recompute Harmony across models. The resulting trajectories show only minor fluctuations in both mean and variance, and the benchmarks remain in the same qualitative regions of the mean-variance plane, indicating that benchmark size alone does not drive our mean-variance characterization.

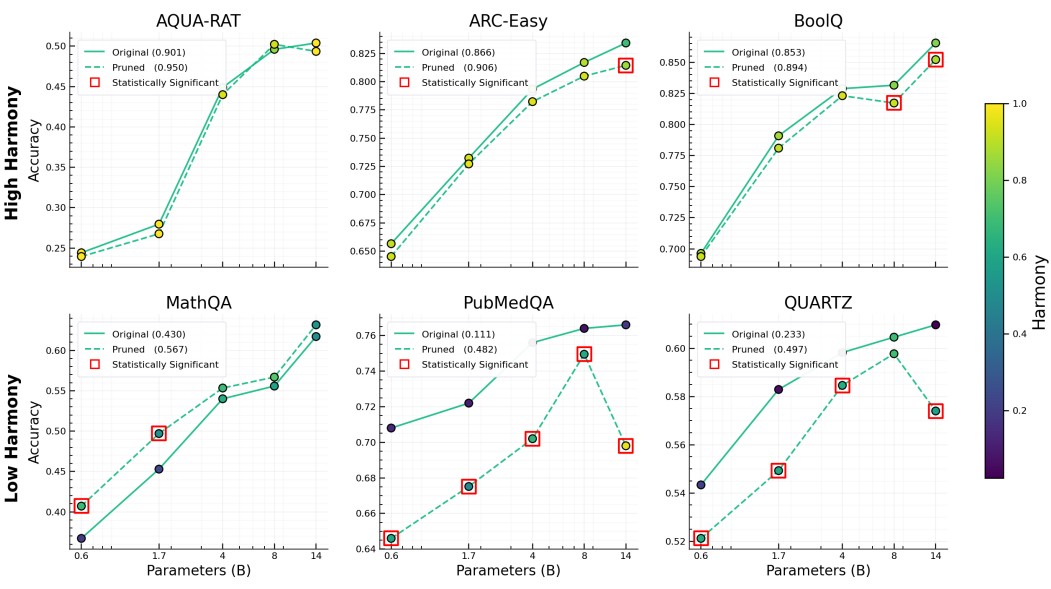

Figure 20: Pruning experiments replicated with pruning ratio $p = 0.1$ for all benchmarks.

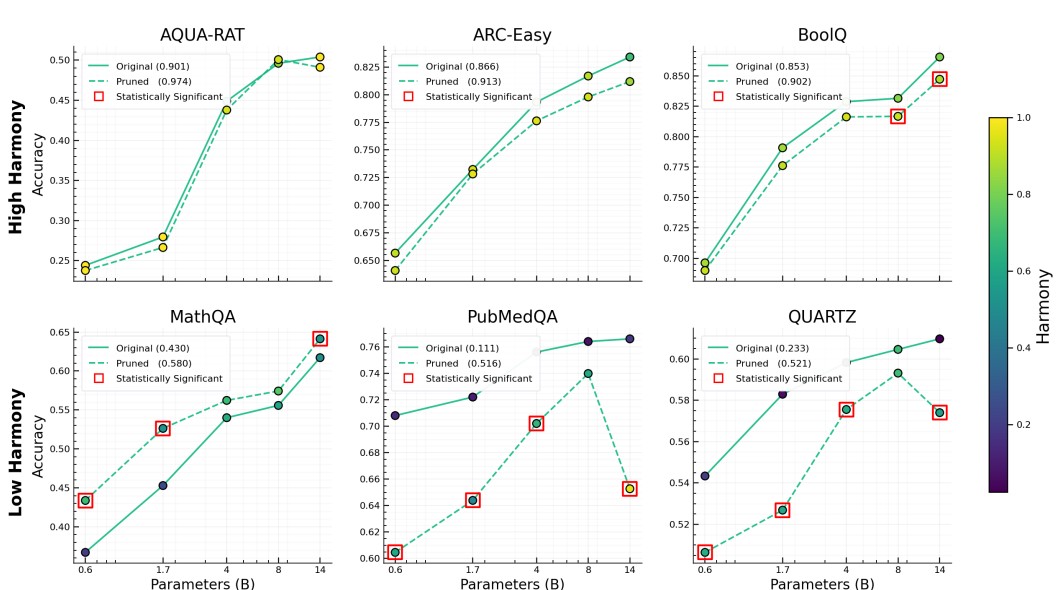

Figure 21: Pruning experiments replicated with pruning ratio $p = 0.2$ for all benchmarks.

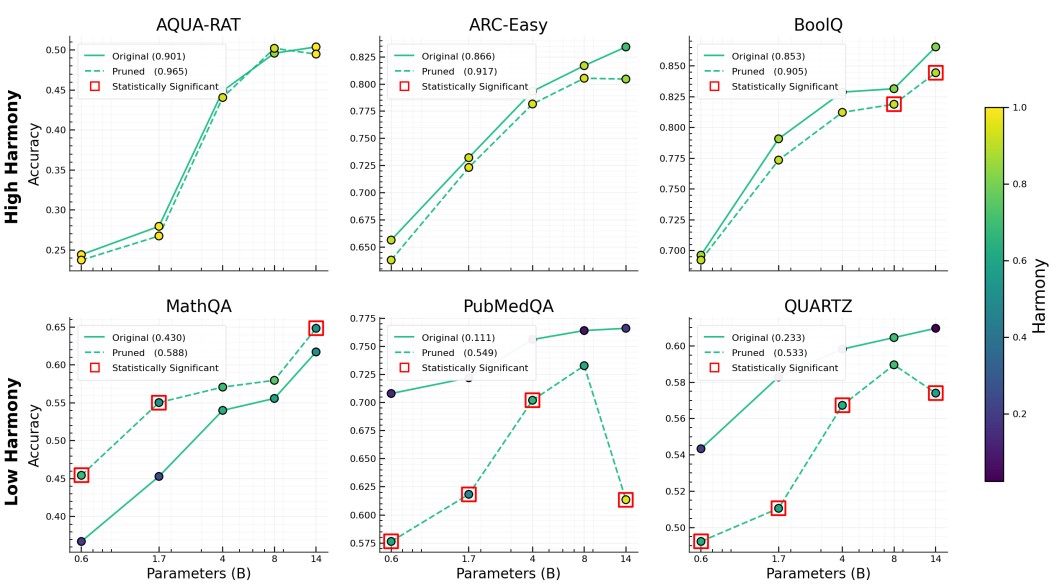

Figure 22: Pruning experiments replicated with pruning ratio $p = 0.3$ for all benchmarks.

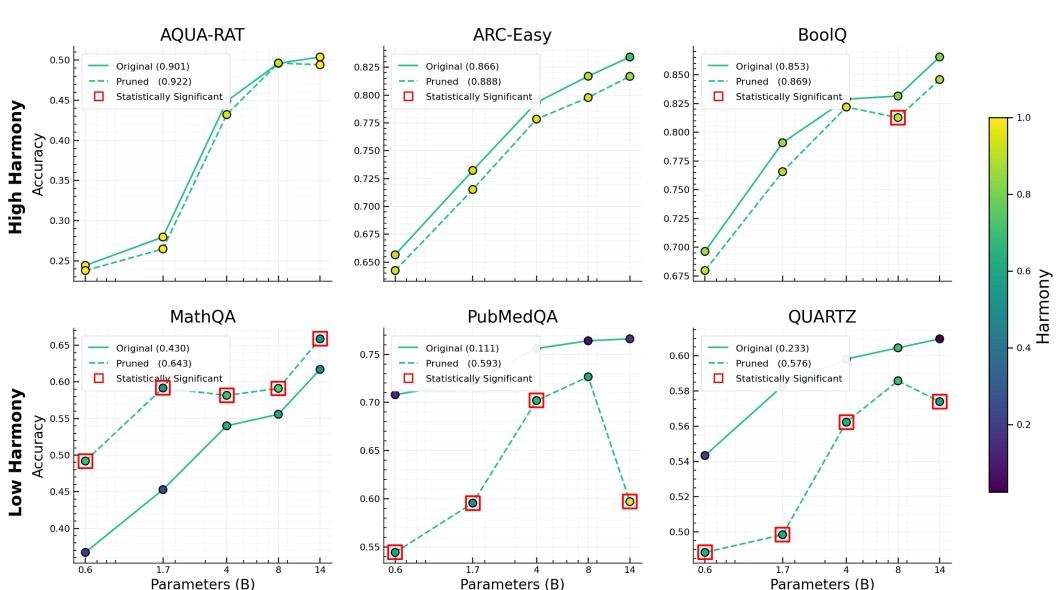

Figure 23: Pruning experiments replicated with pruning ratio $p = 0.4$ for all benchmarks.

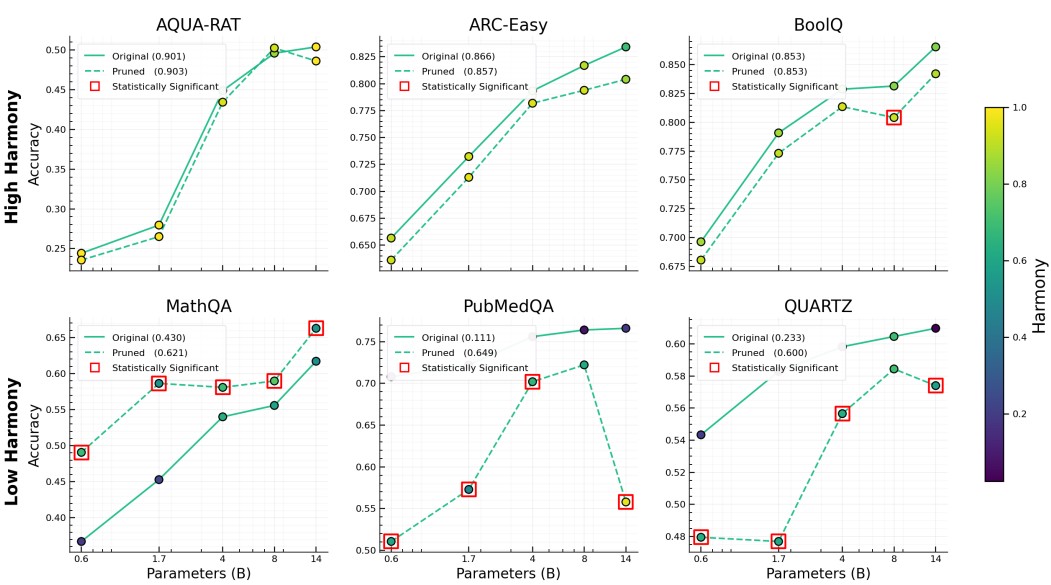

Figure 24: Pruning experiments replicated with pruning ratio $p = 0.5$ for all benchmarks.

