# OpenReview forum: "The Flaw of Averages: Quantifying Uniformity of Performance on Benchmarks"
_ICLR.cc/2026/Conference — Submitted to ICLR 2026_

### Official Review · Reviewer_85MX · 2025-10-23

**Soundness:** 2
**Presentation:** 3
**Contribution:** 2
**Rating:** 4
**Confidence:** 3

**Summary:**

This paper suggest benchmark harmony as a new metric to complement performance metrics like accuracy. A model's harmony score on a benchmark is a measure of variance between the performance on different (semantically meaningful) subsets of the benchmark. For a given benchmark, the authors consider the mean and variance of harmony accross models. They argue that high mean harmony and low harmony variance indicate more reliable evaluation.

**Strengths:**

- Considering attributes of benchmarks beyond accuracy seems generally useful.
- The proposed metric seems reasonably informative and relatively lightweight to compute
- The writing is clear and easy to follow.

**Weaknesses:**

- Some of the formulas used in the paper have large number of moving parts without strong apparent reason. It is not clear, whether results strongly depend on details in the used parameters, which would enable cherry-picking results.
    - For example, why not simply define harmony as the (weighted) variance of subset performances? Similarly, what is the intuition behind the formula used for pruning?
- It seems like harmony scores could become very misleading whenever the clustering picks up on aspects of question difficulty. As an extreme example, if a valid benchmark was partioned into its easier and harder half, harmony would depend on model quality in a U-shape, potentially yielding both low mean harmony and decently high harmony variance.
- While Figure 3 shows that ground-truth harmony and harmony based on the proposed clustering correlate on two datasets, it is unclear whether this remains true accross benchmarks. In addition, the regression coefficient is decently off from one in one of the two examples, suggesting that harmony measured by the proposed method is not really comparable between benchmarks.
    - More broadly, the motivation for the specific validation setup (for example, only 4 MMLU subdomains) is a bit unclear.
- Overall, it remains a bit unclear what reporting harmony offers compared to directly reporting subdomain scores, especially when there are only a few subdomains (as in the examples from Figure 3)

**Questions:**

- Figure 5: If I understand correctly, you prune low-harmony benchmarks more aggressively. How does the experiment ensure that the larger differences in accuracy for low-harmony benchmarks are not simply explained by pruning more data points?

---

> ### Author Response · Authors · 2025-11-17
> **Response to Reviewer 85MX: Part 1**
>
> We thank the reviewer for their thoughtful review and greatly appreciate that the reviewer finds our research focus useful, our proposed metric informative, and our writing clear. Below, we provide detailed responses to your specific concerns. We have also updated the manuscript with changes highlighted in brown color, which we reference throughout this response.
>
> ## Design Choices and Formulations (W1)
> **TL;DR: Our formulation of Harmony is chosen to satisfy clear criteria (coverage and uniformity, normalization, and interpretability) that simple variance-based alternatives do not meet.**
>
> We definitely agree that our formulations should be well-justified. In the updated manuscript, we provide justification for the design choices and formulations pointed out by the reviewer (in App. C) and discuss them below:
>
> Our definition of Harmony is guided by *three* criteria: (i) it should jointly capture **coverage and uniformity** of model performance, (ii) it should be a **bounded quantity** (e.g., [0,1]) making benchmarks with different numbers of subdomains *comparable*, and (iii) it should serve as an **interpretable reliability index**, where high Harmony means that most dataset mass lies on subdomains whose performance is close to the benchmark mean, and low Harmony means that performance is concentrated on certain subdomains. We therefore adopt normalized Shannon entropy as the *canonical measure of uncertainty (hence uniformity)* that **satisfies these criteria**. In contrast, (weighted) variance captures dispersion but *neither* provides a normalized [0,1] scale *nor* supports interpreting Harmony as the effective ratio of subdomains with performance close to the benchmark mean.
>
> Our pruning rate is similarly designed to satisfy three simple constraints: (i) it should **increase monotonically as Harmony decreases** (less harmonious benchmarks are pruned more), (ii) it should be *bounded away from 0 and 1* so every benchmark is pruned to a **comparable and non-degenerate** extent, and (iii) it should capture the **severity** of low Harmony. To achieve this, we map Harmony to a pruning rate via a *smooth, convex* schedule ranging from 5% to 50% that saturates for very low Harmony, as in Eq. (7). The exponent 1.5 makes the schedule slightly convex, so pruning grows slowly as Harmony decreases from 1 but more steeply when Harmony is very low, and clipping at H=0.1 avoids excessively large rates for extremely disharmonious benchmarks.
>
> ## Utility of Harmony for Difficulty Based Clustering (W2)
> **TL;DR: Harmony is defined with respect to semantic subdomains, not difficulty bins. When a semantic subdomain happens to be substantially easier or harder, low Harmony would correctly flag this as a genuine reliability issue rather than an artifact of our method.**
>
> We appreciate the reviewer’s concern that Harmony could, in principle, be misleading if the partitioning were to group examples primarily by difficulty (e.g., *easy* half vs. *hard* half). However, the setup we study **does not treat such difficulty splits as valid subdomains**. In Sec. 2.1-2.3, we define subdomains as **semantic subsets** of the stated benchmark domain and require a semantic partition of the benchmark guided by a similarity function that measures semantic similarity between datapoints. Our predictive similarity (Sec. 2.3, App. B) is *specifically* introduced to satisfy this requirement.
>
> Concretely, we (i) induce partitions from predictive similarity and spectral clustering, and (ii) empirically **validate** that these partitions **recover known semantic domains, not arbitrary difficulty bins**. On RedundantQA, MMLU, GPQA, and M3Exam, we show that predictive similarity achieves the **highest agreement with ground truth domains** among *a diverse range of baselines* (App. B.3), and we demonstrate that the resulting Harmony estimates track ground truth distributional imbalance in controlled variants (Sec 2.4 and App. B.4). This evidence suggests that **our clusters correspond to interpretable semantic subdomains**, rather than easy or hard subdomains of the benchmark.
>
> In the reviewer’s hypothetical case where a benchmark is partitioned according to difficulty levels, we agree that Harmony would be dominated by difficulty effects and could display non-monotonic trends (e.g., U-shaped). However, such a partition is *outside* of our intended usage, since it does not correspond to semantically meaningful subdomains of the benchmark domain. If a semantic partition nonetheless aligns with systematic difficulty differences (e.g., some subdomains are much easier), then low Harmony is **desirable**, because it indicates that the aggregate score is *disproportionately influenced* by such subdomains in exactly the sense we formalize in Sec. 2.2.

---

> ### Author Response · Authors · 2025-11-17
> **Response to Reviewer 85MX: Part 2**
>
> ## Generalization to New Benchmarks and Correlation Between Ground-Truth and Proposed Harmony (W3)
> **TL;DR: We extend our validation setup to GPQA and M3Exam benchmarks. Across all four benchmarks (MMLU, RedundantQA, GPQA, M3Exam), Harmony computed from predictive similarity clusters closely tracks Harmony computed from the ground truth domain partitions.**
>
> Thank you for raising this. Figure 2 (previously 3) in the main text visualizes *two ends* of the Harmony spectrum for illustration. The full validation over all variants is reported in Figures 7 and 8, where our clustering method with predictive similarity achieves Pearson $r=0.865$ and Spearman $\rho=0.949$, indicating a **strong linear relationship between ground-truth Harmony and our estimated Harmony**. We have updated the caption of Fig. 2 in the revision to explicitly refer to full plots.
>
> Regarding the limited selection of subdomains in the validation setup (e.g., the use of 4 MMLU subdomains), our goal was to *span a wide range of dominant ratios* while keeping the number of synthetic variants (and thus the figures) **interpretable**. Adding more subdomains would substantially increase the number of variants and clutter the visualization, without qualitatively changing the conclusions. We have clarified this design choice in the revised manuscript (Sec. 2.4).
>
> We extend our validation beyond MMLU and RedundantQA to M3Exam [2] and GPQA [3] in the updated App. B.3 and B.4 to further assess generalization across benchmarks with ground-truth domain partitions. Our findings show that predictive similarity **remains the best alternative among several baselines**.
>
> ## Value of Harmony Beyond Subdomain Scores (W4)
>
> **TL;DR: Harmony is not intended to replace per-subdomain reporting. Rather, it provides a reliability index that makes it feasible to compare and analyze benchmarks and models at scale.**
>
> We certainly agree that per-subdomain scores are *highly informative*, and we do **not** intend Harmony to replace them. In our experiments, we continue to report the underlying subdomain performances when appropriate (e.g., the App. I and App. J). Therefore, we view Harmony as a **complementary summary statistic** rather than a *substitute*.
>
> The main benefit of Harmony over directly inspecting subdomain scores is that it provides a **single, comparable index of reliability** that we can use to analyze many benchmarks and models jointly. For an individual benchmark with only a few subdomains, one can inspect the subdomain performances directly. However, once we consider dozens of benchmarks and models, it becomes *impractical* to manually compare entire vectors of subdomain scores. Harmony gives us a fixed-scale quantity that already accounts for both coverage and uniformity, which we can then correlate with other phenomena (e.g., fragility) and use to rank or group benchmarks in a consistent way across large scale analyses.
>
> ## Varying Pruning Ratios Across Harmony Levels (Q1)
>
> **TL;DR: Although low Harmony benchmarks are pruned more aggressively, our conclusions are based on statistical significance tests that account for dataset size.**
>
> Thank you for pointing this out. By design, we prune low Harmony benchmarks more aggressively, because pruning only a small number of data points would have a *negligible impact* on their Harmony scores. To ensure comparability across benchmarks with different pruning ratios, we evaluate for each benchmark whether the accuracy difference before and after pruning is statistically significant, using **tests that explicitly account for the underlying sample size** (as detailed in the App. G). Our findings show that low Harmony benchmarks *consistently exhibit statistically significant changes* in performance, whereas high Harmony benchmarks *rarely* do so. This suggests that low Harmony benchmarks are *genuinely fragile*, indicating the effect is **not merely due to pruning more data points**.
>
> Thank you again for the constructive review. **We are happy to address any remaining questions, and we hope that our clarifications and revisions merit a higher rating.**

---

> > ### Comment · Reviewer_85MX · 2025-11-21
> >
> > Thank you for the reply. Some follow-up questions/remarks:
> >
> > - I am not sure I understand your point regarding the formulas: As far as I understand, variance can easily be normalized on bounded domains, and I am not sure I understand why the specific interpretation as "effective ratio of subdomains with performance close to the benchmark mean" is more intuitive/useful than any of the many interpretations of variance.
> > - Also, regarding the formulas: They seem to have several hyperparameters (exponent, clipping, bandwidth, etc). Do the specific values of these hyperparameters (within reasonable ranges) affect your conclusions in any way?
> > - Regarding W2, I am not convinced that semantic domains (either ground truth, or clustered) can consistently be separated from difficulty as cleanly as the authors seem to envision.
> > - Regarding W3, I would like to clarify my second point: Let's say I would buy that ground truth harmony is a useful measure, and would like to make a decision based on a harmony threshold of 0.8. For this, I do not only need my estimate to correlate well with harmony, I need to know the full regression equation; but that equation does not appear to be particularly stable in Figures 2 and 3.
> > - "we prune low Harmony benchmarks more aggressively, because pruning only a small number of data points would have a negligible impact on their Harmony scores." But why not simply prune the high-harmony benchmarks at the same ratio? It seems like that would a) make the reader's job easier because they don't need to check the validity of your statistical correction and b) reduce the risk of Figure 4 being misleading (as it presumably does not employ any correction for the amount of pruning?)

---

> > > ### Author Response · Authors · 2025-11-22
> > > **Response to Reviewer 85MX: Part 1**
> > >
> > > Thank you for the thoughtful follow-up. We really appreciate the chance to clarify these points and improve our work.
> > >
> > > ## Comparison of Entropy and Variance
> > > Thank you for emphasizing this comparison. Harmony is intended to jointly capture **coverage and uniformity**. Entropy is suited for this, because it increases *only when* (a) mass is spread over **more subdomains** (coverage) and (b) that mass is spread **more evenly** rather than being dominated by a few subdomains (uniformity).
> > >
> > > In contrast, (normalized) variance measures *only the magnitude of dispersion*. As a result, different patterns of concentration can yield the **same variance**, making variance **less diagnostic** of the distribution of the performance. This renders the interpretations of variance *uninformative* for our purpose. In particular, variance **cannot distinguish** whether error is (a) driven by a small number of severely underperforming subdomains or (b) a modest spread across many subdomains.
> > > For example, with 6 equally weighted subdomains and a benchmark mean of 0.7, consider:
> > >
> > > Model A: [0.85, 0.55, 0.7, 0.7, 0.7, 0.7]
> > >
> > > Model B: [0.7866, 0.7866, 0.7866, 0.6134, 0.6134, 0.6134].
> > >
> > > Both have **identical normalized variance** ($s^2 = 0.0075$), yet they represent **different reliability profiles**. Model A exhibits two *extreme* subdomains, while Model B shows *moderate* deviations across all subdomains, where each one remains within a reasonable range around the mean. Entropy-based Harmony favors Model B because it reflects better uniformity, which **variance fails to distinguish**.
> > >
> > > ## Controlling for Hyperparameters
> > > Thanks for pointing this out. Recall that our pruning ratio formula is:
> > >
> > > $ p = clip_{[p_{\min},p_{\max}]}(p_{\min} + (p_{\max}-p_{\min}) (\frac{1-\mathrm{clip}(H;h_0,1)}{1-0.1})^{\gamma}) $.
> > >
> > > To test sensitivity, we sweep one hyperparameter at a time (others fixed) and compare to Fig. 4:
> > >
> > > Exponent hyperparameter $\gamma \in $ {$1.0, 1.25, 1.5, 2$}:
> > > - $\gamma=1.0$ $\rightarrow$ Same results for high Harmony benchmarks, 4B, 8B models have insignificant post-pruning accuracy changes in MathQA and 8B has insignificant post-pruning accuracy change in PubmedQA.
> > > - $\gamma=1.25$ $\rightarrow$ Same results for high Harmony benchmarks, 8B has insignificant post-pruning accuracy change in PubmedQA.
> > > - $\gamma=1.5$ $\rightarrow$ Same results for both high and low Harmony benchmarks.
> > > - $\gamma=2.0$ $\rightarrow$ Same results for high Harmony benchmarks, 8B has insignificant post-pruning accuracy change in PubmedQA.
> > >
> > > Harmony clipping floor hyperparameter $h_0 \in $ {$0.05, 0.1, 0.2$}:
> > > - $h_0=0.05 \rightarrow$ Same results for both high and low Harmony benchmarks.
> > > - $h_0=0.1 \rightarrow$ Same results for both high and low Harmony benchmarks.
> > > - $h_0=0.2 \rightarrow$ Same results for both high and low Harmony benchmarks.
> > >
> > > Minimum pruning ratio hyperparameter $p_{min} \in $ {$0.02, 0.05, 0.1$}:
> > > - $p_{min}=0.02 \rightarrow$ Same results for both high and low Harmony benchmarks.
> > > - $p_{min}=0.05 \rightarrow$ Same results for both high and low Harmony benchmarks.
> > > - $p_{min}=0.1 \rightarrow$ Same results for both high and low Harmony benchmarks.
> > >
> > > Maximum pruning ratio hyperparameter $p_{max} \in $ {$0.4, 0.5, 0.6$}:
> > > - $p_{max}=0.4 \rightarrow$ Same results for high Harmony benchmarks, 8B has insignificant post-pruning accuracy change in PubmedQA.
> > > - $p_{max}=0.5 \rightarrow$ Same results for both high and low Harmony benchmarks.
> > > - $p_{max}=0.6 \rightarrow$ Same results for high Harmony benchmarks, 8B has insignificant post-pruning accuracy change in PubmedQA.
> > >
> > > Across the 30 model benchmark pairs in Fig. 4, qualitative outcomes change in **at most three cases and most often in none**. Therefore our main finding (i.e., low Harmony benchmarks consistently exhibit significant accuracy shifts under pruning, while high Harmony benchmarks do not) **is stable cross reasonable pruning hyperparameter choices.**
> > >
> > > ## Using Same Pruning Ratio for High and Low Harmony Benchmarks
> > > Thanks for raising this. To directly rule out pruning ratio as a confound, we add new experiments (App. P), in which we prune benchmarks from Figure 4 at five fixed ratios $p \in $ {$0.1, 0.2, 0.3, 0.4, 0.5$}. Across all ratios, we observe **the same qualitative results**:
> > > - Low Harmony benchmarks *consistently show statistically significant* accuracy shifts,
> > > - High Harmony benchmarks *rarely do so*, even at **higher** pruning rates.
> > >
> > > Moreover, the accuracy changes due to pruning tend to saturate, as beyond moderate pruning (around $p=0.3$), further increases in the pruning ratio produce *marginal shifts*. While post-pruning accuracies naturally vary with how much data is removed, the central conclusions are **invariant to pruning budget**, indicating that Figure 4’s results are **not** explained by the pruning ratio alone.

---

> > > ### Author Response · Authors · 2025-11-22
> > > **Response to Reviewer 85MX: Part 2**
> > >
> > > ## Utility of Harmony Interpretation
> > > The *effective ratio of subdomains with performance close to the benchmark mean* is useful because it *directly* answers an evaluation question practitioners care about: **on what fraction of the dataset can I trust this model to behave similarly?** This presents a **reliability statistic** that (i) is comparable across benchmarks, (ii) is easy to reason about (e.g., model behaves like the average on 80% of the dataset), and (iii) differentiates models that achieve the same mean by concentrating performance on a few subdomains vs. performing uniformly.
> > >
> > > ## Semantic Domains and Difficulty
> > > Thank you for this important nuance. We acknowledge that semantic subdomains and difficulty might not be perfectly separable. Crucially, our method does **not** require them to be. Harmony is defined with respect to semantic subdomains, and if those subdomains are inherently easier or harder, Harmony will reflect that imbalance.
> > >
> > > Concretely, consider the MENTAT dataset example raised by Reviewer F1zp. A model can obtain a high aggregate score across categories (e.g., diagnosis, treatment) even when that score is *disproportionately increased by an easier subdomain* (e.g., diagnosis). In this case, the benchmark-level metric **overstates real-world reliability**. Moreover, if performance is meaningfully lower on treatment decisions, where errors carry higher cost, the aggregate score would **hide a critical failure mode**. Harmony is designed to expose this *mismatch*. Low Harmony indicates that the overall score is not representative of performance across subdomains and therefore should not be used in isolation. Instead, it suggests inspection of subdomain-level results prior to deployment, **preventing a potential critical error**.
> > >
> > > More broadly, the reviewer’s concern *highlights* an intended use-case of Harmony: whenever semantic subdomains wield uneven influence over the overall score (whether due to inherent difficulty or imbalance) **the aggregate score becomes less trustworthy**, and *Harmony rings as a quantitative alarm bell*.
> > >
> > > ## Regression Equations
> > > Thanks for raising this. Figures 7-9 show that our estimated Harmony maintains a strong monotonic relationship with ground truth Harmony across all variants and benchmarks. Crucially, Harmony is introduced as a **relative reliability signal**: our intent is *not to dichotomize* benchmarks into universally “good” vs. “bad” categories, but *to enable principled one-to-one comparisons*. In this setting, the absolute scale is not required, since what matters is that **higher estimated Harmony consistently indicates higher ground-truth Harmony.**
> > >
> > > Relatedly, for decision rules that seek benchmark-agnostic absolute cutoffs (e.g., Harmony $\geq 0.8$), we recommend using *calibration invariant criteria such as rank or percentile Harmony.* This follows directly from the fact that different benchmarks can induce benign shifts in slope/intercept due to differing domain granularity and dominant-ratio distributions, even when the estimator tracks the ordering *extremely well*.
> > >
> > > Thank you again for the insightful follow-up, which we greatly appreciate. **We are happy to address any remaining concerns and hope that our clarifications warrant a higher rating.**

---

> > > > ### Comment · Reviewer_85MX · 2025-11-23
> > > >
> > > > Thank you for the reply.
> > > >
> > > > - Regarding the variance vs entropy point, are there not also pairs of distributions with the same entropy but different variances? I think I fail to see your intuition for why distribution B should be preferable. Taking a step back, I guess the choice between the two measures is rather arbitrary/dependent on how specific scenarios are weighted; Given that, my confusion mostly is about the choice of the (from my perspective) much more complicated entropy measure (that first needs to transform performance metrics into a "probability distribution" over tasks) rather than simple variance.
> > > >
> > > > - Reading the relevant parts to understand your interpretation in terms of subdomains again another confusion came up: Even if  performance was constant across subdomains, wouldn't your current definition of harmony strongly depend on the relative sizes of the different subdomains, due to the weighting turnining the p_i non-uniform?
> > > >
> > > > - Regarding the regression equations: You say that "higher estimated Harmony consistently indicates higher ground-truth Harmony". But that only seems to be true within the setting of either of the two experiments, not if you combine the data from both experiments. As such, the scope of signal reliability appears to be somewhat restricted?
> > > >
> > > > - Rereading parts of the paper I also stumbled across the following sentence in the abstract: " For instance, ARC-Easy is overwhelmed by questions on Biological Concepts, overshadowing other critical subdomains such as Geography, Physics, Chemistry, and Environmental Science". Where can I find more details to support this claim in the paper? The only relevant numbers I could find are in table 10, and they do not seem congruent with the statement from the abstract.
> > > >
> > > > [I will get back at you regarding the point on semantic domains and difficulty at a later point]

---

> ### Author Response · Authors · 2025-11-24
> **Response to Reviewer 85MX**
>
> We thank the reviewer for their continued engagement and thoughtful feedback. We are grateful for the chance to clarify and strengthen our work.
>
> ## Variance vs. Entropy
> Entropy captures **skewness/uniformity** of benchmark mass across subdomains, whereas variance measures **the average squared distance from each subdomain to the benchmark mean**. Our goal is not to measure dispersion, but to quantify (i) how much benchmark mass lies on subdomains whose performance is near the mean, and (ii) whether this near-mean mass is evenly distributed across subdomains or concentrated in a few. Entropy computed over the near-mean subdomain mass directly reflects both of these notions. Hence, we use entropy rather than variance.
>
> For example, with $n=10$ subdomains, a 0.1 variance increase could arise *either* from five subdomains each increasing their squared deviation from the mean by 0.2, or from a single subdomain increasing its squared deviation by 1.0. **Variance treats the two situations as equivalent, while entropy reveals the greater skewness in the latter.**.
>
> Regarding Model A vs. B, Model B is preferable under our reliability notion because its deviations from the mean are distributed across all subdomains instead of being concentrated in a few. In Model A, two subdomains are outliers, for which the aggregate score fails to be representative. In Model B, all subdomains remain within a moderate range around the mean, so the aggregate is representative of performance across the benchmark.
>
> ## Case of Constant Accuracy Across Subdomains
> The reviewer’s observation is correct. When a model’s performance is identical across subdomains, Harmony is reduced to the entropy of subdomain weights. However, **it is our intent that Harmony is aware of subdomain sizes**.
>
> Harmony is explicitly intended to be a joint reliability index over (i) coverage of subdomains and (ii) uniformity of performance across them. When performance is perfectly uniform, *the uniformity component becomes uninformative*. Thus, Harmony naturally collapses to a coverage signal and penalizes benchmarks whose mass is concentrated in a few subdomains. We took this design choice because even with uniform performance, skewed subdomain mass can make aggregates **unreliable** by *letting dominant subdomains drive the score while others remain under-evaluated*.
>
> If one wanted a uniformity-only statistic, one could remove size weighting. We deliberately do not, because **benchmark reliability metric should penalize both uneven representation and uneven performance**.
>
> ## Regression Equations
> Thank you for flagging the distinction between within-benchmark vs. across-benchmark reliability.
>
> To test the reviewer’s specific concern, we now additionally collapse all points from Figures 7, 8, and 9 into a single pool and recompute correlations (App B.4). In this setting, **correlation remains strong with Pearson $r = 0.869$ and Spearman $\rho = 0.879$** (Fig. 10). This shows that the estimator **preserves ordering** within each benchmark and across benchmarks.
>
> Therefore, the scope of the reliability signal **is not restricted** to a single experimental setting. Estimated Harmony **remains a reliable proxy** for ground-truth Harmony both **within and across** benchmarks, supporting its intended use as a benchmark-agnostic reliability index.
>
> ## ARC-Easy Example
> Thanks for flagging this phrasing. The quantitative evidence appears in the ARC-Easy subdomain breakdown (Table 10). Biological Concepts constitutes a substantial portion of ARC-Easy (24.1%), whereas other science subdomains are significantly smaller (e.g., Physics (9.4%), Environmental Science (6.5%), Astronomy (7.6%),  Chemistry (4.8%)). This imbalance is what the abstract intended to highlight.
>
> We agree that the term *overwhelmed* may suggest Biology forms a majority of the benchmark. Our intended meaning was that Biology is **several times larger** than many other scientific subdomains and **disproportionately influences** the aggregate score. We have revised the abstract to better align with the breakdown in Table 10.
>
> Thank you again for the thoughtful follow-up. **We hope our clarifications and revisions resolve the remaining concerns and warrant a higher rating.**

---

> > ### Comment · Reviewer_85MX · 2025-11-25
> >
> > Thank you for the additional discussion.
> >
> > I understand your technical point regarding entropy vs variance, but remain unconvinced about the motivation (I do not see why "how much benchmark mass lies on subdomains whose performance is near the mean" would be a more useful metric than dispersion between scores on different subdomains).
> >
> > Regarding the second point, I am not really convinced that uneven subdomain sizes are generally indicative of a lack of benchmark reliability, especially when performance is constant across these subdomains. In particular, different subdomains can naturally be less important/less common in many realistic settings, in which case even representation would arguably hurt more than it helps. Overall, it seems like separating concerns of subset size from between-subset variation would generally be more informative.
> >
> > Thank you for running the additional analysis. Looking at either of the figures, while I agree that there is clear linear signal, it seems like "preserves ordering" is too strong of a statement, given the copious amounts of noise.
> >
> > Overall, I am not convinced of the core proposed method, so I will keep my score.

---

> > > ### Author Response · Authors · 2025-11-26
> > >
> > > Thank you again for the very thoughtful and sustained engagement with our work. We wanted to briefly summarize what we believe we have addressed so far, and then ask for your guidance on what we still can do to better respond to your remaining concerns.
> > >
> > > In the revision, we (i) ran extensive sensitivity analyses over pruning hyperparameters and found that our main conclusions are stable, (ii) added fixed-ratio pruning experiments to rule out pruning rate as a confound in the fragility results, (iii) extended our Harmony validation to additional benchmarks and to a combined analysis across all settings, (iv) clarified the validation design and cross-referenced the full regression plots, (v) revised the ARC-Easy abstract language to more accurately match the subdomain breakdown, and (vi) more clearly positioned Harmony as a complementary reliability index rather than a replacement for subdomain-level reporting.
> > >
> > > Our understanding is that your main remaining reservations are conceptual and are specifically about (a) the choice of an entropy-based notion of reliability over a variance-based one, and (b) our decision to couple subdomain size and between-subdomain variation in a single metric rather than separating these axes. If that is accurate, we would be very grateful for your advice on what additional changes would meaningfully address this. For example, would it help if we (1) added a parallel variance-based reliability measure and compared both side-by-side or (2) explicitly decomposed Harmony into “size” and “uniformity” components?
> > >
> > > We fully respect that you may ultimately remain unconvinced about the core design choice, but if there are specific analyses, ablations, or framing changes that you feel could substantially strengthen the paper, we would very much appreciate your guidance.

---

### Official Review · Reviewer_5RAY · 2025-10-23

**Soundness:** 1
**Presentation:** 3
**Contribution:** 3
**Rating:** 2
**Confidence:** 4

**Summary:**

The authors study benchmark reliability from a distributional perspective and introduce benchmark HARMONY, which measures how uniformly a model’s performance is distributed across the subdomains of a benchmark. The authors, however, represent HARMONY as essentially a measure of benchmark quality, positing that for a given model set, HARMONY is an independently informative measure, tracking uniform competence across discovered subdomains.

**Strengths:**

* The topic the authors select for study, measures of benchmark reliability, is timely, and more research in this area is definitely needed.
* The authors have provided a pretty extensive set of experimental results; I particularly appreciate their inclusion of the fully open Olmo2 models and their relatively large benchmark selection.
* Although their focus in the main paper on MCQA comparisons may be problematic in the specifics, in general, I think that focus improves the rigor of the paper and the trustworthiness of the measure.
* The description of the measure is quite thorough and detailed, which helps the reader understand the core contributions easily.
* The controlled pruning experiments demonstrate that low-harmony benchmarks are fragile.

**Weaknesses:**

* The paper's core methodology relies on 'predictive similarity' - clustering questions by the similarity of models' output probability distributions. However,
in MCQA settings, these distributions are heavily concentrated on answer tokens (A/B/C/D or short answer phrases). It is not clear to me that the spectral clustering approach described by the authors would lead to meaningful or interpretable clusters in this setting. The conditional output probabilities generated by models would be (A) different for every model and (B) heavily concentrated on a small set of tokens.
* The authors' mean-variance plane interpretation of benchmark reliability does not control for benchmark size; this has the potential to be a significant confound, as larger benchmarks can be expected to exhibit more stable variance characteristics. I am not convinced from the current analysis that HARMONY measure they would be independently informative when comparing benchmarks which are identical in size.
* I have a doubt about the derivation of the clusters. In Section 2.3, we have -- "we sweep 2 ≤ k ≤ 20 and select the value maximizing the silhouette score". This means that k is data-driven and likely varies across benchmarks. Although the Harmony formula includes size weights: wi = |Ai|/|B|, larger benchmarks will tend to yield more stable cluster size estimates and lower sampling variance in wi. I cannot find in the paper a report on how many clusters each benchmark yielded, whether k correlated with benchmark size, whether k correlated with HARMONY.
* HARMONY, as a measure of a *benchmark's* reliability, is potentially confounded by the fact that benchmark HARMONY is aggregated over a model set. Not all models are intended to be generalists, and not all benchmarks are intended to measure generalist capability. Furthermore, there are practical concerns. The temptation on the part of researchers would be to compare HARMONY across studies, but this will almost never be possible, because new models will come out and the model set and the HARMONY scores will no longer be directly comparable. There is also a risk of researchers cherry picking model sets to make benchmarks appear more or less harmonious, depending on their goals.
* In Figure 2, the "Benchmark Size" field is not defined in the caption and is difficult to interpret from visual inspection, it's hard to figure out which benchmarks correspond to which sizes.

**Questions:**

* In the main plot in Figure 2, what is the strength of correlation between harmony mean and harmony variance? From visual inspection it looks like it would be a pretty strong negative correlation, but I'm curious to know the exact value.
* Why did the authors not conduct asystematic analysis of cluster characteristics, including an examination of whether cluster count or size distribution correlates with HARMONY?
* Can the authors show that HARMONY clusters correspond to semantic categories? Specifically: (1) qualitative examples showing cluster coherence, (2) comparison of discovered clusters against ground-truth domain labels where available (beyond the 2 validation cases), and (3) ablation showing that predictive similarity outperforms random clustering of equal size and number?

---

> ### Author Response · Authors · 2025-11-15
> **Response to Reviewer 5RAY: Part 1**
>
> We thank the reviewer for their constructive review and greatly appreciate that the reviewer finds our research focus timely, our experiments in-depth and rigorous, and our methodology thorough and detailed. Below, we address each of the reviewer’s points in detail. We have also updated the manuscript with changes highlighted in blue, which we reference throughout this response.
>
> ## Predictive Similarity and Clustering in MCQA Settings (W1)
>
> **TL;DR: In MCQA, we utilize distributions over answer phrases, so concentration is an informative signal rather than a limitation. Spectral clustering on predictive similarity yields the most accurate semantic partition among several alternatives and shows stability across models.**
>
> We thank the reviewer for raising this point. As noted in Sec. 2.3, we use distributions over *answer phrases* rather than option indices (A/B/C/D). **Concentrated probability mass on a small set of relevant tokens is desirable**, because predictive similarity measures similarity through where models place probability mass. Even when distributions are sharp, high probability phrases and how mass is shared across plausible answers provide a strong signal for comparing questions. App. B.6 further shows that truncating each distribution to its top-50 tokens leaves the predictive similarity neighborhoods almost unchanged, indicating that our metric is driven by the *concentrated head of the distribution, rather than tails.*
>
> Empirically, spectral clustering with predictive similarity achieves the **best recovery** of known semantic domains in MMLU, RedundantQA, GPQA, and M3Exam among 7 baseline metrics (App. B3), and it **outperforms** them in measuring semantic similarity (App. B2) and capturing Harmony (App. B4). These results indicate that the induced clusters are indeed *meaningful and align with human-labeled subject domains.*
>
> Regarding model dependence, one could compute Harmony using a single, model-agnostic partition, measuring uniformity relative to that fixed partition. We instead adopt model-aware partitions because different models draw different task-relevant boundaries in input space [4]. Hence, a partition that aligns with one model can be misaligned with another, averaging that model’s concentrated errors across subsets and making it appear *artificially harmonious*. By grouping data points that a model finds similar and measuring how evenly performance is distributed over those induced subdomains, Harmony becomes sensitive to model specific boundaries that a fixed partition might obscure. Importantly, our experiments in App. B.5 shows that predictive similarity partitions are consistent across models, so **model-dependent similarity does not undermine cross-model comparability**.
>
> ## Controlling for Benchmark Size (W2)
>
> **TL;DR: Empirical tests show that benchmark size does not confound our mean-variance characterization. Harmony is size-agnostic by definition.**
>
> Thank you for pointing out this potential confound. In the revised manuscript, we include a dedicated section (App. N) with two complementary analyses: (i) a correlation study showing **no statistically significant** linear relationship between benchmark size and either the mean or variance of Harmony across models, and (ii) a controlled subsampling experiment where we down-sample several benchmarks at different ratios while preserving the underlying distribution and find **only minor fluctuations** in Harmony. We also find that reducing the size of a benchmark by large factors **does not systematically increase** the variance of Harmony across models and preserves stable positions in the mean-variance plane. Together, these results support that benchmark size **does not explain** the structure observed in our mean-variance characterization. Hence, Harmony remains informative because it depends on how data points are distributed across semantic subdomains rather than benchmark size.
>
> ## Controlling for $k$ (W3)
>
> **TL;DR: $k$ shows no significant correlation with Harmony.**
>
> We thank the reviewer for raising this point. As described in Sec. 2.3, for each benchmark we sweep 2$\le k \le20$ and select the value maximizing the silhouette score (which is a well-established metric of clustering quality [1]). Thus, $k$ is determined in a *data-driven* way rather than hand-tuned to improve Harmony.
>
> In the revised manuscript, we include a dedicated section (App. O), where we find that $k$ is strongly associated with benchmark size (larger benchmarks can support more clusters), but exhibits **no statistically significant linear correlation with Harmony**. This lack of correlation indicates that our reliability measure is **not spuriously driven by the choice of $k$**, making our Harmony-based analysis robust to the choice of $k$.

---

> ### Author Response · Authors · 2025-11-15
> **Response to Reviewer 5RAY: Part 2**
>
> ## Dependence on Model Set and Generalist Assumption (W4)
>
> **TL;DR: We deliberately analyze Harmony of benchmarks using a diverse set of general-purpose LLMs (Llama, Qwen, Gemma, OLMo, Phi) that are *explicitly trained and released as generalists* rather than domain specialists.**
>
> We agree that not all benchmarks aim to measure generalist capability, and we do not claim otherwise. We do not interpret Harmony as a test of whether a benchmark is generalist, but as a diagnostic of whether the benchmark is *representative of its stated domain* for the evaluated model population.
>
> In our experiments, the models (Llama, Qwen, Gemma, OLMo, Phi) are all released as **broadly capable, general-purpose LMs** trained on large, diverse corpora and evaluated on a wide range of tasks. Thus, it is natural to ask whether such models exhibit uniform performance on standard benchmarks. For domain-specialist models, Harmony remains *useful*, as low Harmony in such settings highlights that performance is concentrated in certain subdomains, effectively identifying the *peaks of that model’s expertise*.
>
> Harmony itself is defined per model-benchmark pair and we summarize these per-model Harmony values via means and variances over a fixed population of generalist LLMs. Thus, our claims about benchmark reliability are explicitly made with respect to a specified model set, not as an absolute property of the benchmark. As with accuracy, cross-study comparisons of Harmony are meaningful when there is sufficient overlap in model sets and evaluation protocols.
>
> We acknowledge the possibility of cherry-picking model sets, but this concern applies to *any* benchmark metric (including accuracy). Since Harmony can be easily recomputed by researchers on their own model suite, a favorable set of models for a benchmark in one study does not prevent others from revealing different Harmony profiles. In our work, we **mitigate** cherry-picking concerns by (i) using a diverse collection of widely used, generalist models and (ii) fully reporting the model set and implementation details.
>
> ## Clarifying Benchmark Size and Correlation of Harmony Mean and Variance (W5 & Q1)
>
> We thank the reviewer for raising this point. We have revised Figure 3 to make benchmark size easier to interpret and now explicitly report Pearson correlation between Harmony mean and Harmony variance (r=−0.785 with p=0.0001).
>
> ## Lack of Systematic Analysis for Cluster Characteristics (Q2)
>
> We thank the reviewer for this suggestion. In addition to the *in-depth analyses* we already perform for other components of our study (e.g., predictive similarity, multi-dimensional evaluation, pruning), we now provide a **dedicated analysis of cluster characteristics** (App. N and O). Our analysis shows that Harmony is **not** trivially determined by cluster count or benchmark size.
>
> ## Semantic Coherence of Clusters and Further Validation (Q3)
>
> **TL;DR: Across Apps. B.2–B.4, predictive similarity consistently outperforms seven alternative similarity metrics in measuring semantic similarity, recovering semantic domains, and capturing Harmony. We extend our validation to two additional benchmarks with ground-truth domains (M3Exam and GPQA).**
>
> We definitely agree it is important to verify that Harmony clusters correspond to meaningful semantic categories. In the updated manuscript, we provide qualitative examples (App. M). We also commit to releasing the *full set of clusters* for all benchmarks.
>
> Moreover, we extend our validation beyond MMLU and RedundantQA to M3Exam [2] and GPQA [3] in App. B.3 and B.4 to further assess generalization across benchmarks with ground-truth domain partitions. Our findings show that predictive similarity **remains the best alternative among several baselines**.
>
> Regarding the requested ablation for random clustering, we want to emphasize that our paper already compares predictive similarity based clustering against seven strong baselines, and predictive similarity outperforms all of them in three aspects (App. B.2-B.4). These are substantially stronger and more semantically informed than random clustering. If the reviewer still deems it important, we would be more than happy to include random clustering as a baseline.
>
> Thank you again for your insightful review. **We would be happy to address any remaining questions or concerns you may have, and we hope that our clarifications and revisions merit a higher evaluation.**
>
> References:
>
> [1] Silhouettes: A graphical aid to the interpretation and validation of cluster analysis. 1987. https://www. sciencedirect.com/science/article/pii/0377042787901257.
>
> [2] M3Exam: A multilingual, multimodal, multilevel benchmark for examining large language models. 2023. arXiv preprint arXiv:2306.05179.
>
> [3] Gpqa: A graduate-level google-proof q&a benchmark. 2023. https://arxiv.org/abs/2311.12022
>
> [4] Probing the decision boundaries of in-context learning in large language models. 2024. https://arxiv.org/abs/2406.11233.

---

> ### Author Response · Authors · 2025-11-22
>
> Dear Reviewer 5RAY,
>
> We are truly grateful for your constructive feedback. If anything is still unclear or you would like us to expand on any point, we would be more than happy to do so. We also hope that our clarifications and revisions address your concerns and merit a higher evaluation.

---

### Official Review · Reviewer_EZ1V · 2025-11-02

**Soundness:** 4
**Presentation:** 4
**Contribution:** 3
**Rating:** 6
**Confidence:** 3

**Summary:**

This paper introduces HARMONY, an entropy-based metric that measures how uniformly a model’s performance is distributed across subdomains of a benchmark. It aims to reveal when aggregate accuracy obscures uneven competency.

**Strengths:**

- The paper addresses an important and under-discussed issue in benchmark-based evaluation: the degree to which average scores mask uneven performance across subdomains.
- The authors present a sufficiently large-scale empirical study across many model families and benchmarks.
- Overall, the depth of analysis and in particular how confounding factors influence the findings was great. The authors made a significant effort to provide the reader with all necessary information to understand their methodology, along with all necessary information to arrive at genuine and robust interpretations of the findings (e.g., by reporting statistical significance in plots).

**Weaknesses:**

- The paper does not clearly define what “benchmark reliability” means and uses the term in ways that overlap with concepts such as validity without clarifying the distinction.
- Relatedly, The paper mentions external valdity in the related work section but validity more broadly is even more relevant in this context in my opinion, see for example [1] and [2] (but also many others, many of Hannah Wallach’s work is relevant here!)
- Section 2.3 is hard to follow. Given that the partition induction & clustering are key parts of the approach, I would kindly ask the authors to provide some more context (and intuition) why they took the design decisions they took. Some open questions to me are: Why should the similarity be model-aware? Doesn't that mean that HARMONY of a benchmark changes depending which models are used to assess HARMONY? Isn't this easily gameable by developers? Why did the authors think that spectral clustering was preferable over other clustering methods? Why did they set 2<=k<=20 and decided that maximizing the silhouette score was a prudent choice? Adding some explanation what they hope this expresses would help the reader's understanding.
- Some citations are missing throughout the text, e.g., general statements like the first sentence in line 032 which should be supported by a reference (as an example, a potential citation here could be [3]). Same with the second sentence that follows right after. Overall, there are a couple of instances like this in the paper where I recommend adding more references.
- I would have liked to see more of a discussion how HARMONY scores should affect benchmark design or what the practical takeaway for benchmark designers should be. I.e., is the authors' standpoint that it's better to not have any general benchmarks that span multiple domains in the first place? I.e., should all benchmarks be specialized? This also seems to cut into a lot of validity discussions, i.e., it's much harder to design a valid benchmark for a broad abstract construct (e.g., "undergraduate-level knowledge" like MMLU than for a very specific domain (e.g., fundamentals of astronomy) and I would expect that HARMONY for the former would be much harder to achieve than for the latter.

Small nits that affected presentation score:
- The circle sizes in the legend in Figure 2 are visually almost indistinguishable.


[1] https://arxiv.org/abs/2505.10573
[2] https://arxiv.org/pdf/2412.01934
[3] https://dl.acm.org/doi/pdf/10.1145/3708359.3712152?casa_token=dsjgmGDH8AUAAAAA:CvbRJT-NGxwBc9vr3RegWENQzOPbfHXz78JCBN1UlsuiPIPTenGWWcYB2XS0FRVEFjpFk24ZpHDg

**Questions:**

See weaknesses.

I'm generally willing to updating my score if the points in the weaknesses section (in particular on reliability definition and the difference to validity and Section 2.3 questions) are clarified.

---

> ### Author Response · Authors · 2025-11-17
> **Response to Reviewer EZ1V: Part 1**
>
> We thank the reviewer for their insightful review and greatly appreciate highlighting the importance of our research question, the scale of our study, and especially our effort to support genuine and robust interpretations of the findings. Below, we address all the points raised by the reviewer directly. We have also updated the manuscript with changes highlighted in purple, which we reference throughout this response.
>
> ## Benchmark Reliability vs. Validity (W1&W2)
> **TL;DR: We define reliability as how well the aggregate benchmark score reflects performance across benchmark subdomains, whereas validity asks whether the benchmark measures the intended skills and justifies the inferences. Reliability is a necessary, but not sufficient, condition for validity.**
>
> Thank you for raising this concern and pointing us to [1], [2], and related work by Wallach. In our work, we use benchmark reliability in a *distributional sense*: a benchmark is *reliable* if its aggregate metric is a representative summary of performance over its stated domain. We operationalize this with Harmony, our entropy-based measure of how evenly accuracy is distributed across subdomains.
>
> This notion is **related to, but distinct from**, validity. Validity frameworks ask whether benchmark scores genuinely measure *the intended skills* and support the intended inferences from the evaluations. In contrast, we focus on whether the *internal aggregation* is *representative* by questioning whether the single benchmark score faithfully reflects performance across its subdomains. Therefore, when aggregate score is *unreliable* in this sense, the validity of conclusions drawn from the benchmark can be undermined. However, high Harmony alone is **a necessary, not a sufficient,** condition to establish full validity.
>
> In the revised manuscript, we have (i) tightened our definition of benchmark reliability (Sec. 2.2), (ii) situated this notion within standard validity frameworks, emphasizing that our method is intended to *complement, not replace*, broader validity-oriented analyses of benchmarks (Sec. 5), and (iii) expanded the related work section to discuss [1], [2], and Wallach’s work.
>
> ## Context and Intuition for Partition Induction (W3)
>
> **TL;DR: Harmony is model-aware yet remains comparable across models, as induced partitions are empirically consistent between models. It is not easier to game than standard evaluation metrics. We use spectral clustering and select $k$ via silhouette score maximization to obtain a small number of compact, well-separated subdomains.**
>
> Regarding model dependence: One could compute Harmony using a single, model-agnostic partition. We instead use model-aware partitions because *different model families draw different task-relevant boundaries in input space* [4]. Hence, a partition that aligns with one model can be misaligned with another, smoothing that model’s concentrated errors across subsets, which makes it appear *harmonious*. By grouping data points that a given model treats as interchangeable, Harmony becomes sensitive to model-specific fragility that a fixed partition might obscure. However, in practice, our experiments in App. B.5 show that predictive similarity partitions are largely **aligned** across models, so **model-dependent similarity does not undermine cross-model comparability**.
>
> Regarding gameability: Harmony for a benchmark-model pair is high when a model’s subdomain accuracies are similar to the overall benchmark accuracy. To artificially inflate it without *genuinely improving robustness*, one would need to make the model behave similarly on large neighborhoods of examples, which is **not achievable by tweaking a few hand-picked items**. In practice, Harmony can only be increased by (i) *improving weak subsets*, which is desirable, or (ii) *degrading strong subsets*, which necessarily lowers overall accuracy. Since we always report Harmony together with accuracy (and recommend doing so), such trade-offs would be **transparent**. Moreover, since we use Harmony values from a wide range of models, benchmark-level analysis with multiple models becomes **harder to *game*** than a single benchmark-model pair.
>
> Regarding the choice of clustering algorithm: We chose spectral clustering because our similarity measure defines a dense affinity graph over data points. Spectral methods are *specifically designed* to find well-separated groups in such graphs, even when clusters are non-spherical or vary in size [2]. In contrast, alternatives like k-means on raw representations implicitly assume roughly spherical clusters in Euclidean space and performed **worse** in preliminary experiments at recovering known subdomains. We added a short explanation of this graph-based motivation to Sec. 2.3.

---

> ### Author Response · Authors · 2025-11-17
> **Response to Reviewer EZ1V: Part 2**
>
> Regarding the choice of silhouette score and range of $k$: We restrict $k$ to $2\le k\le20$ because we are interested in **coarse subdomains** rather than *tiny* clusters. $k=1$ would make Harmony degenerate (only one subset), while larger $k$ values quickly produce clusters with very few items, which makes per-cluster accuracy estimates noisy and Harmony unstable. Within this range, we select $k$ by maximizing the silhouette score, a standard clustering criterion that compares how close each point is to its own cluster versus the nearest other cluster [3]. Maximizing this score yields partitions that are both **compact and well separated**, which is *exactly* what we need for each cluster to behave as a *coherent* subdomain.
>
> ## Missing Citations (W4)
> We thank the reviewer for pointing this out. In the revised manuscript, we added appropriate citations to support our general statements and claims (e.g., lines 33, 48, 278, etc.).
>
> ## Interpretation of Harmony and Implications for Benchmark Design (W5)
>
> **TL;DR: Harmony is a diagnostic of whether aggregate scores are representative of performance across subdomains. It surfaces trade-offs between breadth and uniformity, complements evaluation validity, and leaves decisions about redesign and specialization to benchmark practitioners.**
>
> We view Harmony primarily as a methodology to quantify how representative aggregate scores on a benchmark are, not as a prescription to abandon general benchmarks. A high Harmony value indicates that most dataset mass lies on subdomains whose performance is close to the benchmark mean, so an aggregate score is reasonably representative. In contrast, a low Harmony value indicates that a few subdomains dominate the aggregate score, so it should be interpreted with *caution*. Our position is that, especially for broad, multi-domain benchmarks, having an explicit domain partition is *highly valuable*, as it reduces dependence on a single aggregate score and allows practitioners to interpret model performances in a **finer-grained and fairer** way across subdomains.
>
> In this light, we do **not** advocate that all benchmarks be specialized or that multi-domain benchmarks should be abandoned. Multi-domain benchmarks are naturally harder to make harmonious than narrow, single-domain benchmarks. Therefore, we see Harmony as a tool that *surfaces* this trade-off. How to react to a given Harmony profile (e.g., by rebalancing items or embracing specialization) is ultimately at the discretion of benchmark designers.
>
> Finally, we see Harmony as **complementary** to validity discussions, since it provides an empirical lens on *how consistently the intended construct is reflected across subdomains* instead of replacing conceptual arguments about what the benchmark is supposed to measure. We have clarified this distinction more explicitly in the revised manuscript (Sec. 5).
>
> ## Circle Sizes in Figure 2 (W6)
>
> We thank the reviewer for raising this point. We have revised Figure 3 (previously Figure 2) to make benchmark size easier to interpret.
>
> Thank you again for your thoughtful review. **We would be happy to address any further questions or concerns you may have, and we hope that our clarifications and revisions warrant a higher evaluation.**
>
> References:
>
> [1] Probing the decision boundaries of in-context learning in large language models. 2024. https://arxiv.org/abs/2406.11233.
>
> [2] A tutorial on spectral clustering. 2007. https://arxiv.org/abs/0711.0189
>
> [3] Silhouettes: A graphical aid to the interpretation and validation of cluster analysis. 1987. https://www. sciencedirect.com/science/article/pii/0377042787901257.

---

> ### Author Response · Authors · 2025-11-22
>
> Dear Reviewer EZ1V,
>
> We are truly grateful for your constructive feedback. If anything is still unclear or you would like us to expand on any point, we would be more than happy to do so. We also hope that our clarifications and revisions address your concerns and merit a higher evaluation.

---

> > ### Comment · Reviewer_EZ1V · 2025-11-28
> >
> > Dear authors, your comments largely addressed my concerns and I appreciate the revisions in the updated PDF. I think that it would be great to add something along the lines of the above comment on the interpretation & implications of high/low HARMONY to the paper itself, but independent of this change, I would like to increase my score to an 8.
> >
> > @AC/@SAC: It seems like the score edit function on OpenReview was deactivated for some reason and I cannot change my official scores right now. Please note my score change here.

---

> > > ### Author Response · Authors · 2025-12-01
> > >
> > > We sincerely appreciate the reviewer updating their score and helping us clarify and strengthen our work. In line with their suggestions, we have further revised the manuscript to more clearly articulate the interpretation of Harmony and its implications, with all modifications marked in purple.

---

### Official Review · Reviewer_F1zp · 2025-11-03

**Soundness:** 2
**Presentation:** 3
**Contribution:** 2
**Rating:** 6
**Confidence:** 3

**Summary:**

Benchmarks critically shape model development, but their reliability is poorly understood. To this end, the authors introduced HARMONY, a metric quantifying how uniformly a model's performance is distributed across semantic subdomains of a benchmark. They partitioned benchmarks using "predictive similarity", a model-aware clustering based on KL divergence of model probability distributions. They then computed HARMONY (normalized Shannon entropy) for 36 models across 5 families on 19 MCQA benchmarks, positioning each benchmark in a mean-variance plane.

The paper's key contributions include the HARMONY metric, which measures performance uniformity across subdomains where higher values indicate more reliable benchmarks, and intorducing predictive similarity, a novel model-aware partitioning approach based on divergence of model logits. Testing 19 benchmarks, fragility evidence shows that low-HARMONY benchmarks have unstable aggregate scores under pruning while high-HARMONY benchmarks remain stable. They also test for model family specific patterns: Qwen and Llama show negative correlation between model size and HARMONY, while Gemma and OLMo show positive correlation.

**Strengths:**

* The problem motivation is timely and important: benchmarks critically shape model development, are needed for governance mechanisms, and audit methods are genuinely needed.

* The scale and comprehensiveness of the empirical study is thorough, evaluating 36 models across 5 families on 19 benchmarks with detailed ablations and cross-model analysis in the appendices.

* The authors design a synthetic benchmark (RedundantQA) with clean ground-truth partitions (paraphrases vs. distractors), enabling controlled validation of the similarity metric without confounds.

* The pruning experiments provide concrete evidence that low-HARMONY benchmarks are genuinely fragile and that their aggregate scores shift significantly when rebalancing, while high-HARMONY benchmarks remain stable. This is a great finding!

I like the paper,  but I have a few uncertainties that I would need to have sufficiently resolved before increasing my scores.

**Weaknesses:**

## Weakness 1

The paper defines HARMONY using normalized Shannon entropy but never justifies this choice over alternative uniformity measures (e.g., Gini coefficient, Rényi entropy, coefficient of variation). Appendix B compares variants of KL divergence, but not fundamentally different approaches to measuring performance uniformity. Can you please comment on why Shannon entropy is specifically the right measure for "uniformity of competence"? Is HARMONY more robust than other measures to gaming by benchmark designers?

## Wekaness 2

While theoretically grounded (app B), it's unclear whether the symmetrized KL divergence approach to model-aware similarity is genuinely novel or how it compares to existing model-based similarity metrics in the literature. The paper could be strengthend by making the novelty claim more explicitly justified.

## Weakness 3

The main text does not clearly specify how model-specific partitions are unified across 36 models to produce the mean-variance plane in Figure 2. It should explicitly state whether HARMONY is computed per-model then aggregated, or if a reference partition is used.
Can you comment on whether you compute 36 separate partitions (one per model) and then compute HARMONY for each or is a single reference partition used across all models?

## Weakness 4

The paper does not address whether the clustering procedure generalizes to benchmarks with overlapping/ambiguous semantic categories beyond the clean cases tested (RedundantQA, MMLU) with clearly separable domains. How does your method perform on benchmarks where categories have significant semantic overlap or ambiguous boundaries?

## Weakness 5

The paper asserts uniform performance indicates "broad competence" (2.2) but provides no rigorous justification for why this is desirable beyond empirical stability in Figure 5. Some legitimate task domains may have inherently heterogeneous difficulty. Also, the paper does not seem to distinguish between low HARMONY caused by poor benchmark design and low HARMONY reflecting genuine domain heterogeneity (e.g., some subdomains are objectively harder).

To give a concrete example: This MCQA dataset https://arxiv.org/abs/2502.16051 can be used to evaluate LM decision-making in clinical mental healthcare settings. The dataset is split across task categories (due to day-to-day clinical decision-making composing different tasks) that vary significantly in complexity and nature. Some categories even have no true answer (due to the inherent ambiguities even to human domain experts). Thus, model performances vary drastically across categories due to different task natures and conflicts with moral ambiguity/model alignment methods. This dataset (if used as a benchmark) would probabyl get a low Harmony score and I am not sure if that is a good thing and whether this problem will increase with increasingly more nuanced and domain-specific benchmarks. (I'm also happy to hear arguments how such benchmarks would be bad to strengthen the Harmony justificaiton)

Essentially, I think Harmony measures away to link accuracy/performance increase to a somewhat linear increase in performance. Why should we expect uniform performance across subdomains to be desirable? Aren't some domains legitimately harder than others and wouldn't enforcing harmony erase meaningful task heterogeneity? How do you distinguish between low HARMONY that indicates a design flaw versus low HARMONY that reflects legitimate variation in task difficulty across subdomains?

## Weakness 6

By computing model-specific partitions, HARMONY seems to measure whether a particular model has learned semantic structure, not whether the benchmark itself is well-designed. This could conflate model alignment with benchmark audit, potentially undermining the claim to benchmark reliability assessment. If HARMONY is model-specific, how is this a benchmark audit rather than a model audit? Couldn't a randomly initialized model coincidentally achieve high HARMONY on arbitrary clusters, implying the benchmark is "reliable"? Also, to what extent can a model's predictive similarity be manipulated to inflate harmony?

## Weakness 7

The paper recommends "report HARMONY alongside accuracy" but provides no concrete thresholds, decision rules, or guidance for how practitioners should act on this information (e.g., when to disqualify benchmarks, how to weight accuracy vs. harmony tradeoffs). What are the practical decision rules? Should benchmarks with HARMONY < 0.5 be excluded? How should practitioners trade off accuracy improvements against harmony decreases (Figure 5)?

## Weakness 8

Appendix H demonstrates hierarchical multi-dimensional evaluation but provides no principled criterion for when to stop recursively partitioning (e.g., you could partition MMLU Biology down to single-question granularity). This could undermine the generality of the framework. At what point will using HARMONY lead to evaluate benchmarks just to calculating HARMONY of increasingly narrow subcategories and thus loosing its value to make a statement about an entire benchmark?

**Questions:**

I've embedded my questions into the weaknesses.

---

> ### Author Response · Authors · 2025-11-18
> **Response to Reviewer F1zp: Part 1**
>
> We sincerely appreciate the reviewer’s insightful evaluation and thank them for their positive assessment of our motivation, the scale and comprehensiveness of our experiments, and the value of RedundantQA and the pruning experiments. Below, we address each concern raised by the reviewer in detail. We have also updated the manuscript with changes highlighted in orange, which we reference throughout this response.
>
> ## Formulation Choice for Harmony (W1)
>
> We thank the reviewer for raising this. In the revised manuscript, we provide discussion for our formulation choices for Harmony in App. C, including a comparison between normalized Shannon entropy and the alternatives pointed out by the reviewer.
>
> ## Novelty and Comparison to Existing Similarity Metrics (W2)
>
> **TL;DR: Predictive similarity is methodologically distinct from prior similarity measures and consistently outperforms seven strong baselines.**
>
> We thank the reviewer for raising the question of novelty. In App. B.1 we define seven alternative similarity metrics, including both model-dependent and model-independent approaches, and *thoroughly* evaluate them in App. B.2–B.4. Across these experiments, predictive similarity **consistently outperforms all baselines** in (i) capturing semantic similarity (App. B.2), (ii) inducing coherent semantic partitions (App. B.3), and (iii) reflecting ground-truth Harmony (App. B.4).
>
> To the best of our knowledge, prior model-based similarity metrics primarily operate on intermediate representations (e.g., embeddings, gradients), whereas our approach is based on the full output probability distributions. This **methodological difference** combined with its **consistent empirical advantages** over strong baselines *supports our claim of novelty*. We have updated App. B.1 to clarify this argument.
>
> ## Specification of Aggregating Harmony (W3)
>
> Thanks for pointing this ambiguity out. We separately induce partitions for each model in our experimental setup, compute Harmony for each, and then compute the mean and variance of these Harmony scores. We have clarified this in Sec. 3 in the revised manuscript.
>
> ## Generalization to Hard-to-Partition Benchmarks (W4)
>
> **TL;DR: Our clustering method produces coherent partitions even in semantically dense, overlapping domains (e.g., MMLU Biology and deeper levels in hierarchical clustering), demonstrating robustness to ambiguous benchmark categories.**
>
> Thank you for emphasizing this important concern. Beyond recovering the underlying domain partitions in MMLU and RedundantQA, our clustering method successfully partitions *semantically dense* subsets such as MMLU Biology and Physics (App. I and J), which are *harder to decompose due to substantial semantic overlap*. This suggests that our approach does **not** rely on *clean and clearly separable* categories, but is robust to settings with overlapping or ambiguous semantics.
>
> In the updated manuscript, we provide qualitative examples of the induced clusters (App. M), drawn from deeper levels of the hierarchical clustering where clusters are more specialized and thus harder to partition. We find that our method continues to produce *coherent clusters* even at these *lower depths*, which provides a **stringent test of its ability to handle overlapping and ambiguous categories**. We further commit to releasing the *full set of clusters* for all benchmarks.
>
> ## Justification of Desirability of High Harmony (W5)
>
> **TL;DR: Harmony does not assume all subdomains are equally easy.  It quantifies when a single aggregate score can be interpreted as broad competence. Low Harmony instead warns that per-subdomain analysis is needed and thus helps preserve, not erase, task heterogeneity.**
>
> Thanks for raising this point. As discussed in Sec. 2.2, our claim is not that all subdomains should be equally easy. Rather, we observe that when a benchmark is summarized by a single aggregate score, that score is typically interpreted as reflecting broad competence across its semantic subdomains. Harmony makes this implicit assumption explicit and **measures how valid it is**.
>
> Consider the MENTAT dataset: an LM obtains a high overall score across diagnosis, treatment, and other categories. In practice, this single number will be used in model selection and may lead practitioners to believe the model is broadly suitable for mental healthcare settings. Yet, the aggregate score might be *inflated* by diagnosis questions while the model *performs poorly on treatment decisions or morally ambiguous cases*, which are the scenarios where misuse would be most **harmful**. Harmony would be low in this case, **signaling that the aggregate score is not representative of performance across *critical* subdomains and that practitioners should inspect finer-grained metrics before deployment.**

---

> ### Author Response · Authors · 2025-11-18
> **Response to Reviewer F1zp: Part 2**
>
> Importantly, Harmony is agnostic to why performance is uneven. Low Harmony can stem from poor benchmark design (e.g., unbalanced coverage) or genuine heterogeneity in task difficulty. In both cases, the single aggregate metric is an **unreliable proxy for broad competence,** and that is what Harmony is *designed to diagnose*. Disentangling design flaws from domain heterogeneity requires domain expertise and is beyond the scope of the metric itself.
>
> ## Distinction between Model-Specific and Benchmark-Level Harmony (W6)
> **TL;DR: Harmony is computed per model, but benchmark reliability is defined by its distribution across many models. A single model with high Harmony neither makes a benchmark reliable nor changes our conclusions.**
>
> Thank you for emphasizing this distinction. Harmony is computed for *each* model and our benchmark conclusions are based on the *aggregation* of Harmony over a model suite. Harmony is a benchmark-model interaction statistic, and the reliability of a benchmark is assessed by how this statistic behaves across many independent models. We view a benchmark as more *reliable* when **most models distribute their performance uniformly across induced subdomains**, yielding high mean and low variance in Harmony.
>
> A randomly initialized model could, *in principle*, have roughly uniform accuracy across subdomains and thus a high Harmony. However, in our analyses we (i) restrict the model set to a large collection of widely used, general-purpose models (Llama, Qwen, Gemma, Phi, OLMo) and (ii) always interpret Harmony together with accuracy, and recommend doing so. Therefore, **a single model that is both near-chance and high Harmony does not lead us to declare a benchmark reliable.**
>
> To artificially inflate Harmony without *genuinely improving robustness*, one would need to equalize performance over large neighborhoods, which is **difficult to achieve by tweaking a few items**. Harmony can only be increased by (i) *improving weak subsets* or (ii) *degrading strong subsets*. Since we always report Harmony together with accuracy, such trade-offs would be **transparent**. Moreover, since we use Harmony values from a wide range of models, benchmark-level analysis with multiple models becomes **harder to *game*** than a single benchmark-model pair.
>
> ## Practical Decision Rules for Using Harmony (W7)
> **TL;DR: Harmony is intended as a diagnostic complement to accuracy, not a hard filter. We recommend using it to (i) prefer models that do not sacrifice Harmony for small accuracy gains and (ii) treat benchmarks whose Harmony improves significantly under pruning with caution.**
>
> We agree that it is important to make *proper* use of Harmony in future model evaluations. Our goal is to introduce Harmony as **an additional axis of reliability** rather than *to prescribe a single global cutoff* (e.g., exclude all benchmarks with Harmony $< 0.5$), since acceptable levels should be determined *case-by-case*.
>
> We envision two concrete decision patterns:
>
> 1- Use the accuracy-Harmony plane and prefer models with higher Harmony among models with comparable accuracy. When accuracy gains with Harmony drops indicates that the **model is over-specializing to certain subdomains**.
>
> 2- Run the pruning-based stress test we introduce. If moderate pruning **increases Harmony** and **leads to statistically significant changes** in model performance, this indicates that the benchmark is *fragile*. We suggest that such benchmarks should be treated with **caution**.
>
> ## Stopping Criteria in Hierarchical Harmony Evaluation (W8)
> **TL;DR: Harmony permits arbitrary hierarchical partitioning, but in our experiments we use depth-1 partitions for all quantitative claims, and treat deeper splits only as optional diagnostics.**
>
> Following the methodology of [1], our implementation allows the benchmark to be recursively partitioned down to single-question granularity. However, the results in App. J (Tables 8–25) are **all computed at depth-1** (i.e., the partition of each benchmark into its subdomains). The goal of App. I and J is precisely to **inspect how different Harmony scores manifest at a finer level of granularity**, *not to redefine* the benchmark as a collection of arbitrarily small sub-benchmarks.
>
> We recommend stopping refinement once (i) the nodes correspond to *semantically meaningful categories*, and (ii) each node still contains **a non-trivial number of questions** rather than a few items. This way, the primary quantity remains the benchmark-level Harmony, while deeper partitions serve to explain why a benchmark is stable or unstable, not to fragment it into narrower evaluations.
>
> Thank you again for your careful review. **We would be happy to address any remaining questions or concerns, and we hope that our clarifications and revisions merit a higher evaluation.**
>
> References:
>
> [1] EvalTree: Profiling Language Model Weaknesses via Hierarchical Capability Trees. 2025. https://arxiv.org/abs/2503.08893

---

> ### Author Response · Authors · 2025-11-22
>
> Dear Reviewer F1zp,
>
> We are truly grateful for your constructive feedback. If anything is still unclear or you would like us to expand on any point, we would be more than happy to do so. We also hope that our clarifications and revisions address your concerns and merit a higher evaluation.

---

> > ### Comment · Reviewer_F1zp · 2025-11-27
> >
> > Dear authors,
> >
> > Thank you for your detaield responses and, in particular, for uploading a thorough rebuttal revision, making it easy to see the improvements and adjustments. All of my concerns have been either resolved or at least addressed within a reasonable scope for conference publication (meaning you should not suffer from my scientific curiosity). I've increased my scores accordingly (overall 6 -> 8).
> >
> > Regarding Weakness 5: I see your point and your clarification was helpful. I would propose to adjust the framing in section 2.2 to better communicate this distinction. In particular, I agree with “low harmony should lead to more nuanced performance analysis” and be less normative about “high harmony is desirable”, as I am still not sure if that is true. I buy into the statement “low harmony requires a closer look”, if that makes sense.
> >
> > For example, lines 109-110 state _“A harmonious benchmark therefore mitigates these distortions by balancing coverage and promoting comparable performance across subdomains.”_, which I think could be strengthened by essentially adding these sentences from your rebuttal comment _“Low Harmony can stem from poor benchmark design (e.g., unbalanced coverage) or genuine heterogeneity in task difficulty. In both cases, the single aggregate metric is an unreliable proxy for broad competence, and that is what Harmony is designed to diagnose. Disentangling design flaws from domain heterogeneity requires domain expertise and is beyond the scope of the metric itself.”_ Happy to discuss this.

---

> > > ### Author Response · Authors · 2025-12-01
> > >
> > > We sincerely appreciate the reviewer updating their score and once again thank them for helping us clarify and improve our work. Following their suggestion, we have revised Sec. 2.2 to emphasize that Harmony is a diagnostic tool indicating when aggregate scores may be misleading, rather than a normative preference for uniformly high Harmony. In particular, we have incorporated the reviewer’s proposed clarification (“Low Harmony can stem…”) to explicitly distinguish between benchmark design issues and genuine domain heterogeneity. We thank the reviewer again for helping us improve the precision and clarity of our presentation.

---

### Author Response · Authors · 2025-12-03
**Summary of Revisions and Reviewer Discussions: Part 1**

We would like to thank all reviewers for their thoughtful feedback and engagement with our submission. We briefly summarize how the reviews evolved and how our revisions respond to the key concerns raised by the reviewers.

# Discussion with Reviewer `85MX`
Reviewer `85MX` raised several important questions. Below, we summarize each issue and how we resolve them.

> Pruning Formula and Hyperparameter Robustness

The reviewer questioned whether our experiment design choices might explain the observed fragility patterns. In response, we ran two new experiments:
- Hyperparameter Sweeps: We systematically varied every hyperparameter in the pruning ratio formula. Qualitative outcomes changed in **at most 3 out of 30 settings, and usually not at all**. In each case, low Harmony benchmarks *continued to show significant post-pruning shifts*, whereas high Harmony benchmarks *remained stable*.
- Fixed Ratio Pruning (App. P): We additionally pruned all benchmarks at uniform global rates (10%, 20%, 30%, 40%, 50%). Across **all** ratios, low Harmony benchmarks *remained fragile*, while high Harmony benchmarks *remained stable*.

After these experiments, the reviewer **did not raise any further questions** on this issue. It is our understanding that these *additional analyses satisfactorily addressed their concern*.

> Interpretation and Value of Harmony

The reviewer questioned what Harmony offers beyond reporting subdomain scores. We clarified that harmony is a scalable diagnostic metric that enables comparison of reliability across many benchmarks and models simultaneously, where manually inspecting subdomain scores becomes impractical. After our clarification, the reviewer **did not pursue this concern further**. Our impression is that *their question about the value of Harmony was resolved to their satisfaction.*

> Difficulty vs. Semantic Partitions

The reviewer raised a concern about whether *semantic domains* can be separated from difficulty. We clarified that:
- Harmony is defined with respect to semantic subdomains, not difficulty.
- When semantic subdomains do systematically differ in difficulty, this is **precisely** the situation where a **single aggregate score becomes misleading**.
- In such cases, low Harmony is **a desirable warning signal**, indicating that practitioners should examine subdomain-level performance before drawing conclusions about broad competence.

After our clarification, the reviewer **did not return to this concern**. Our impression is that *our clarification addressed the core of their concern on this point*.

> Reliability of Harmony Estimation

The reviewer questioned whether our estimator reliably tracks ground truth Harmony across benchmarks. In response, we:
- **Extended validation** to four benchmarks (MMLU, RedundantQA, GPQA, M3Exam).
- Included an analysis combining all benchmarks, **showing cross-benchmark consistency.**

Afterward, the reviewer explicitly stated **“I agree there is clear linear signal”**, with their only remaining concern being whether *"preserves ordering"* is too strong a phrase.

Here we respectfully disagree, because the estimator shows a **consistently strong monotonic** relationship with ground truth Harmony both (i) **within each benchmark**, and (ii) **across benchmarks**. Given this *consistency*, we believe **Harmony estimation is reliable** for the comparative purposes for which it is intended. The reviewer **raised no methodological concerns** after the expanded validation.

> Formulation of Harmony (Entropy vs. Variance)

The reviewer requested a clearer justification for our Harmony formulation. In response, we added:
- A **detailed comparison** of entropy vs. variance and other alternatives (App. C).
- An explanation of design desiderata: *coverage, uniformity, normalization, and interpretability.*

Following this, the reviewer explicitly stated **”I understand your technical point regarding entropy vs. variance”**, and shifted to expressing a conceptual preference. It's our understanding that *the technical concern was resolved*, and only a philosophical disagreement remained.

---

> ### Author Response · Authors · 2025-12-03
> **Summary of Revisions and Reviewer Discussions: Part 2**
>
> > Summary
> - We addressed the **experimental and methodological concerns** raised by Reviewer `85MX` (pruning confounds, hyperparameter sensitivity, estimator reliability) with new experiments and analyses. *None of these were challenged again after our responses*.
> - The reviewer’s final reservations are *conceptual*, reflecting personal preferences about metric design rather than any unaddressed technical concerns.
>
> # Discussion with Reviewer `5RAY`
> Reviewer `5RAY` raised several concerns, in particular about:
> - Whether benchmark size and the number of clusters confound our findings.
> - Whether our partitioning approach produces meaningful clusters.
> - Whether model set dependence and pruning choices undermine the interpretation of Harmony.
>
> Although the reviewer **did not re-engage after our detailed response**, we **took these concerns seriously** and designed a series of targeted analyses to directly address their concerns.
>
> > Controlling for Benchmark Size and Cluster Counts
>
> In the revised manuscript (App. N, O), we perform a principled analysis and show that:
> - There is **no statistically significant correlation** between *benchmark size and either the mean or variance of Harmony*.
> - There is **no statistically significant correlation** between *number of clusters and Harmony*.
>
> Together, **these analyses rule out benchmark size and the choice of k as confounding factors**. We believe this *conclusively addresses the reviewer’s concern* on this point.
>
> > Clustering Quality
>
> In App. B, we define seven baselines and show that **predictive similarity and spectral clustering consistently outperforms all of them** in (i) capturing semantic similarity, (ii) recovering the ground truth domains, and (iii) tracking the ground truth Harmony.
>
> We also **extend validation beyond** MMLU and RedundantQA to GPQA and M3Exam, where the predictive-similarity + spectral-clustering combination **remains the strongest performer**.
>
> Finally, we provide qualitative cluster examples in App M, illustrating that **the induced clusters are semantically coherent**. Taken together with the results above, we believe this successfully addresses the reviewer’s concern and supports our claim that **our partitioning approach discovers meaningful structure**.
>
> > Model Set Dependence and Fragility Under Pruning
>
> We clarify that Harmony is computed per model, and benchmark level conclusions are drawn over a fixed, diverse suite of general purpose LLMs (e.g., Llama, Qwen), rather than hand-picked models.
>
> Our experiments in App. B.5 show that **predictive similarity partitions are highly consistent across models**, *addressing the concern that different model sets might induce incompatible or idiosyncratic partitions*.
>
> To address concerns about pruning and fragility, we add **extensive robustness analyses** (as discussed above for Reviewer 85MX).
>
> > Summary
>
> Although Reviewer `5RAY` did not participate in the discussion phase, we took their concerns seriously. In our revision, we responded by adding targeted analyses and new experiments to address these points. Specifically, we (i) ruled out benchmark size and cluster count as confounds through principled statistical analyses (Apps. N and O), (ii) showed that predictive similarity + spectral clustering consistently recovers semantic structure and tracks ground truth Harmony (App. B, M), and (iii) demonstrated that predictive similarity partitions are stable across model families and scales (App. B.5). We also added extensive pruning robustness analyses to ensure that our fragility results are not artifacts of the pruning procedure. We believe **these additions provide thorough, good-faith resolutions to the methodological issues they highlighted.**

---

### Meta-Review · Area_Chair_ZdpP · 2026-01-06

**Summary:**

This paper questions the average performance (as usually reported in papers) of LLMs on stratified benchmarks, i.e., benchmarks where one can find meaningful sub-domains of instances. To this extent, they introduce the HARMONY score, the normalized entropy of the scores of a model w.r.t. the sub-domains. The claim is that benchmarks with high mean and low variance HARMONY scores are more reliable.

Reviewers recognized that the discussing SoTA just in terms of aggregated performance (via averages) is indeed an important problem that can skew our understanding of progress. I totally agree with them. I am not sure if another aggregate metric is the solution however, especially if the HARMONY of a dataset is another average computed over only a few models. Another solution is reporting the full distribution of scores or more than one statistics (than the single entropy), which however requires careful plotting and discourages highlighting making definitive statements like X is better than Y. I better understand HARMONY as a way to filter out unreliable benchmarks, as it is a process that can be done once, after measuring it over many many more models than those used in the paper.

Concerning the methodology, reviews were polarized. Reviewers 85MX and 5RAY are the sceptical ones about the HARMONY metric and it being model-specific, and highlighted how other measures of uniformity could be (equivalently) good alternatives. On the other hand, reviewers F1zp and EZ1V are more supportive and asked for technical details missing that got provided by the authors in the rebuttal. I hear the concern of the skeptical reviewers, especially 85MX. I don't seem to see in Appendix C a comparison against other metrics, it would be useful to see what is the added benefit of normalized entropies.
As argued above, I am not exactly convinced on HARMONY as a score that should accompany model accuracy in a paper's table.

The paper is ultimately borderline, with both reasons to accept or reject. I believe the reasons to reject are slightly more and the paper can benefit from an additional cycle of revision, mostly clarifying why other uniformity metrics are not enough and better arguing in favor of another aggregate score.  Ultimately, this can be turned in a strong submission.

**Reviewer Concerns:**

EZ1V asked about missing details, rewriting of section that was not clear, and the distinction between reliability and validity of a benchmark. Authors sufficiently clarified this. One crucial question that has not been properly answered is "Doesn't that mean that HARMONY of a benchmark changes depending which models are used to assess HARMONY?" The authors answer focuses on different partitioning, but misses the point that to estimate the expectation in Eq 1, they have to do a Monte Carlo estimate over a bunch of models. Clearly, if one picks 5 models once and 5 different models then, the estimate can vary. More evidence needs to be used to see how much they vary.

The main unanswered concern of reviewer F1zp (although they are happy with the answer the authors provided) is that what happens with other measures of uniformity (I agree with this point being raised) which is the same for reviewer 85MX. The authors added appendix C, but they only discuss in words potential shortcomings. They should also add empirical evidence that in their experiments they cannot use these alternative scores instead of HARMONY.

5RAY major left concern is focused on "HARMONY, as a measure of a benchmark's reliability, is potentially confounded by the fact that benchmark HARMONY is aggregated over a model set and not all benchmarks are intended to measure generalist capability". The author's answer is that they are evaluating very large LLMs that claim to be generalist. This brings the perspective that HARMONY score is not a tool for all benchmarks and all models, but only for specific LLMs on certain stratified settings, which goes a bit against the overall claims in the paper. In any case, the Monte Carlo estimation issue discussed above, related to the concern of reviewer 5RAY,  is still not addressed.

**Reviewer Scores:**

Reviewer 85MX expressed his will to keep his score, not being convinced by the motivation and evaluation of HARMONY scores. They could have dampened the other reviewers' comments if a discussion happened.

5RAY did not engage in the rebuttal. They could have increased their score to 4 at least.

F1zp and EZ1V expressed their interest in increasing the score, but at the same time, they could have kept it at 6 after properly engaging with 85MX and 5RAY.

---

### Decision · Program_Chairs · 2026-01-26

Reject